# A Tight Convergence Analysis of Inexact Stochastic Proximal Point Algorithm for Stochastic Composite Optimization Problems

**Shulan Zhu**[1], **Chenglong Bao**[1,*], **Defeng Sun**[2], **Yancheng Yuan**[2,*]
[1]Tsinghua University
[2]The Hong Kong Polytechnic University

## Abstract

The **in**exact **s**tochastic **p**roximal **p**oint **a**lgorithm (isPPA) is popular for solving stochastic composite optimization problems with many applications in machine learning. While the convergence theory of the (inexact) PPA has been well established, the known convergence guarantees of isPPA require restrictive assumptions. In this paper, we establish the stability and almost sure convergence of isPPA under mild assumptions, where smoothness and (restrictive) strong convexity of the objective function are not required. Imposing a local Lipschitz condition on component functions and a quadratic growth condition on the objective function, we establish last-iterate iteration complexity bounds of isPPA regarding the distance to the solution set and the Karush–Kuhn–Tucker (KKT) residual. Moreover, we show that the established iteration complexity bounds are tight up to a constant by explicitly analyzing the bounds for the regularized Fréchet mean problem. We further validate the established convergence guarantees of isPPA by numerical experiments.

## 1 Introduction

We consider the following stochastic composite optimization problem:

$$\min_{x \in \mathbb{R}^d} \ \phi(x) \triangleq F(x) + r(x), \quad \text{where} \quad F(x) \triangleq \mathbb{E}_{s \sim P}\left[f(x; s)\right], \qquad \text{(ComOpt)}$$

where $\mathcal{S}$ is a sample space, $P$ represents a distribution over $\mathcal{S}$, $r \colon \mathbb{R}^d \to (-\infty, +\infty]$ is a proper and closed function, and for $P$-almost $s \in \mathcal{S}$, the component function $f(\cdot; s) \colon \mathbb{R}^d \to (-\infty, +\infty]$ is proper and closed, while the composite component function $f(\cdot; s) + r(\cdot)$ is proper and closed convex. A special case of problem (ComOpt) is the regularized finite-sum problem, where the sample space $\mathcal{S}$ is denoted by the finite discrete domain $\{1, \cdots, n\}$ for some positive integer $n$, the function $F$ is defined as a finite sum of component functions, and the regularizer $r$ is typically used to impose sparsity on the parameter $x$. Such discrete models are common in statistics and machine learning, including linear regression model with Lasso regularizer or elastic net regularizer.

### 1.1 Motivation and Related Works

Numerous stochastic first-order methods have been proposed and analyzed for solving the stochastic problem (ComOpt). The classical stochastic gradient descent (SGD) method developed by Robbins & Monro (1951) remains popular due to its simplicity. Nevertheless, theoretical and empirical results have shown that SGD suffers from instability and has significant difficulties and limitations in stepsize selection (Moulines & Bach, 2011; Asi & Duchi, 2019). While SGD is commonly used in practice, its drawbacks motivated developing alternative methods to alleviate these issues. One such method is the stochastic proximal point algorithm (sPPA) developed by Bertsekas (2011) for discrete problems and extended to continuous settings by Ryu & Boyd (2014). A significant advantage of sPPA over SGD is the stability established in (Asi & Duchi, 2019). Specifically, when

*Corresponding authors. Email: `clbao@tsinghua.edu.cn`, `yancheng.yuan@polyu.edu.hk`

each component function is convex and there exists a uniform upper bound on subgradients over the optimal solution set, sPPA exhibits the following property: if the stepsizes are square summable but not summable and the subdifferential exchange with the expectation, then the iterates generated by sPPA are bounded almost surely.

Several theoretical convergence results have been proven for sPPA under different assumptions. Common assumptions include (restricted) strong convexity (Ryu & Boyd, 2014; Patrascu & Necoara, 2018; Davis & Drusvyatskiy, 2019; Asi et al., 2020) and (Lipschitz) smoothness (Ryu & Boyd, 2014; Toulis et al., 2016; Patrascu & Necoara, 2018; Yuan & Li, 2023) or (global) Lipschitz continuity (Patrascu & Necoara, 2018; Davis & Drusvyatskiy, 2019) of component functions. Many other assumptions have also been proposed to obtain bounds of convergence rate for sPPA. For instance, when applied to problem (ComOpt) with weak linear regularity of $F$ and $r = 0$, Patrascu (2021) proved an asymptotic $\mathcal{O}(k^{-\beta})$ sublinear convergence rate of sPPA with diminishing stepsizes $\alpha_k = \alpha_0 k^{-\beta}$ in the expected squared distance to the optimal solution set. Some studies assumed "easy optimization" (Patrascu, 2021; Asi & Duchi, 2019; Asi et al., 2020), which requires that optimal solutions minimize each component function. However, this strong assumption is often violated in machine learning models. Most analyses considered the exact sPPA with a batchsize of one, which assumes the subproblem in each iteration is solved exactly. This is a strong assumption that is typically difficult to achieve in practice. The only analysis of inexact sPPA was conducted by Yuan & Li (2023), who established ergodic convergence under the assumption that $r$ satisfies Lipschitz continuity over the entire domain. Nevertheless, this condition is violated by the elastic net regularizer. We refer to Table 1 in Appendix A for a detailed comparison of these convergence rate results and their underlying assumptions.

Based on the above discussion, in this paper, we consider the **i**nexact **s**tochastic **p**roximal **p**oint **a**lgorithm (isPPA) and provide the corresponding convergence analysis. Specifically, we aim to:

- Establish convergence rate guarantees under assumptions that are satisfied by most common regularized regression models in practice, including linear regression and logistic regression, while also allowing for nondifferentiability and local Lipschitz continuity of component functions and the regularizer.

- Impose reasonable stopping criteria that can be implemented computationally for inexactly solving subproblems.

- Derive various versions of convergence rate bounds, with some possessing theoretical insight while others being practically meaningful for assessing algorithm termination.

We mainly focus on the bounds for last-iterate convergence rates due to: a) their easier extension to nonconvex settings than ergodic convergence rates, which rely strongly on convexity; b) last iterates preserving properties like sparsity versus averaged iterates.

## 1.2 CONTRIBUTIONS

The main contributions of this paper can be summarized as follows:

- Under mild conditions, we prove the stability of isPPA, extending the stability result for exact sPPA in literature (Asi & Duchi, 2019) to the inexact variants.

- Assuming a local Lipschitz condition on component functions and a quadratic growth condition on the objective function, on the event that the iterate sequence remains bounded, we derive nonasymptotic last-iterate convergence rates for isPPA in terms of the expected squared distance to the optimal solution set. Specifically, this method converges linearly to an $\mathcal{O}(\alpha_0)$ neighborhood of the optimal solution set with constant stepsizes $\alpha_0$, and can achieve an $\mathcal{O}(k^{-\beta})$ asymptotic rate with diminishing stepsizes $\alpha_k = \alpha_0 k^{-\beta}$ ($\beta \in (0, 1]$). Based on these results, we can further obtain the corresponding rates in terms of the expected Karush–Kuhn–Tucker (KKT) residual. Since the KKT residual is commonly used in algorithm termination criteria, these rates have clear practical applications.

- By using the Fréchet mean problem with squared $\ell_2$-norm regularizer as an illustrative example, we provide lower bounds on the convergence rates of isPPA for solving problem (ComOpt) under the assumptions proposed in this paper, demonstrating that our derived convergence rate guarantees are tight up to constant factors.

Additionally, we verify the theoretical results by preliminary numerical experiments.

## 1.3 NOTATIONS AND BASIC ASSUMPTIONS

Denote the set $[n] \triangleq \{1, \cdots, n\}$. For any $x \in \mathbb{R}^d$, denote its $q$-norm as $\|x\|_q$. For any $x \in \mathbb{R}^d$ and $\mathcal{X} \subset \mathbb{R}^d$, let $\operatorname{dist}(x, \mathcal{X}) \triangleq \inf_{y \in \mathcal{X}} \|x - y\|_2$ be the Euclidean distance from $x$ to $\mathcal{X}$, and set $\operatorname{proj}(x, \mathcal{X})$ to be the projection of $x$ onto $\mathcal{X}$ if $\mathcal{X}$ is nonempty and closed convex. For any proper and closed convex function $p \colon \mathbb{R}^d \to (-\infty, +\infty]$, the subdifferential of $p$ at $x$ is denoted by $\partial p(x)$, and the proximal mapping $\operatorname{prox}_p(\cdot)$ associated with $p$ is defined by

$$\operatorname{prox}_p(x) \triangleq \arg \min_{y \in \mathbb{R}^d} \left\{ p(y) + \frac{1}{2} \|y - x\|_2^2 \right\} \quad \text{for all } x \in \mathbb{R}^d.$$

It is known from (Rockafellar, 1976b, Proposition 1) that $\operatorname{prox}_p(\cdot)$ is nonexpansive, i.e., Lipschitz continuous with constant 1. Consider the stochastic composite optimization problem (ComOpt). Let $\mathcal{X}^* \triangleq \arg \min_{x \in \mathbb{R}^d} \phi(x)$ denote the optimal solution set and set $\phi^* \triangleq \inf_{x \in \mathbb{R}^d} \phi(x)$. For each $s \in \mathcal{S}$, the composite component function $\varphi(\cdot; s) \colon \mathbb{R}^d \to (-\infty, +\infty]$ is defined by

$$\varphi(x; s) \triangleq f(x; s) + r(x) \quad \text{for all } x \in \mathbb{R}^d,$$

where $f(\cdot; s)$ and $r(\cdot)$ denote the component function and regularizer of problem (ComOpt), respectively. Then, the objective function $\phi(x) = \mathbb{E}_{s \sim P}[\varphi(x; s)]$ for all $x \in \mathbb{R}^d$. For each $k \in \mathbb{Z}_+$, we use $S_k^{1:m} \triangleq \{S_k^i\}_{i=1}^m \subset \mathcal{S}$ to denote a random minibatch of size $m$ and define

$$\overline{f}(x; S_k^{1:m}) \triangleq \frac{1}{m} \sum_{i=1}^m f(x; S_k^i) \quad \text{for all } x \in \mathbb{R}^d, \tag{1.1a}$$

$$\overline{\varphi}\left(x; S_k^{1:m}\right) \triangleq \frac{1}{m} \sum_{i=1}^m \varphi(x; S_k^i) = \overline{f}\left(x; S_k^{1:m}\right) + r(x) \quad \text{for all } x \in \mathbb{R}^d. \tag{1.1b}$$

To solve problem (ComOpt), we consider the isPPA method shown in Algorithm 1. The symbol "$\overset{\epsilon_k}{\approx}$" indicates that the next iterate $x_{k+1}$ is obtained by approximately solving the following subproblem:

$$\min_{x \in \mathbb{R}^d} \overline{f}(x; S_k^{1:m}) + r(x) + \frac{1}{2\alpha_k} \|x - x_k\|_2^2$$

until $x_{k+1}$ satisfies an accuracy $\epsilon_k$ specified by some stopping criterion (to be defined in Section 2). To establish the convergence properties of isPPA (Algorithm 1), we make the following assumptions.

**Assumption 1.** *The function $F \colon \mathbb{R}^d \to \mathbb{R}$ is locally Lipschitz and $r \colon \mathbb{R}^d \to (-\infty, +\infty]$ is a proper closed function. For $P$-almost all $s \in \mathcal{S}$, $f(\cdot; s) \colon \mathbb{R}^d \to (-\infty, +\infty]$ is a proper closed function satisfying $\operatorname{dom}(f(\cdot; s)) \supset \operatorname{dom}(r)$, and $\varphi(\cdot; s) \colon \mathbb{R}^d \to (-\infty, +\infty]$ is convex.*

**Assumption 2.** *The optimal solution set $\mathcal{X}^*$ is nonempty. There exists a scalar $\sigma_\phi \in \mathbb{R}_{++}$ such that*

$$\mathbb{E}_{s \sim P}\left[\|\varphi'(x^*; s)\|_2^2\right] \leq \sigma_\phi^2 \quad \text{for all } x^* \in \mathcal{X}^* \text{ and all measurable selections } \varphi'(x^*; s) \in \partial\varphi(x^*; s).$$

*For any fixed $x^* \in \mathcal{X}^*$ and $\phi'(x^*) \in \partial\phi(x^*)$, there exists a measurable mapping $\varphi'(x^*; \cdot)$ such that*

$$\mathbb{E}_{s \sim P}\left[\varphi'\left(x^*; s\right)\right] = \phi'(x^*) \quad \text{with} \quad \varphi'\left(x^*; s\right) \in \partial\varphi(x^*; s) \text{ for } P\text{-almost } s \in \mathcal{S},$$

*where $\mathbb{E}_{s \sim P}\left[\varphi'\left(x^*; s\right)\right]$ is defined as the integral $\int_{\mathcal{S}} \varphi'(x^*; s) \mathrm{d} P(s)$.*

**Assumption 3.** *For any bounded open subset $U \subset \operatorname{dom}(r)$, there exists a measurable function $L_{f,U} \colon \mathcal{S} \to \mathbb{R}_+$, satisfying $\sqrt{\mathbb{E}_{s \sim P}[L_{f,U}(s)^2]} \leq L_F(U)$ for some constant $L_F(U) \in \mathbb{R}_+$ depending on $U$, such that for $P$-almost all $s \in \mathcal{S}$,*

$$|f(x; s) - f(y; s)| \leq L_{f,U}(s)\|x - y\|_2$$

*holds for all $x, y \in U$.*

**Assumption 4.** *The objective function $\phi$ satisfies the quadratic growth condition on $\mathcal{X}^*$ globally, that is, there exists $c_1 > 0$ such that*

$$\phi(x) \geq \phi^* + c_1 \operatorname{dist}(x, \mathcal{X}^*)^2$$

*holds for all $x \in \mathbb{R}^d$.*

It is worth noting that Assumption 4 is not essential for establishing the stability or almost sure convergence of isPPA. Moreover, this assumption can be relaxed to its localized version (Assumption 5) when deriving the convergence rate of isPPA with square summable diminishing stepsizes, i.e., $\alpha_k = \alpha_0 k^{-\beta}$ for some $\beta \in (\frac{1}{2}, 1]$.

**Assumption 5.** *The objective function $\phi$ satisfies the quadratic growth condition on $\mathcal{X}^*$ locally, that is, there exists $c_1 > 0$ and $\delta > 0$ such that*

$$\phi(x) \geq \phi^* + c_1 \text{dist}(x, \mathcal{X}^*)^2 \quad \text{for all } x \in \mathcal{U}(\mathcal{X}^*, \delta),$$

*where $\mathcal{U}(\mathcal{X}^*, \delta)$ denotes the set $\{x \in \mathbb{R}^d \mid \text{dist}(x, \mathcal{X}^*) \leq \delta\}$.*

**Remark 1.1.** A sufficient condition for the optimal solution set $\mathcal{X}^*$ to be nonempty is the coercivity of the objective function $\phi$, which commonly holds for regularized regression models. The last condition in Assumption 2 is implied by the equality

$$\mathbb{E}_{s \sim P} [\partial \varphi(x; s)] = \partial \phi(x) \quad \text{for all } x \in \text{dom}(\phi), \tag{1.2}$$

where $\mathbb{E}_{s \sim P} [\partial \varphi(x; s)]$ is the expected subdifferential defined by

$$\mathbb{E}_{s \sim P} [\partial \varphi(x; s)] \triangleq \left\{ v_{\phi,x} \in \mathbb{R}^d \mid v_{\phi,x} = \int_{\mathcal{S}} \varphi'(x; s) \mathrm{d} P(s) \right.$$
$$\left. \text{with } \varphi'(x; \cdot) \text{ measurable and } \varphi'(x; s) \in \partial \varphi(x; s) \text{ for } P\text{-almost } s \in \mathcal{S} \right\}.$$

In the discrete case where $\mathcal{S} = [n]$, it follows from (Rockafellar, 1998, Theorem 23.8) that $\cap_{s \in \mathcal{S}} \text{ri}(\text{dom}(\varphi(\cdot; s))) \neq \emptyset$ is sufficient for (1.2), with $\text{ri}(\text{dom}(\varphi(\cdot; s)))$ relaxed to $\text{dom}(\varphi(\cdot; s))$ for polyhedral $\varphi(\cdot; s)$. For the continuous case, sufficient conditions are given in (Rockafellar & Wets, 1982; Bertsekas, 1973). Specifically, if $\varphi(\cdot; s)$ is real-valued and convex for each $s \in \mathcal{S}$, (1.2) holds due to (Bertsekas, 1973, Proposition 2.2).

---

**Algorithm 1** Inexact Stochastic Proximal Point Algorithm (isPPA)

---

**Parameters:** initial point $x_1$, stepsizes $\{\alpha_k\}_{k \geq 1}$, accuracy parameters $\{\epsilon_k\}_{k \geq 1}$, minibatch size $m$, maximum iteration count $K$

1: **for** $k = 1, 2, \cdots, K$ **do**

2:     Draw a random minibatch $S_k^{1:m}$ with $S_k^i \overset{\text{i.i.d.}}{\sim} P$

3:     Update

$$x_{k+1} \overset{\epsilon_k}{\approx} \arg\min_{x \in \mathbb{R}^d} \left\{ \overline{f}(x; S_k^{1:m}) + r(x) + \frac{1}{2\alpha_k} \|x - x_k\|_2^2 \right\} \tag{1.3}$$

     where $\overline{f}(x; S_k^{1:m})$ is defined in (1.1a)

4: **end for**

**Output:** last iterate $x_K$

---

## 2 ANALYSIS OF INEXACT STOCHASTIC PROXIMAL POINT ALGORITHM

In this section, we establish the convergence properties of isPPA, including the stability, almost sure convergence, and convergence rate guarantees in terms of the squared distance to the optimal solution set and the KKT residual.

Given $\epsilon_k \geq 0$, we consider the following three criteria and say that $x_{k+1}$ is obtained according to (1.3) if it satisfies one of these criteria:

$$\left\| x_{k+1} - \text{prox}_{\alpha_k \overline{\varphi}(\cdot; S_k^{1:m})}(x_k) \right\|_2 \leq \epsilon_k, \tag{SCA}$$

$$\Phi_{\alpha_k, x_k}(x_{k+1}; S_k^{1:m}) - \Phi^*_{\alpha_k, x_k, S_k^{1:m}} \leq \frac{\epsilon_k^2}{2\alpha_k}, \tag{SCB}$$

$$\text{dist}\left(0, \partial \Phi_{\alpha_k, x_k}(x_{k+1}; S_k^{1:m})\right) \leq \frac{\epsilon_k}{\alpha_k}, \tag{SCC}$$

where function $\Phi_{\alpha_k, x_k}(\cdot; S_k^{1:m}) \colon \mathbb{R}^d \to (-\infty, +\infty]$ is defined by

$$\Phi_{\alpha_k, x_k}\left(x; S_k^{1:m}\right) \triangleq \overline{\varphi}\left(x; S_k^{1:m}\right) + \frac{1}{2\alpha_k} \|x - x_k\|_2^2 \quad \text{for all } x \in \mathbb{R}^d,$$

and $\Phi^*_{\alpha_k,x_k,S^{1:m}_k} \triangleq \min_{x \in \mathbb{R}^d} \Phi_{\alpha_k,x_k}(x; S^{1:m}_k)$ denotes the minimum value. Leveraging the fact that $\Phi_{\alpha_k,x_k}(\cdot; S^{1:m}_k)$ is $\frac{1}{\alpha_k}$-strongly convex under Assumption 1, it is straightforward to verify that

$$\text{Criterion (SCC)} \quad \Rightarrow \quad \text{Criterion (SCB)} \quad \Rightarrow \quad \text{Criterion (SCA).} \tag{2.2}$$

Therefore, it is sufficient to establish the convergence results for isPPA with $\{x_k\}$ satisfying criterion (SCA). Due to (2.2), the derived conclusions also hold for isPPA with $\{x_k\}$ satisfying either criterion (SCB) or criterion (SCC).

Unless otherwise stated, we will use the following notations for simplicity. For any $\beta \in \mathbb{R}$, the function $\varsigma_\beta \colon \mathbb{R}_{++} \to \mathbb{R}$ is defined by

$$\varsigma_\beta(t) \triangleq \begin{cases} \frac{t^\beta - 1}{\beta} & \text{if } \beta \neq 0, \\ \ln(t) & \text{if } \beta = 0, \end{cases} \tag{2.3}$$

for all $t \in \mathbb{R}_{++}$. For any bounded open subset $U \subset \mathrm{dom}(r)$, we assume without loss of generality that $\mathbb{E}_{s \sim P}[L_{f,U}(s)^2] > 0$ and set

$$\eta_{f,U} \triangleq 1 - \frac{\mathrm{Var}\,(L_{f,U}(s))}{\mathbb{E}_{s \sim P}\,[L_{f,U}(s)^2]} \quad \text{and} \quad \rho_{f,m,U} \triangleq \left(\sqrt{\frac{1 + (m-1)\eta_{f,U}}{m}} + 1\right)^2. \tag{2.4}$$

Fix any $k \in \mathbb{Z}_+$. Let $\mathcal{F}_k \triangleq \sigma(S^{1:m}_1, \cdots, S^{1:m}_k)$ be the $\sigma$-field generated by the first $k$ random minibatch $\{S^{1:m}_i\}^k_{i=1}$, and denote the conditional expectation $\mathbb{E}_k[\cdot] \triangleq \mathbb{E}[\cdot \mid \mathcal{F}_{k-1}]$. Additionally, we denote the proximal point $\mathrm{prox}_{\alpha_k \overline{\varphi}(\cdot; S^{1:m}_k)}(x_k)$ by $\tilde{x}_{k+1}$.

## 2.1 STABILITY AND ALMOST SURE CONVERGENCE

Before establishing convergence rate guarantees for isPPA, we first prove conditions sufficient to ensure boundedness of the iterates $\{x_k\}$ generated by (1.3). By adopting the proof of (Asi & Duchi, 2019, Theorem 3.2 and Corollary 3.5) with suitable modifications, we have the following theorem. The proof is omitted here and deferred to Appendix B.1.

**Theorem 2.1.** *Let Assumption 1 and 2 hold, and let $\{x_k\}$ be generated by isPPA (Algorithm 1) with stepsizes $\{\alpha_k\}$ and parameters $\{\epsilon_k\}$. Suppose that the parameters $\{\epsilon_k\}$ satisfy $\epsilon_k = \gamma \alpha^2_k$ for each $k \in \mathbb{Z}_+$ with $\gamma \in \mathbb{R}_+$. Then the following assertions hold:*

*(i) For all $\overline{x} \in \mathcal{X}^*$ and $k \in \mathbb{Z}_+$, we have*

$$\mathbb{E}_k\left[\|x_{k+1} - \overline{x}\|^2_2\right] \leq \left(1 + \alpha^2_k\right)\|x_k - \overline{x}\|^2_2 + \left(1 + \alpha^2_k\right)\alpha^2_k\left(\frac{\sigma^2_\phi}{m} + \gamma^2\right). \tag{2.5}$$

*(ii) Assume further that the stepsizes $\{\alpha_k\}$ satisfy $\sum^\infty_{k=1} \alpha^2_k < \infty$. Then*

$$\sup_{k \in \mathbb{Z}_+} \|x_k\|_2 < \infty \quad \text{with probability } 1. \tag{2.6}$$

**Remark 2.1.** For the finite-sum regularized regression model with real-valued nonnegative convex component functions and a real-valued convex coercive regularizer, Remark 1.1, combined with the compactness of the subdifferential of a real-valued convex function, implies that Assumption 1 and 2 hold. By Theorem 2.1, this observation shows that isPPA exhibits stability under mild conditions.

With Theorem 2.1 established, we are now positioned to demonstrate almost sure convergence of isPPA with diminishing stepsizes, assuming local Lipschitz continuity of component functions. Our proof sketch incorporates and modifies the methodologies from Davis & Drusvyatskiy (2019, Lemma 4.2) and Bertsekas (2011, Proposition 9) to initially derive convergence results for exact sPPA (i.e., $\epsilon_k \equiv 0$), and subsequently extends these findings to inexact settings using the Cauchy-Schwarz inequality. See Appendix B.2 for a detailed proof. It is worth noting that our proof technique diverges from that employed by Asi & Duchi (2019, Proposition 3.8); see Remark B.1 in Appendix B.2 for more details.

**Theorem 2.2.** *Let Assumption 1-3 hold, and let $\{x_k\}$ be generated by isPPA (Algorithm 1) with stepsizes $\{\alpha_k\}$ and parameters $\{\epsilon_k\}$. Suppose that the stepsizes $\{\alpha_k\}$ satisfy $\sum^\infty_{k=1} \alpha_k = \infty$ and $\sum^\infty_{k=1} \alpha^2_k < \infty$, and the parameters $\{\epsilon_k\}$ satisfy $\epsilon_k = \gamma \alpha^2_k$ for each $k \in \mathbb{Z}_+$ with $\gamma \in \mathbb{R}_+$. Then with probability 1, both $\liminf_{k \to \infty} \phi(\tilde{x}_k) = \phi^*$ and there exists $x^* \in \mathcal{X}^*$ such that $\lim_{k \to \infty} \|x_k - x^*\|_2 = 0$ hold.*

## 2.2 CONVERGENCE RATE IN THE SQUARED DISTANCE TO OPTIMAL SOLUTION SET

Recall the function $\varsigma_\beta$ defined in (2.3) and the notation $\rho_{f,m,U}$ introduced in (2.4). In this part, we consider the bounds on the last-iterate convergence rate of isPPA, measured by the expected squared distance to the optimal solution set, under a local Lipschitz conditon on $f(\cdot; s)$ and a quadratic growth conditon on $\phi$.

Unless stated otherwise, if there is a bounded open subset of $\mathrm{dom}(r)$ that contains the sequences $\{x_k\}$ and $\{\tilde{x}_k\}$, we denote this subset by $U$. Let $s \in (0, 2)$ be an arbitrary scalar and define

$$\tilde{C}_{f,m,U,c_1,\gamma}(\alpha) \triangleq \rho_{f,m,U} L_F(U)^2 + \frac{(1 + 2c_1\alpha)(1 + sc_1\alpha)}{(2 - s)c_1}\gamma^2 \tag{2.7}$$

for all $\alpha \in \mathbb{R}_+$. By utilizing Lemma B.3 (see Appendix B) and following the proof arguments of (Moulines & Bach, 2011, Theorem 1), we can thus derive nonasymptotic convergence rates for isPPA as summarized in Theorem 2.3 and 2.4 below.

**Theorem 2.3.** *Let Assumption 1-4 hold, and let $\{x_k\}$ be generated by isPPA (Algorithm 1) with constant stepsize $\alpha_k = \alpha_0$ and parameter $\epsilon_k = \gamma\alpha_0^{\frac{3}{2}}$ for all $k \in \mathbb{Z}_+$, where $\alpha_0 \in \mathbb{R}_{++}$ and $\gamma \in \mathbb{R}_+$. Denote by $\delta_1 \triangleq \mathrm{dist}(x_1, \mathcal{X}^*)^2$ and let $s \in (0, 2)$ be an arbitrary scalar. Then, on the event that $\sup_{k \in \mathbb{Z}_+} \|x_k\|_2 < \infty$, we have*

$$\mathbb{E}\left[\mathrm{dist}(x_k, \mathcal{X}^*)^2\right] \leq \left(\frac{1}{1 + sc_1\alpha_0}\right)^{k-1}\delta_1 + \frac{\tilde{C}_{f,m,U,c_1,\gamma}(\alpha_0)}{sc_1}\alpha_0 \tag{2.8}$$

*for all $k \in \mathbb{Z}_+$, where $\tilde{C}_{f,m,U,c_1,\gamma}(\cdot)$ is defined in (2.7).*

**Theorem 2.4.** *Let Assumption 1-4 hold, and let $\{x_k\}$ be generated by isPPA (Algorithm 1) with diminishing stepsize $\alpha_k = \alpha_0 k^{-\beta}$ and parameter $\epsilon_k = \gamma\alpha_k^{\frac{3}{2}}$ for all $k \in \mathbb{Z}_+$, where $\alpha_0 \in \mathbb{R}_{++}$, $\beta \in (0, 1]$ and $\gamma \in \mathbb{R}_+$. Denote by $\delta_1 \triangleq \mathrm{dist}(x_1, \mathcal{X}^*)^2$ and let $s \in (0, 2)$ be an arbitrary scalar. Then, on the event that $\sup_{k \in \mathbb{Z}_+} \|x_k\|_2 < \infty$, the following assertions hold:*

*(i) If $\beta \in (0, 1)$, then*

$$\begin{aligned}
\mathbb{E}\left[\mathrm{dist}(x_k, \mathcal{X}^*)^2\right] &\leq \exp\left(-\frac{sc_1\alpha_0}{1 + sc_1\alpha_0}\varsigma_{1-\beta}(k)\right)\delta_1 \\
&\quad + \tilde{C}_{f,m,U,c_1,\gamma}(\alpha_0) \cdot 4^\beta \frac{\alpha_0^2}{1 - \exp(-\frac{sc_1\alpha_0}{1 + sc_1\alpha_0})} \cdot \frac{1}{k^\beta} \\
&\quad + \tilde{C}_{f,m,U,c_1,\gamma}(\alpha_0) \cdot 4^\beta \alpha_0^2 \cdot \exp\left(-\frac{sc_1\alpha_0}{2(1 + sc_1\alpha_0)}k^{1-\beta}\right) \cdot \varsigma_{1-2\beta}\left(\frac{k}{2}\right)
\end{aligned} \tag{2.9}$$

*holds for all $k \in \mathbb{Z}_+$, where $\tilde{C}_{f,m,U,c_1,\gamma}(\cdot)$ is defined in (2.7).*

*(ii) If $\beta = 1$, then*

$$\mathbb{E}\left[\mathrm{dist}(x_k, \mathcal{X}^*)^2\right]$$
$$\leq \begin{cases} \left[\frac{4\delta_1}{1+sc_1\alpha_0} + \frac{4\tilde{C}_{f,m,U,c_1,\gamma}(\alpha_0)(2+sc_1\alpha_0)\alpha_0^2}{2-sc_1\alpha_0}\right] \cdot \left(\frac{1}{k}\right)^{\frac{2sc_1\alpha_0}{2+sc_1\alpha_0}} & \text{if } c_1\alpha_0 < \frac{2}{s}, \\ \frac{4\delta_1}{3} \cdot \frac{1}{k} + 4\tilde{C}_{f,m,U,c_1,\gamma}(\alpha_0)\alpha_0^2 \cdot \frac{\ln(k)}{k} & \text{if } c_1\alpha_0 = \frac{2}{s}, \\ \frac{4\delta_1}{1+sc_1\alpha_0} \cdot \left(\frac{1}{k}\right)^{\frac{2sc_1\alpha_0}{2+sc_1\alpha_0}} + \frac{4\tilde{C}_{f,m,U,c_1,\gamma}(\alpha_0)(2+sc_1\alpha_0)\alpha_0^2}{sc_1\alpha_0-2} \cdot \frac{1}{k} & \text{if } c_1\alpha_0 > \frac{2}{s}, \end{cases} \tag{2.10}$$

*holds for all $k \in \mathbb{Z}_+$, where $\tilde{C}_{f,m,U,c_1,\gamma}(\cdot)$ is defined in (2.7).*

Focusing on dominant terms, we can immediately derive the asymptotic convergence rate for isPPA as shown in Corollary 2.5, which provides a clearer estimate of the convergence rate obtained in Theorem 2.4.

**Corollary 2.5.** *Consider the setting of Theorem 2.4 and let $s \in (0, 2)$ be an arbitrary scalar. On the event that $\sup_{k \in \mathbb{Z}_+} \|x_k\|_2 < \infty$, the following assertions hold:*

*(i) If $\beta \in (0, 1)$, then $\mathbb{E}\left[\mathrm{dist}(x_k, \mathcal{X}^*)^2\right] \leq \mathcal{O}\left(k^{-\beta}\right)$ holds for all $k \in \mathbb{Z}_+$.*

*(ii) If $\beta = 1$, then*

$$\mathbb{E}\left[\operatorname{dist}(x_k, \mathcal{X}^*)^2\right] \leq \begin{cases} \mathcal{O}\left(k^{-\frac{2sc_1\alpha_0}{2+sc_1\alpha_0}}\right) & \text{if } c_1\alpha_0 < \frac{2}{s}, \\ \mathcal{O}\left(k^{-1}\ln(k)\right) & \text{if } c_1\alpha_0 = \frac{2}{s}, \\ \mathcal{O}\left(k^{-1}\right) & \text{if } c_1\alpha_0 > \frac{2}{s}, \end{cases}$$

*holds for all $k \in \mathbb{Z}_+$.*

The proofs of Theorem 2.3, 2.4, and Corollary 2.5 can be found in Appendix C.

**Diminishing stepsizes and localized quadratic growth condition.** In the context of isPPA with diminishing stepsizes $\alpha_k = \alpha_0 k^{-\beta}$ and parameters $\epsilon_k = \gamma\alpha_k^2$, where $\alpha_0 \in \mathbb{R}_{++}$, $\beta \in (\frac{1}{2}, 1]$ and $\gamma \in \mathbb{R}_+$, and considering the square summability of $\{\alpha_k\}$, Theorem 2.1 ensures the almost sure boundedness of $\{x_k\}$ under Assumption 1 and 2. In this setting, since $\alpha_k \leq \alpha_0$ and $\gamma\alpha_k^2 = \gamma\alpha_k^{1/2} \cdot \alpha_k^{3/2} \leq \gamma\alpha_0^{1/2} \cdot \alpha_k^{3/2}$, the iterates $\{x_k\}$ also satisfy Criterion (SCA) with $\tilde{\epsilon}_k \triangleq \tilde{\gamma}\alpha_k^{3/2}$, where $\tilde{\gamma} \triangleq \gamma\alpha_0^{1/2}$. Consequently, all the results in Theorem 2.4 and Corollary 2.5 hold with probability 1, even without assuming $\sup_{k\in\mathbb{Z}_+} \|x_k\|_2 < \infty$. Additionally, based on the fact that $\{\operatorname{dist}(x_k, \mathcal{X}^*)\}$ converges to zero almost surely as implied by Theorem 2.2, Assumption 4 in Theorem 2.4 can be replaced with its localized version (Assumption 5). Under this localized quadratic growth condition, convergence results analogous to those in Theorem 2.4 and Corollary 2.5 can be achieved for sufficiently large $k$.

**Relationship with prior work.** When each component function is Lipschitz smooth and restricted strongly convex, and the regularizer $r$ is zero, the asymptotic convergence rate descirbed in Theorem 4 of (Ryu & Boyd, 2014) for standard sPPA (i.e., exact sPPA with a minibatch size of one) with stepsizes $\alpha_k = \alpha_0 k^{-1}$ can be deduced from Corollary 2.5. Similarly, the last-iterate convergence rate guarantees provided in Theorem 1 of (Toulis et al., 2016) for standard sPPA with stepsizes $\alpha_k = \alpha_0 k^{-\beta}$ ($\beta \in (\frac{1}{2}, 1]$) can be derived from Theorem 2.4. However, these guarantees established by Toulis et al. (2016) require more restrictive assumptions, such as the twice continuous differentiability and a specific structural condition on component functions. The last-iterate convergence guarantees in Theorem 4.3 of (Patrascu, 2021) apply exclusively to standard sPPA, and achieving the linear rate of (Patrascu, 2021, Corollary 4.5) requires the problem to be easy to optimize. The $\mathcal{O}(k^{-1})$ ergodic convergence rates in Theorem 10 of (Yuan & Li, 2023) for inexact sPPA with stepsizes $\alpha_k = \alpha_0 k^{-1}$ are established under stringent conditions including Lipschitz smoothness of component functions, Lipschitz continuity of the regularizer across the entire domain, and a sufficiently large minibatch size. Most previous work have focused on the convergence properties of standard sPPA when solving problem (ComOpt) with Lipschitz smooth component functions. As noted in Remark 2.1, the assumptions applied in this paper are reasonable and feasible in practice, and the above comparisons indicate that our derived convergence rate estimates are competitive.

## 2.3 Convergence Rate in the KKT Residual

The bounds on the rate of convergence in terms of the expected squared distance to the optimal solution set given in Section 2.2 possess significant theoretical importance. However, termination criteria in practice are frequently associated with the KKT residual. Consequently, it is essential to establish convergence rate guarantees with respect to the KKT residual. Fortunately, the following relationship between the KKT residual and squared distance to the optimal solution set holds.

**Lemma 2.6.** *Let $\phi\colon \mathbb{R}^d \to (-\infty, +\infty]$ be a proper and closed convex function, and denote the set of minimizers of $\phi$ by $\mathcal{X}^*$. If $\mathcal{X}^*$ is nonempty, then for any $x \in \mathbb{R}^d$, we have*

$$\left\|x - \operatorname{prox}_\phi(x)\right\|_2 \leq 2\operatorname{dist}(x, \mathcal{X}^*),$$

The established relationship enables us to extend all conclusions from Section 2.2 to the corresponding convergence rate bounds in terms of the expected KKT residual. For the sake of brevity, we focus exclusively on the asymptotic convergence rates of isPPA with diminishing stepsizes in this part.

**Corollary 2.7.** *Consider the setting of Theorem 2.4 and let $s \in (0, 2)$ be an arbitrary scalar. On the event that $\sup_{k\in\mathbb{Z}_+} \|x_k\|_2 < \infty$, the following assertions hold:*

(i) If $\beta \in (0,1)$, then $\mathbb{E}\left[\left\|x_k - \mathrm{prox}_\phi(x_k)\right\|_2\right] \leq \mathcal{O}\left(k^{-\frac{\beta}{2}}\right)$ holds for all $k \in \mathbb{Z}_+$.

(ii) If $\beta = 1$, then

$$\mathbb{E}\left[\left\|x_k - \mathrm{prox}_\phi(x_k)\right\|_2\right] \leq \begin{cases} \mathcal{O}\left(k^{-\frac{sc_1\alpha_0}{2+sc_1\alpha_0}}\right) & \text{if } c_1\alpha_0 < \frac{2}{s}, \\ \mathcal{O}\left(k^{-\frac{1}{2}}\left(\ln(k)\right)^{\frac{1}{2}}\right) & \text{if } c_1\alpha_0 = \frac{2}{s}, \\ \mathcal{O}\left(k^{-\frac{1}{2}}\right) & \text{if } c_1\alpha_0 > \frac{2}{s}, \end{cases}$$

holds for all $k \in \mathbb{Z}_+$.

The proofs of Lemma 2.6 and Corollary 2.7 are deferred to Appendix D.

## 3 LOWER BOUNDS OF CONVERGENCE RATE

In this section, we establish a lower bound for the convergence rate of isPPA. The main results are summarized in the following proposition; see Appendix E for a detailed proof.

**Proposition 3.1.** *Let $\{p_i\}_{i=1}^n \subset \mathbb{R}^d$ be given points and $\lambda \in \mathbb{R}_+$. Consider the finite-sum convex optimization problem:*

$$\min_{x \in \mathbb{R}^d} \ \phi(x) \triangleq F(x) + r(x) = \frac{1}{n}\sum_{i=1}^n \|x - p_i\|_2^2 + \frac{\lambda}{2}\|x\|_2^2. \tag{3.1}$$

*Denote the set $[n] \triangleq \{1,\cdots,n\}$. Let $\{x_k\}$ be generated by (1.3) with stepsize $\alpha_k \in \mathbb{R}_{++}$ and parameter $\epsilon_k \equiv 0$ for all $k \in \mathbb{Z}_+$. Then the following assertions hold:*

(i) *Assumption 1-4 hold for (3.1) and $x^* \triangleq \frac{2}{(2+\lambda)n}\sum_{i=1}^n p_i$ is the unique optimal solution.*

(ii) *Set $\Sigma \triangleq \max_{i\in[n]}\|p_i\|_2$. Then $\sup_{k\in\mathbb{Z}_+}\|x_k\|_2 \leq \|x_1\|_2 + \Sigma < \infty$, i.e., $\{x_k\}$ is bounded.*

(iii) *Assume further that there exists $i_0 \neq j_0 \in [n]$ with $p_{i_0} \neq p_{j_0}$. Then $\sigma^2 \triangleq \frac{1}{n}\sum_{j=1}^n\|p_j - \frac{1}{n}\sum_{i=1}^n p_i\|_2^2 > 0$ and for any $k \in \mathbb{Z}_+$,*

$$\mathbb{E}\left[\|x_{k+1} - x^*\|_2^2\right]$$
$$\geq \begin{cases} \frac{4\sigma^2\alpha_0^2}{m[(2+\lambda)\alpha_0+1]^2} & \text{if } \alpha_k \geq \alpha_0 \text{ for some } \alpha_0 \in \mathbb{R}_{++}, \\ \frac{4\sigma^2\alpha_0^2}{m[(2+\lambda)\alpha_0+1]^4} \cdot k^{-\beta} & \text{if } \alpha_k = \frac{\alpha_0}{k^\beta} \text{ for some } \alpha_0 \in \mathbb{R}_{++} \text{ and } \beta \in (0,1]. \end{cases}$$

**Tightness of derived bounds.** Combined with the upper bounds derived in Section 2.2, the lower bound established in Proposition 3.1 shows that for isPPA with last iterate output, solving problems (ComOpt) satisfying Assumption 1-4, the following holds if the convergence rate is measured by the expected squared distance and no other structure is known: a) when the stepsizes $\{\alpha_k\}$ have a uniform positive lower bound, the algorithm guarantees only convergence to a neighborhood of the optimal solution (see Figure 1(b)), and thus the bound on the convergence rate of isPPA with constant stepsizes from Theorem 2.3 cannot be improved; b) for the diminishing stepsizes $\{\alpha_k\}$ with $\alpha_k = \alpha_0 k^{-\beta}$ ($\beta \in (0,1]$), the asymptotic convergence rate of $\mathcal{O}(k^{-\beta})$ from Corollary 2.5 is tight up to constant factors (see Figure 1(c)).

**Comparison with deterministic PPA.** It is known that if the sequence of stepsizes $\{\alpha_k\}$ satisfies $\alpha_k \to \alpha_\infty$ for some $\alpha_\infty \in \mathbb{R}_{++}$, then inexact deterministic PPA exhibits linear convergence under quadratic growth condition and can achieve superlinear convergence as $\alpha_k \to \infty$ (Rockafellar, 1976a; Luque, 1984). In contrast, our analysis shows that even when solving Fréchet mean problem (3.1) with Lipschitz smooth component functions and a strongly convex objective function, isPPA with lower bounded stepsizes merely approaches a neighborhood of the optimal solution but fails to achieve exact convergence to the optimal solution. This discrepancy may stem from the noise introduced during subproblem minimization through the term $\overline{f}(\cdot; S_k^{1:m})$ (see (1.3)). As shown in

Appendix C, this noise contributes to the term $\rho_{f,m,U}L_F(U)^2$ in the nonasymptotic estimate of Theorem 2.3, while the first term in this estimate geometrically converges to zero. The distinctions between the deterministic PPA and stochastic variants are empirically validated by the numerical results presented in Figure 1(a) and 1(b).

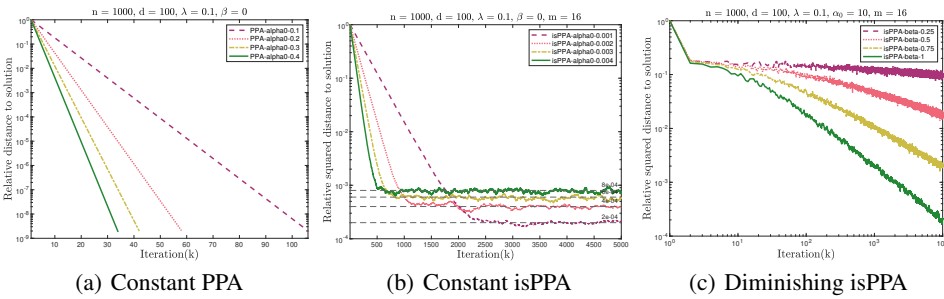

(a) Constant PPA  (b) Constant isPPA  (c) Diminishing isPPA

Figure 1: Performance comparison of PPA and isPPA for solving the Fréchet mean problem (3.1). Legends "Algo-alpha0-$\alpha_0$" and "Algo-beta-$\beta$" denote methods with constant stepsizes $\alpha_k = \alpha_0$ and diminishing stepsizes $\alpha_k = \alpha_0 k^{-\beta}$ ($\beta \in (0, 1]$), respectively, where "Algo" refers to either PPA or isPPA.

## 4 Numerical Experiments

In this section, we present numerical results to validate the theoretical convergence rate guarantees derived in Section 2. For brevity, the discussion here is limited to the performance of isPPA on the synthetic data, with details on real data sets available in Appendix F. We generate the synthetic data using the model $b = Ax^* + \sigma\xi$, where $A$ is sampled from standard normal distribution, $x^*$ is the predefined true solution, $\xi$ is a noise vector from a standard normal distribution, and $\sigma$ denotes the noise level. The number of nonzero elements in $x^*$ is set to $\lfloor \rho_s d \rfloor$ with $\rho_s = 1\%$. The results are averaged over 5 trials with consistent parameters and initialization across each trial of isPPA.

### 4.1 Lasso Linear Regression Model

Consider the linear regression model with a Lasso regularizer (Tibshirani, 1996):

$$\min_{x \in \mathbb{R}^d} \psi_{\text{Lasso}}(x) \triangleq \frac{1}{2}\|Ax - b\|_2^2 + \lambda_1\|x\|_1, \tag{4.1}$$

where matrix $A \in \mathbb{R}^{n \times d}$ and vector $b \in \mathbb{R}^n$ are given data, with $\lambda_1 \in \mathbb{R}_{++}$ as the regularization parameter. For each trial of isPPA with diminishing stepsizes $\alpha_k = \alpha_0 k^{-\beta}$ where $\beta \in (\frac{1}{2}, 1]$, given that the optimal solution may not be unique, we first obtain an approximation, denoted by $\hat{x}^*$, of the optimal solution, to which the sequence $\{x_k\}$ converges almost surely as indicated in Theorem 2.2. See Appendix F.1 for details on obtaining $\hat{x}^*$.

For the Lasso linear regression model (4.1), we utilize the efficient semismooth Newton (SSN) method (Li et al., 2018, Algorithm SSN) to solve the inner-loop subproblem (1.3) inexactly, with the sequence $\{x_k\}$ satisfying criterion (SCA); see Appendix F.1 for further details. In our numerical experiments with synthetic data, we test with $n = 10000$ and $d = 1000$, setting $\sigma$ to zero in noiseless experiments and to $10^{-2}$ otherwise. The regularization parameter $\lambda_1$ is set to $\lambda_c\|A^\top b\|_\infty$ with $\lambda_c = 10^{-2}$. The accuracy parameter $\epsilon_k$ is defined as $\gamma\alpha_k^2$ with $\gamma = 10^{-2}$, and the minibatch size $m$ is fixed at 32. The top row of Figure 2 displays the convergence curves of isPPA with diminishing stepsizes $\alpha_k = \alpha_0 k^{-\beta}$ for various stepsize exponents $\beta \in \{0.55, 0.75, 0.9, 1\}$, starting with an initial stepsize $\alpha_0 = 50$. It is observed that isPPA with a stepsize exponent of $\beta = 1$ exhibits the best performance. The asymptotic convergence rate measured by the squared distance to the solution set is $\mathcal{O}(k^{-\beta})$, considering $\|x_k - \hat{x}^*\|_2^2$ as an upper bound of $\text{dist}(x_k, \mathcal{X}^*)^2$. It also shows that the asymptotic convergence rate regarding the KKT residual (defined in (F.5)) is $\mathcal{O}(k^{-\beta/2})$. These observations confirm the results derived in Corollary 2.5 and 2.7.

## 4.2 ELASTIC NET LINEAR REGRESSION MODEL

The linear regression model with an elastic net regularizer (Zou & Hastie, 2005) is defined by

$$\min_{x\in\mathbb{R}^d}\ \psi_{\mathrm{elastic}}(x) \triangleq \frac{1}{2}\left\|Ax-b\right\|_2^2 + \lambda_1\left\|x\right\|_1 + \frac{\lambda_2}{2}\left\|x\right\|_2^2, \tag{4.2}$$

where $\lambda_1, \lambda_2 \in \mathbb{R}_{++}$ are the given regularization parameters. It is straightforward to verify that the objective function $\psi_{\mathrm{elastic}}$ is $\lambda_2$-strongly convex over $\mathbb{R}^d$, ensuring the uniqueness of the optimal solution to (4.2). The setup for the numerical experiments of isPPA for solving (4.2) is similar to that previously described for (4.1); see Appendix F.2 for further details. Notably, given the uniqueness of the solution to (4.2), we directly employ the semismooth Newton augmented Lagrangian (SS-NAL) method (Li et al., 2018, Algorithm SSNAL) to approximate the optimal solution, to which the sequence $\{x_k\}$ converges almost surely. This approximate optimal solution is denoted by $\tilde{x}^*$. The numerical results, illustrated in the bottom row of Figure 2, demonstrate asymptotic convergence rates of $\mathcal{O}(k^{-\beta})$ for the squared distance $\left\|x_k - \tilde{x}^*\right\|_2^2$ and $\mathcal{O}(k^{-\beta/2})$ for the KKT residual (defined in (F.8)), thereby validating the findings from Corollary 2.5 and 2.7.

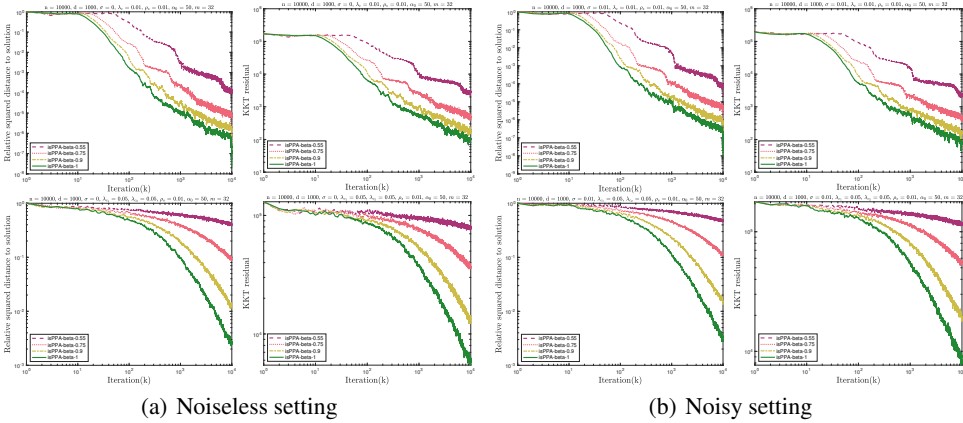

(a) Noiseless setting      (b) Noisy setting

Figure 2: Performance of isPPA in solving the linear regression models (Lasso - top and elastic net - bottom) for four values of stepsize exponents $\beta = 0.55, 0.75, 0.9$ and $1$. The legend "isPPA-beta-$\beta$" denotes isPPA with diminishing stepsizes $\alpha_k = \alpha_0 k^{-\beta}$. (a) The noiseless setting with $\sigma = 0$. (b) The noisy setting with $\sigma = 10^{-2}$.

## 5 CONCLUSION

In this paper, we consider inexact stochastic proximal point algorithm (isPPA) for solving stochastic composite optimization problems, allowing subproblems to be solved inexactly via suitable inner-loop stopping criteria. Under mild conditions, we demonstrate the stability and almost sure convergence of this method. By further assuming a local Lipschitz condition on component functions and a quadratic growth condition on the objective function, we establish convergence rate guarantees. These rates are measured by the expected squared distance to the optimal solution set and the expected KKT residual. While the former provides theoretical insights, the latter offers practical estimates for algorithm termination. Our findings confirm that these convergence rate guarantees are asymptotically tight up to constant factors. Empirical validation from numerical experiments supports our theoretical analyses. In future work, we will focus on enhancing its efficiency with acceleration strategies and variance reduction techniques, and establishing the corresponding convergence analysis.

### REPRODUCIBILITY STATEMENT

Upon acceptance of the paper, we will release the code used for the demonstration examples in the numerical experiments.

ACKNOWLEDGMENTS

Yancheng Yuan is supported by the National Key R&D Program of China under grant 2023YFA1009300, and the Research Center for Intelligent Operations Research. Shulan Zhu and Chenglong Bao are supported by the National Natural Science Foundation of China (No. 12271291). Defeng Sun is supported by the Research Center for Intelligent Operations Research, RGC Senior Research Fellow Scheme No. SRFS2223-5S02, and GRF Project No. 15304721. Shulan Zhu would like to thank Prof. Sun for hosting her research visit at the Hong Kong Polytechnic University, during which the foundational work of this study was initiated.

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

## A    COMPARISON OF CONVERGENCE RESULTS FOR sPPA SCHEMES

Table 1 summarizes and compares convergence rate guarantees and associated assumptions for sPPA-type schemes in solving problem (ComOpt). We use the following abbreviations:

- bd (bounded), cld (closed)
- lin.reg (linear regularity)
- cvx (convex), s.cvx (strongly convex), r.s.cvx (restricted strongly convex), wk.cvx (weakly convex);
- Lip (Lipschitz continuous), loc.Lip (locally Lipschitz continuous), Lip.sm (Lipschitz continuous gradient), $\mathcal{C}^2$ (twice continuously differentiable), sm (differentiable)
- wk.lin.reg (weak linear regularity), qd.grow (quadratic growth), e.qd.grow (expected quadratic growth), e.sharp.grow (expected sharp growth)
- easy.opt (easy to optimize)
- l.e (sufficiently large)
- sq.dist (convergence rate in terms of $\{\mathbb{E}[\|x_k - x^*\|_2^2]\}$), sq.dist.set (convergence rate in terms of $\{\mathbb{E}[\text{dist}(x_k, \mathcal{X}^*)^2]\}$), func.gap (convergence rate in terms of $\{\phi(x_k) - \phi^*\}$), sq.norm.grad.ME (convergence rate in terms of the squared gradient $\ell_2$-norm of Moreau envelope with respect to $\phi$)

Table 1: Comparison of convergence rates for sPPA-type methods solving stochastic composite optimization problem (ComOpt).

| Method | Literature | Model | Assumptions | Stepsize | Rate |
|--------|-----------|-------|-------------|----------|------|
| sPPA | Ryu & Boyd (2014) | $r = 0$ | $f(\cdot; s)$ r.s.cvx & Lip.sm | $\frac{\alpha_0}{k}$ | sq.dist $\mathcal{O}(\frac{1}{k})$ $\{\alpha_0$ l.e.$\}$ |
| | | | $f(\cdot; s)$ r.s.cvx | $\alpha_0$ | linear convergence to $\mathcal{O}(\alpha_0)$ (dist) |
| | Toulis et al. (2016) | $r = 0$ | $f(x; s) = f(a_s^\top x)$; $f(\cdot; s)$ cvx & Lip & $\mathcal{C}^2$ | $\frac{\alpha_0}{k^\beta}$ | sq.dist $\mathcal{O}(\frac{1}{k^\beta})$ $\{\beta \in (\frac{1}{2}, 1)\}$ or $\{\beta = 1$ & $\alpha_0$ l.e.$\}$ |
| | Patrascu (2021) | $r = 0$ | $\partial f(\cdot; s)$ bd on $\mathcal{X}^*$; $F$ wk.lin.reg | $\frac{\alpha_0}{k^\beta}$ | sq.dist.set $\mathcal{O}(\frac{1}{k^\beta})$ $\{\beta \in (0, 1)\}$ or $\{\beta = 1$ & $\alpha_0$ l.e.$\}$ |
| | | | $\partial f(\cdot; s)$ bd on $\mathcal{X}^*$; $F$ wk.lin.reg ($\eta = 0$); easy.opt | $\alpha_0$ | linear convergence (sq.dist.set) |
| New-sPPA | Patrascu & Necoara (2018) | $r = \mathbf{1}_{\mathcal{X}}$ | $f(\cdot; s)$ cvx & Lip; $\mathcal{X} = \cap_{s \in \mathcal{S}} X_s$; $X_s$ cld.cvx & $\mathcal{X}$ lin.reg; | $\alpha_0 = \mathcal{O}(\frac{1}{\sqrt{k}})$ | func.gap $\mathcal{O}(\frac{1}{\sqrt{k}})$ (ergodic) |
| | | | $f(\cdot; s)$ s.cvx & sm; $\mathcal{X} = \cap_{s \in \mathcal{S}} X_s$; $X_s$ cld.cvx & $\mathcal{X}$ lin.reg; | $\alpha_0$ | linear convergence to $\mathcal{O}(\alpha_0)$ (sq.dist) |
| | | | $f(\cdot; s)$ s.cvx & Lip.sm; $\mathcal{X} = \cap_{s \in \mathcal{S}} X_s$; $X_s$ cld.cvx & $\mathcal{X}$ lin.reg; | $\frac{\alpha_0}{k^\beta}$ | sq.dist $\mathcal{O}(\frac{1}{k^\beta})$ $\{\beta \in (0, 1)\}$ or $\{\beta = 1$ & $\alpha_0$ l.e.$\}$ |
| APROX | Asi & Duchi (2019) | $r = \mathbf{1}_{\mathcal{X}}$ | $f(\cdot; s)$ cvx & e.sharp.grow; easy.opt | $\frac{\alpha_0}{k^\beta}$ | linear convergence (sq.dist.set) $\{\beta \in (-\infty, 1)\}$ |
| | | | $f(\cdot; s)$ cvx & e.qd.grow; easy.opt | $\frac{\alpha_0}{k^\beta}$ | linear convergence (sq.dist.set) $\{\beta \in \mathbb{R}\}$ |

| | | | | | *continued from previous page* |
|---|---|---|---|---|---|
| Method | Literature | Model | Assumptions | Stepsize | Rate |
| Model-based | Davis & Drusvyatskiy (2019) | - | $F$ loc.Lip; $r$ cld; $f_x(\cdot;s) + r(\cdot)$ wk.cvx; $f_x(\cdot;s)$ Lip; | $\alpha_0$ | sq.norm.grad.ME $\mathcal{O}(\frac{1}{\sqrt[4]{k}})$ |
| | | | $F$ loc.Lip; $r$ cld; $f(\cdot;s) + r(\cdot)$ cvx; $f(\cdot;s)$ Lip; | $\alpha_0 = \mathcal{O}(\frac{1}{\sqrt{k}})$ | func.gap $\mathcal{O}(\frac{1}{\sqrt{k}})$ (ergodic) |
| | | | $F$ loc.Lip; $r$ cld; $f(\cdot;s) + r(\cdot)$ s.cvx; $f(\cdot;s)$ Lip; | $\alpha_0 = \mathcal{O}(\frac{1}{k})$ | func.gap $\mathcal{O}(\frac{1}{k})$ (ergodic) |
| M-SPP | Yuan & Li (2023) | $r = r + \mathbf{1}_{\mathcal{X}}$ | $f(\cdot;s)$ cvx & Lip.sm; $r$ cvx; $\mathcal{X}$ cld.cvx; $\phi$ qd.grow | $\frac{\alpha_0}{k}$ | func.gap $\mathcal{O}(\frac{1}{k^2} + \frac{1}{mk})$ (ergodic) $\{\,m$ l.e$\}$ $\{\,\alpha_0$ l.e$\}$ |
| | | | $f(\cdot;s)$ cvx & Lip.sm; $r$ cvx; $\mathcal{X}$ cld.cvx | $\alpha_0 = \mathcal{O}(\frac{1}{\sqrt{k}})$ | func.gap $\mathcal{O}(\frac{1}{\sqrt{mk}} + \frac{1}{mk})$ (ergodic) $\{\,\alpha_0 = \text{const}(m,k,\text{Lip}_f)\}$ |
| Inexact M-SPP | Yuan & Li (2023) | $r = r + \mathbf{1}_{\mathcal{X}}$ | $f(\cdot;s)$ cvx & Lip.sm; $r$ cvx & Lip; $\mathcal{X}$ cld.cvx; $\phi$ qd.grow | $\frac{\alpha_0}{k}$ | func.gap $\mathcal{O}(\frac{1}{k^2} + \frac{1}{mk})$ (ergodic) $\{\,m$ l.e$\}$ $\{\,\alpha_0$ l.e & $\epsilon_k = \mathcal{O}(\frac{1}{mk^4})\}$ |
| | | | $f(\cdot;s)$ cvx & Lip.sm; $r$ cvx & Lip; $\mathcal{X}$ cld.cvx | $\alpha_0 = \mathcal{O}(\frac{1}{\sqrt{k}})$ | func.gap $\mathcal{O}(\frac{1}{\sqrt{mk}} + \frac{1}{mk})$ (ergodic) $\{\,\alpha_0 = \text{const}(m,k,\text{Lip}_f)\}$ $\{\,\epsilon_k = \mathcal{O}(\frac{1}{m^2k^5})\}$ |
| isPPA | Theorem 2.3 | - | $F$ loc.Lip; $r$ cld; $f(\cdot;s) + r(\cdot)$ cvx; $f(\cdot;s)$ Lip; $\phi$ qd.grow | $\alpha_0$ | linear convergence to $\mathcal{O}(\alpha_0)$ (sq.dist.set) $\{\,\epsilon_k = \gamma\alpha_0^{\frac{3}{2}}\}$ |
| | Theorem 2.4 | | | $\frac{\alpha_0}{k^\beta}$ | sq.dist.set $\mathcal{O}(\frac{1}{k^\beta})$ $\{\,\epsilon_k = \gamma\alpha_k^{\frac{3}{2}}\}$ $\{\,\beta \in (0,1)\}$ or $\{\,\beta = 1$ & $\alpha_0$ l.e.$\}$ |

Here $\mathbf{1}_{\mathcal{X}}$ denotes the indicator function of a set $\mathcal{X} \subset \mathbb{R}^d$, which is defined by

$$\mathbf{1}_{\mathcal{X}}(x) \triangleq \begin{cases} 0 & \text{if } x \in \mathcal{X}, \\ \infty & \text{otherwise.} \end{cases}$$

The symbol "(ergodic)" appearing in the table indicates the ergodic convergence rate. Otherwise, the rate shown refers to the last-iterate convergence rate by default.

## B  PROOFS FOR THE RESULTS IN SECTION 2.1

In this section, we provide the detailed proof for Theorem 2.1 and 2.2 presented in Section 2.1. Recall that the notation $\tilde{x}_{k+1}$ is defined as the proximal point $\text{prox}_{\alpha_k \overline{\varphi}(\cdot;S_k^{1:m})}(x_k)$, and the symbols $\eta_{f,U}$ and $\rho_{f,m,U}$ given in (2.4) are defined by

$$\eta_{f,U} \triangleq 1 - \frac{\text{Var}\left(L_{f,U}(s)\right)}{\mathbb{E}_{s\sim P}\left[L_{f,U}(s)^2\right]} \quad \text{and} \quad \rho_{f,m,U} \triangleq \left(\sqrt{\frac{1 + (m-1)\eta_{f,U}}{m}} + 1\right)^2.$$

The following three technical lemmas are needed in our proof argument.

**Lemma B.1.** *The following assertions hold:*

*(i) Let Assumption 1 hold. Then the function $\phi$ is proper and closed convex.*

(ii) *Let Assumption 3 hold. Then the function $F$ is $L_F(U)$-Lipschitz over any bounded open subset $U \subset \mathrm{dom}(r)$, i.e.,*

$$|F(x) - F(y)| \leq L_F(U)\|x - y\|_2 \quad \text{for all } x, y \in U.$$

**Lemma B.2.** *Let Assumption 1 hold and let $\{x_k\}$ be generated by isPPA (Algorithm 1) with stepsizes $\{\alpha_k\}$. Then for all $x \in \mathbb{R}^d$ and $k \in \mathbb{Z}_+$, we have*

$$\|\tilde{x}_{k+1} - x\|_2^2 \leq \|x_k - x\|_2^2 - 2\alpha_k \left[\overline{\varphi}\left(\tilde{x}_{k+1}; S_k^{1:m}\right) - \overline{\varphi}\left(x; S_k^{1:m}\right)\right] - \|\tilde{x}_{k+1} - x_k\|_2^2. \quad \text{(B.1)}$$

**Lemma B.3.** *Let Assumption 1 and 3 hold, and let $\{x_k\}$ be generated by isPPA (Algorithm 1) with stepsizes $\{\alpha_k\}$. Suppose that the sequences $\{x_k\}$ and $\{\tilde{x}_k\}$ are contained in some bounded open set $U \subset \mathrm{dom}(r)$. Then for all $x \in \mathbb{R}^d$ and $k \in \mathbb{Z}_+$, we have*

$$\mathbb{E}_k\left[\|\tilde{x}_{k+1} - x\|_2^2\right] \leq \|x_k - x\|_2^2 - 2\alpha_k \cdot \mathbb{E}_k\left[\phi(\tilde{x}_{k+1}) - \phi(x)\right] + \rho_{f,m,U} L_F(U)^2 \cdot \alpha_k^2, \quad \text{(B.2)}$$

*which implies*

$$\mathbb{E}_k\left[\|x_{k+1} - x\|_2^2\right] \leq \left(1 + \frac{1}{t}\right)\left(\|x_k - x\|_2^2 - 2\alpha_k \cdot \mathbb{E}_k\left[\phi(\tilde{x}_{k+1}) - \phi(x)\right]\right)$$
$$+ \left(1 + \frac{1}{t}\right)\rho_{f,m,U} L_F(U)^2 \cdot \alpha_k^2 + (1 + t)\epsilon_k^2 \quad \text{(B.3)}$$

*for any $t \in \mathbb{R}_{++}$.*

The proof of the previous three lemmas are deferred to Appendix B.3.

## B.1 PROOF OF THEOREM 2.1

**Proof of assertion (i).** Fix any $\overline{x} \in \mathcal{X}^*$ and $k \in \mathbb{Z}_+$. Note that $\phi$ is convex by Lemma B.1(i). It follows that $0 \in \partial\phi(\overline{x})$ and thus, by the second condition in Assumption 2, we can take $\varphi'(\overline{x}; S_k^i) \in \partial\varphi(\overline{x}; S_k^i)$ such that

$$\mathbb{E}_k\left[\varphi'(\overline{x}; S_k^i)\right] = 0 \quad \text{for } i = 1, \cdots, m. \quad \text{(B.4)}$$

Set $\overline{\varphi}'(\overline{x}; S_k^{1:m}) \triangleq \frac{1}{m}\sum_{i=1}^m \varphi'(\overline{x}; S_k^i)$. Then $\overline{\varphi}'(\overline{x}; S_k^{1:m}) \in \partial\overline{\varphi}(\overline{x}; S_k^{1:m})$ and $\mathbb{E}_k\left[\overline{\varphi}'(\overline{x}; S_k^{1:m})\right] = 0$. Based on the convexity of $\overline{\varphi}(\cdot; S_k^{1:m})$ implied by Assumption 1, we have

$$\overline{\varphi}\left(\overline{x}; S_k^{1:m}\right) - \overline{\varphi}\left(\tilde{x}_{k+1}; S_k^{1:m}\right) \leq \left\langle \overline{\varphi}'\left(\overline{x}; S_k^{1:m}\right), \overline{x} - \tilde{x}_{k+1}\right\rangle$$
$$= \left\langle \overline{\varphi}'\left(\overline{x}; S_k^{1:m}\right), \overline{x} - x_k\right\rangle + \left\langle \overline{\varphi}'\left(\overline{x}; S_k^{1:m}\right), x_k - \tilde{x}_{k+1}\right\rangle \quad \text{(B.5)}$$
$$\leq \left\langle \overline{\varphi}'\left(\overline{x}; S_k^{1:m}\right), \overline{x} - x_k\right\rangle + \frac{\alpha_k}{2}\left\|\overline{\varphi}'\left(\overline{x}; S_k^{1:m}\right)\right\|_2^2 + \frac{1}{2\alpha_k}\|\tilde{x}_{k+1} - x_k\|_2^2,$$

where the last inequality uses the Cauchy-Schwarz inequality. By setting $x = \overline{x}$ in inequality (B.1) of Lemma B.2 and combining it with (B.5), we can conclude that

$$\|\tilde{x}_{k+1} - \overline{x}\|_2^2$$
$$\leq \|x_k - \overline{x}\|_2^2 + 2\alpha_k\left[\left\langle \overline{\varphi}'\left(\overline{x}; S_k^{1:m}\right), \overline{x} - x_k\right\rangle + \frac{\alpha_k}{2}\left\|\overline{\varphi}'\left(\overline{x}; S_k^{1:m}\right)\right\|_2^2\right.$$
$$\left. + \frac{1}{2\alpha_k}\|\tilde{x}_{k+1} - x_k\|_2^2\right] - \|\tilde{x}_{k+1} - x_k\|_2^2 \quad \text{(B.6)}$$
$$= \|x_k - \overline{x}\|_2^2 + 2\alpha_k\left\langle \overline{\varphi}'\left(\overline{x}; S_k^{1:m}\right), \overline{x} - x_k\right\rangle + \alpha_k^2\left\|\overline{\varphi}'\left(\overline{x}; S_k^{1:m}\right)\right\|_2^2.$$

Fix any $t \in \mathbb{R}_{++}$. It follows directly from the Cauchy-Schwarz inequality and criterion (SCA) that

$$\|x_{k+1} - x\|_2^2 = \|x_{k+1} - \tilde{x}_{k+1}\|_2^2 + \|\tilde{x}_{k+1} - x\|_2^2 + 2\langle x_{k+1} - \tilde{x}_{k+1}, \tilde{x}_{k+1} - x\rangle$$
$$\leq \|x_{k+1} - \tilde{x}_{k+1}\|_2^2 + \|\tilde{x}_{k+1} - x\|_2^2 + t\|x_{k+1} - \tilde{x}_{k+1}\|_2^2 + \frac{1}{t}\|\tilde{x}_{k+1} - x\|_2^2 \quad \text{(B.7)}$$
$$\leq \left(1 + \frac{1}{t}\right)\|\tilde{x}_{k+1} - x\|_2^2 + (1 + t)\epsilon_k^2 \quad \text{for all } x \in \mathbb{R}^d.$$

It follows from the definition of $\overline{\varphi}'\left(\overline{x}; S_k^{1:m}\right)$ that

$$
\begin{aligned}
\mathbb{E}_k\left[\left\|\overline{\varphi}'\left(\overline{x}; S_k^{1:m}\right)\right\|_2^2\right] &= \mathbb{E}_k\left[\left\|\frac{1}{m}\sum_{i=1}^k \varphi'\left(\overline{x}; S_k^i\right)\right\|_2^2\right] \\
&= \mathbb{E}_k\left[\left\|\frac{1}{m}\sum_{i=1}^k \left(\varphi'\left(\overline{x}; S_k^i\right) - \mathbb{E}_k\left[\varphi'\left(\overline{x}; S_k^i\right)\right]\right)\right\|_2^2\right] \\
&= \frac{1}{m^2}\sum_{i=1}^m \mathbb{E}_k\left[\left\|\varphi'\left(\overline{x}; S_k^i\right) - \mathbb{E}_k\left[\varphi'\left(\overline{x}; S_k^i\right)\right]\right\|_2^2\right] \\
&= \frac{1}{m^2}\sum_{i=1}^m \mathbb{E}_k\left[\left\|\varphi'\left(\overline{x}; S_k^i\right)\right\|_2^2\right],
\end{aligned}
\tag{B.8}
$$

where the second equality uses condition (B.4), the third holds because of the independence of the random variables and the last use again condition (B.4). By setting $x = \overline{x}$, $t = \frac{1}{\alpha_k^2}$ and $\epsilon_k = \gamma\alpha_k^2$ in inequality (B.7), we have

$$
\begin{aligned}
\|x_{k+1} - \overline{x}\|_2^2 &\le \left(1 + \alpha_k^2\right)\|\tilde{x}_{k+1} - \overline{x}\|_2^2 + \left(1 + \alpha_k^2\right)\alpha_k^2\gamma^2 \\
&\le \left(1 + \alpha_k^2\right)\left[\|x_k - \overline{x}\|_2^2 + 2\alpha_k\left\langle\overline{\varphi}'\left(\overline{x}; S_k^{1:m}\right), \overline{x} - x_k\right\rangle + \alpha_k^2\left\|\overline{\varphi}'\left(\overline{x}; S_k^{1:m}\right)\right\|_2^2\right] \\
&\quad + \left(1 + \alpha_k^2\right)\alpha_k^2\gamma^2,
\end{aligned}
$$

where the second inequality follows from (B.6). Taking expectations on both sides of the previous inequality with respect to $\mathcal{F}_{k-1}$ yields that

$$
\begin{aligned}
\mathbb{E}_k\left[\|x_{k+1} - \overline{x}\|_2^2\right] &\le \left(1 + \alpha_k^2\right)\|x_k - \overline{x}\|_2^2 + \left(1 + \alpha_k^2\right)\alpha_k^2\left(\mathbb{E}_k\left[\left\|\overline{\varphi}'\left(\overline{x}; S_k^{1:m}\right)\right\|_2^2\right] + \gamma^2\right) \\
&= \left(1 + \alpha_k^2\right)\|x_k - \overline{x}\|_2^2 + \left(1 + \alpha_k^2\right)\alpha_k^2\left(\frac{1}{m^2}\sum_{i=1}^m \mathbb{E}_k\left[\left\|\varphi'\left(\overline{x}; S_k^i\right)\right\|_2^2\right] + \gamma^2\right) \\
&\le \left(1 + \alpha_k^2\right)\|x_k - \overline{x}\|_2^2 + \left(1 + \alpha_k^2\right)\alpha_k^2\left(\frac{\sigma_\phi^2}{m} + \gamma^2\right),
\end{aligned}
$$

where the first inequality comes from the condition that $\mathbb{E}_k\left[\overline{\varphi}'(\overline{x}; S_k^{1:m})\right] = 0$ and the last is due to Assumption 2, and the equality holds because of (B.8). This completes the proof of inequality (2.5) in assertion (i).

**Proof of assertion (ii).** Now we assume that $\sum_{k=1}^\infty \alpha_k^2 < \infty$. To prove assertion (ii), we first recall the so-called supermartingale convergence lemma, the proof of which can be found in (Robbins & Siegmund, 1971, Theorem 1).

**Lemma B.4.** *Let $Z_k$, $\beta_k$, $\xi_k$ and $\zeta_k$ be nonnegative random variables adapted to the filtration $\mathcal{F}_k$ and satisfying*

$$
\mathbb{E}\left[Z_{k+1} \mid \mathcal{F}_k\right] \le \left(1 + \beta_k\right)Z_k + \xi_k - \zeta_k
$$

*for all $k \in \mathbb{Z}_+$. Then on the event that $\sum_{k=1}^\infty \beta_k < \infty$ and $\sum_{k=1}^\infty \xi_k < \infty$, with probability $1$, we have $\sum_{k=1}^\infty \zeta_k < \infty$ and there exists a random variable $Z_\infty$ such that $\lim_{k\to\infty} Z_k = Z_\infty$.*

Fix any $\overline{x} \in \mathcal{X}^*$. The assumption that $\sum_{k=1}^\infty \alpha_k^2 < \infty$ implies $\alpha_k \le 1$ for $k$ sufficiently large and thus

$$
\sum_{k=1}^\infty \left(1 + \alpha_k^2\right)\alpha_k^2 < \infty.
$$

By Lemma B.4, given the conditions that $\sum_{k=1}^\infty \alpha_k^2 < \infty$ and $\sum_{k=1}^\infty \left(1 + \alpha_k^2\right)\alpha_k^2 < \infty$, we can deduce from the derived inequality (2.5) that the sequence $\{\|x_k - \overline{x}\|_2\}$ converges to some finite value with probability $1$. This implies (2.6), thus completing the proof of assertion (ii).

## B.2 PROOF OF THEOREM 2.2

By assumptions given in Theorem 2.2, we can deduce from Theorem 2.1 that with probability 1,

$$\sup_{k \in \mathbb{Z}_+} \|x_k\|_2 < \infty \tag{B.9}$$

Then, to prove Theorem 2.2, it is sufficient to show that on the event (B.9) holds, with probability 1 both

(i) $\liminf_{k \to \infty} \phi(\tilde{x}_k) = \phi^*$ and

(ii) the sequence $\{x_k\}$ converges to some $x^* \in \mathcal{X}^*$.

On the event that (B.9) holds, criterion (SCA) along with the conditions that $\epsilon_k = \gamma \alpha_k^2$ and $\sum_{k=1}^{\infty} \alpha_k^2 < \infty$ implies $\sup_{k \in Z_+} \|\tilde{x}_k\|_2 < \infty$. Thus, we denote by $U$ a bounded open subset of $\mathrm{dom}(r)$ satisfying

$$\{x_k\}_{k \in \mathbb{Z}_+} \cup \{\tilde{x}_k\}_{k \in \mathbb{Z}_+} \subset U.$$

By setting $t = \frac{1}{\alpha_k^2}$ and $\epsilon_k = \gamma \alpha_k^2$ in inequality (B.3) of Lemma B.3, we have

$$\begin{aligned}
\mathbb{E}_k \left[ \|x_{k+1} - x\|_2^2 \right] &\leq \left(1 + \alpha_k^2\right) \left( \|x_k - x\|_2^2 - 2\alpha_k \cdot \mathbb{E}_k \left[\phi(\tilde{x}_{k+1}) - \phi(x)\right] \right) \\
&\quad + \left(\rho_{f,m,U} L_F(U)^2 + \gamma^2\right) \cdot \left(1 + \alpha_k^2\right) \alpha_k^2
\end{aligned} \tag{B.10}$$

for all $k \in \mathbb{Z}_+$ and $x \in \mathbb{R}^d$. To prove the assertion (i) and (ii) above, we recall the following elementary inequality:

$$1 + t \leq \exp(t) \quad \text{for all } t \in \mathbb{R}. \tag{B.11}$$

**Proof of assertion (i).** Suppose on contrary that there exists $\epsilon > 0$ such that

$$\mathbb{P}\left\{ \liminf_{k \to \infty} \phi(\tilde{x}_k) > \phi^* + 3\epsilon \right\} > 0. \tag{B.12}$$

The definition of $\phi^*$ implies the existence of $\hat{x} \in \mathbb{R}^d$ such that

$$\phi^* + \epsilon \geq \phi(\hat{x}). \tag{B.13}$$

On the event that $\liminf_{k \to \infty} \phi(\tilde{x}_k) > \phi^* + 3\epsilon$, it follows from (B.13) that

$$\liminf_{k \to \infty} \phi(\tilde{x}_k) \geq \phi(\hat{x}) + 2\epsilon. \tag{B.14}$$

Note that we can pick some $K \in \mathbb{Z}_+$ such that

$$\phi(\tilde{x}_k) \geq \liminf_{k \to \infty} \phi(\tilde{x}_k) - \epsilon \quad \text{and} \tag{B.15a}$$

$$\left(\rho_{f,m,U} L_F(U)^2 + \gamma^2\right) \cdot \alpha_k \leq \epsilon \quad \text{for all } k \in \mathbb{Z}_+ \text{ and } k \geq K, \tag{B.15b}$$

where the first inequality uses the definition of the limit inferior of $\{\phi(\tilde{x}_k)\}$, and the second follows from the fact $\lim_{k \to \infty} \alpha_k = 0$ implied by the condition $\sum_{k=1}^{\infty} \alpha_k^2 < \infty$. Without loss of generality, we assume that $K \geq 2$.

- Fix any $k \in \mathbb{Z}_+$ with $k \geq K$. Then (B.14) together with (B.15a) yields $\phi(\tilde{x}_{k+1}) - \phi(\hat{x}) \geq \epsilon$ and thus

$$\mathbb{E}_k \left[\phi(\tilde{x}_{k+1}) - \phi(\hat{x})\right] \geq \epsilon.$$

It then follows from the previous inequality and (B.10) with $x = \hat{x}$ that

$$\mathbb{E}_k \left[ \|x_{k+1} - \hat{x}\|_2^2 \right] \leq \left(1 + \alpha_k^2\right) \left( \|x_k - \hat{x}\|_2^2 - 2\alpha_k \epsilon + \left(\rho_{f,m,U} L_F(U)^2 + \gamma^2\right) \cdot \alpha_k^2 \right).$$

By taking the expectation of both sides of the preceding inequality and applying the law of total expectation, we can derive

$$\begin{aligned}
\mathbb{E} \left[ \|x_{k+1} - \hat{x}\|_2^2 \right] &\leq \left(1 + \alpha_k^2\right) \left( \mathbb{E} \left[ \|x_k - \hat{x}\|_2^2 \right] - 2\alpha_k \epsilon + \left(\rho_{f,m,U} L_F(U)^2 + \gamma^2\right) \cdot \alpha_k^2 \right) \\
&\leq \left(1 + \alpha_k^2\right) \left( \mathbb{E} \left[ \|x_k - \hat{x}\|_2^2 \right] - \alpha_k \epsilon \right),
\end{aligned}$$

where the second inequality comes from (B.15b). Thus,

$$
\begin{aligned}
0 \leq \mathbb{E}\left[\|x_{k+1} - \hat{x}\|_2^2\right] &\leq \left(1 + \alpha_k^2\right)\left(\mathbb{E}\left[\|x_k - \hat{x}\|_2^2\right] - \alpha_k \epsilon\right) \\
&\leq \left(1 + \alpha_k^2\right)\left(\left(1 + \alpha_{k-1}^2\right)\left(\mathbb{E}\left[\|x_{k-1} - \hat{x}\|_2^2\right] - \alpha_{k-1}\epsilon\right) - \alpha_k\epsilon\right) \\
&\leq \left(1 + \alpha_k^2\right)\left(\left(1 + \alpha_{k-1}^2\right)\mathbb{E}\left[\|x_{k-1} - \hat{x}\|_2^2\right] - \alpha_{k-1}\epsilon - \alpha_k\epsilon\right) \\
&\leq \cdots \leq \left(1 + \alpha_k^2\right)\left(\prod_{i=K}^{k-1}\left(1 + \alpha_i^2\right)\mathbb{E}\left[\|x_K - \hat{x}\|_2^2\right] - \epsilon\sum_{i=K}^k \alpha_i\right)
\end{aligned}
$$

which combined with (B.11) and the assumption that $\sum_{k=1}^\infty \alpha_k^2 < \infty$ implies

$$
\begin{aligned}
\sum_{i=K}^k \alpha_i &\leq \frac{1}{\epsilon}\prod_{i=K}^{k-1}\left(1 + \alpha_i^2\right)\mathbb{E}\left[\|x_K - \hat{x}\|_2^2\right] \leq \frac{1}{\epsilon}\exp\left(\sum_{i=K}^{k-1}\alpha_i^2\right)\mathbb{E}\left[\|x_K - \hat{x}\|_2^2\right] \\
&\leq \frac{1}{\epsilon}\exp\left(\sum_{i=K}^\infty \alpha_i^2\right)\mathbb{E}\left[\|x_K - \hat{x}\|_2^2\right] \quad \text{for all } k \in \mathbb{Z}_+ \text{ and } k \geq K.
\end{aligned}
$$

Taking $k$ to $\infty$ yields that

$$
\sum_{k=K}^\infty \alpha_k \leq \frac{1}{\epsilon}\exp\left(\sum_{i=K}^\infty \alpha_i^2\right)\mathbb{E}\left[\|x_K - \hat{x}\|_2^2\right]. \tag{B.16}
$$

- Additionally, for each $k \in \mathbb{Z}_+$, it holds that

$$
\begin{aligned}
&\mathbb{E}_k\left[\|x_{k+1} - \hat{x}\|_2^2\right] \\
&\leq \left(1 + \alpha_k^2\right)\left(\|x_k - \hat{x}\|_2^2 - 2\alpha_k \cdot \mathbb{E}_k\left[\phi(\tilde{x}_{k+1}) - \phi^* + \phi^* - \phi(\hat{x})\right]\right) \\
&\quad + \left(\rho_{f,m,U}L_F(U)^2 + \gamma^2\right)\cdot\left(1 + \alpha_k^2\right)\alpha_k^2 \\
&\leq \left(1 + \alpha_k^2\right)\left(\|x_k - \hat{x}\|_2^2 + 2\alpha_k\epsilon\right) + \left(\rho_{f,m,U}L_F(U)^2 + \gamma^2\right)\cdot\left(1 + \alpha_k^2\right)\alpha_k^2,
\end{aligned}
$$

where the first inequality is due to (B.10) with $x = \hat{x}$, and the second follows from (B.13) and the fact that $\phi(\tilde{x}_{k+1}) \geq \phi^*$. Taking the expectation of both sides of the preceding inequality and applying the law of total expectation gives

$$
\mathbb{E}\left[\|x_{k+1} - \hat{x}\|_2^2\right] \leq \left(1 + \alpha_k^2\right)\left(\mathbb{E}\left[\|x_k - \hat{x}\|_2^2\right] + 2\alpha_k\epsilon + \left(\rho_{f,m,U}L_F(U)^2 + \gamma^2\right)\cdot\alpha_k^2\right).
$$

Applying this recursively $k$ times leads to

$$
\begin{aligned}
&\mathbb{E}\left[\|x_{k+1} - \hat{x}\|_2^2\right] \\
&\leq \prod_{i=1}^k\left(1 + \alpha_i^2\right)\|x_1 - \hat{x}\|_2^2 + 2\epsilon\sum_{i=1}^k \alpha_i \prod_{j=i}^k\left(1 + \alpha_j^2\right) \\
&\quad + \left(\rho_{f,m,U}L_F(U)^2 + \gamma^2\right)\sum_{i=1}^k \alpha_i^2 \prod_{j=i}^k\left(1 + \alpha_j^2\right) \\
&\leq \prod_{i=1}^k\left(1 + \alpha_i^2\right)\left(\|x_1 - \hat{x}\|_2^2 + 2\epsilon\sum_{i=1}^k \alpha_i + \left(\rho_{f,m,U}L_F(U)^2 + \gamma^2\right)\sum_{i=1}^k \alpha_i^2\right) \\
&\leq \exp\left(\sum_{i=1}^k \alpha_i^2\right)\left(\|x_1 - \hat{x}\|_2^2 + 2\epsilon\sum_{i=1}^k \alpha_i + \left(\rho_{f,m,U}L_F(U)^2 + \gamma^2\right)\sum_{i=1}^k \alpha_i^2\right),
\end{aligned}
$$

and thus

$$
\begin{aligned}
&\mathbb{E}\left[\|x_K - \hat{x}\|_2^2\right] \\
&\leq \exp\left(\sum_{i=1}^{K-1}\alpha_i^2\right)\left(\|x_1 - \hat{x}\|_2^2 + 2\epsilon\sum_{i=1}^{K-1}\alpha_i + \left(\rho_{f,m,U}L_F(U)^2 + \gamma^2\right)\sum_{i=1}^{K-1}\alpha_i^2\right).
\end{aligned} \tag{B.17}
$$

Combining (B.16) with (B.17) confirms that

$$
\sum_{k=1}^{\infty} \alpha_k \le \sum_{k=1}^{K-1} \alpha_k + \frac{1}{\epsilon} \exp\left(\sum_{i=K}^{\infty} \alpha_i^2\right) \mathbb{E}\left[\|x_K - \hat{x}\|_2^2\right]
$$

$$
\le \sum_{k=1}^{K-1} \alpha_k + \frac{1}{\epsilon} \exp\left(\sum_{i=1}^{\infty} \alpha_i^2\right) \left(\|x_1 - \hat{x}\|_2^2 + 2\epsilon \sum_{i=1}^{K-1} \alpha_i + \left(\rho_{f,m,U} L_F(U)^2 + \gamma^2\right) \sum_{i=1}^{K-1} \alpha_i^2\right)
$$

$$
< \infty
$$

on the event that $\liminf_{k\to\infty} \phi(\tilde{x}_k) > \phi^* + 3\epsilon$. Then (B.12) implies that

$$
\mathbb{P}\left\{\sum_{k=1}^{\infty} \alpha_k < \infty\right\} \ge \mathbb{P}\left\{\liminf_{k\to\infty} \phi(\tilde{x}_k) > \phi^* + 3\epsilon\right\} > 0,
$$

which leads to a contradiction of the assumption that $\sum_{k=1}^{\infty} \alpha_k = \infty$. Therefore, on the event that (B.9) holds, $\liminf_{k\to\infty} \phi(\tilde{x}_k) = \phi^*$ holds with probability 1.

**Proof of assertion (ii).** Since $\mathcal{X}^*$ is nonempty by Assumption 2, assertion (i) derived above implies $\liminf_{k\to\infty} \phi(\tilde{x}_k) = \phi^* \in \mathbb{R}$. Then by using the definition of the limit inferior of $\{\phi(\tilde{x}_k)\}$, one can easily deduce that there exists a convergent subsequence $\{\phi(\tilde{x}_k)\}_{k\in\mathcal{K}}$ of $\{\phi(\tilde{x}_k)\}$ such that

$$
\lim_{k\to\infty, k\in\mathcal{K}} \phi(\tilde{x}_k) = \liminf_{k\to\infty} \phi(\tilde{x}_k) = \phi^*. \tag{B.18}
$$

Fix any $\overline{x} \in \mathcal{X}^*$. It then follows from (B.10) with $x = \overline{x}$ that

$$
\mathbb{E}_k\left[\|x_{k+1} - \overline{x}\|_2^2\right]
$$

$$
\le \left(1 + \alpha_k^2\right)\left(\|x_k - \overline{x}\|_2^2 - 2\alpha_k \cdot \mathbb{E}_k\left[\phi(\tilde{x}_{k+1}) - \phi(\overline{x})\right] + \left(\rho_{f,m,U} L_F(U)^2 + \gamma^2\right) \cdot \alpha_k^2\right)
$$

$$
\le \left(1 + \alpha_k^2\right)\|x_k - \overline{x}\|_2^2 + \left(\rho_{f,m,U} L_F(U)^2 + \gamma^2\right) \cdot \alpha_k^2 \left(1 + \alpha_k^2\right) \quad \text{for all } k \in \mathbb{Z}_+,
$$

which combined with Lemma B.4 and the condition $\sum_{k=1}^{\infty} \alpha_k^2 < \infty$ confirms that the sequence $\{\|x_k - \overline{x}\|_2\}$ converges to some finite value with probability 1. That is, for each $\overline{x} \in \mathcal{X}^*$,

the sequence $\{\|x_k - \overline{x}\|_2\}$ converges with probability 1.

Combining the previous result with the fact that

$$
\lim_{k\to\infty} \alpha_k = 0 \quad \text{and} \quad \|x_{k+1} - \tilde{x}_{k+1}\|_2 \le \epsilon_k = \gamma \alpha_k^2 \quad \text{for all } k \in \mathbb{Z}_+ \tag{B.19}
$$

yields

the sequence $\{\|\tilde{x}_k - \overline{x}\|_2\}$ converges with probability 1. $\tag{B.20}$

The proof of assertion (ii) is as follows:

- If $\mathcal{X}^* = \{x^*\}$ is a singleton, then by (B.20), we have $\{\|\tilde{x}_k - x^*\|_2\}$ converges with probability 1. Note that $\{\tilde{x}_k\}$ is bounded and thus its subsequence $\{\tilde{x}_k\}_{k\in\mathcal{K}}$ is bounded. Then there exists a subsequence $\{\tilde{x}_k\}_{k\in\mathcal{K}'}$ of $\{\tilde{x}_k\}_{k\in\mathcal{K}}$ such that $\{\tilde{x}_k\}_{k\in\mathcal{K}'}$ converges to some $\tilde{x} \in \mathbb{R}^d$. Since $\phi$ is closed by Lemma B.1(i), it follows from (B.18) that

$$
\phi(\tilde{x}) \le \liminf_{k\to\infty, k\in\mathcal{K}'} \phi(\tilde{x}_k) = \lim_{k\to\infty, k\in\mathcal{K}'} \phi(\tilde{x}_k) = \lim_{k\to\infty, k\in\mathcal{K}} \phi(\tilde{x}_k) = \phi^*,
$$

  which implies $\phi(\tilde{x}) = \phi^*$ and $\tilde{x} = x^*$. Hence, on the event that $\{\|\tilde{x}_k - x^*\|_2\}$ converges, we have

$$
\lim_{k\to\infty} \|\tilde{x}_k - x^*\|_2 = \lim_{k\to\infty, k\in\mathcal{K}'} \|\tilde{x}_k - x^*\|_2 = \|\tilde{x} - x^*\|_2 = 0,
$$

  where the second equality comes from the fact that $\{\tilde{x}_k\}_{k\in\mathcal{K}'}$ converges to $\tilde{x}$.

- Now we consider the case where $\mathcal{X}^*$ contains two or more points. Since $\phi$ is convex by Lemma B.1(i), then $\mathcal{X}^*$ is convex and thus infinite. Thus, $\mathcal{X}^*$, being a infinite subset of the seperable metric space $(\mathbb{R}^d, \|\cdot\|_2)$, is seperable and we can choose a countable dense subset $V$ of $\mathcal{X}^*$. For each $v \in V$, by (B.20) we have $\{\|\tilde{x}_k - v\|_2\}$ converges with probability 1, which combined with the countability of $V$ leads to the conclusion that with probability 1,

the sequence $\{\|\tilde{x}_k - v\|_2\}$ converges for all $v \in V$. $\tag{B.21}$

By following an argument nearly identical to that in the discssion for the case where $\mathcal{X}^*$ is a singleton, we can obtain that there exists a subsequence $\{\tilde{x}_k\}_{k \in \mathcal{K}'}$ converges to some $\tilde{x} \in \mathcal{X}^*$. Fix any $\epsilon > 0$. The fact that $V$ is dense in $\mathcal{X}^*$ implies that there exists $v_\epsilon \in V$ such $\|\tilde{x} - v_\epsilon\|_2 < \epsilon$. Then, on the event that (B.21) holds, we have

$$\lim_{k \to \infty} \|\tilde{x}_k - v_\epsilon\|_2 = \lim_{k \to \infty, k \in \mathcal{K}'} \|\tilde{x}_k - v_\epsilon\|_2 = \|\tilde{x} - v_\epsilon\|_2 < \epsilon,$$

where the second equality uses the fact that $\{\tilde{x}_k\}_{k \in \mathcal{K}'}$ converges to $\tilde{x}$. By combining the preceding inequality with the triangle inequality, one can easily derive

$$\limsup_{k \to \infty} \|\tilde{x}_k - \tilde{x}\|_2 \leq \lim_{k \to \infty} \|\tilde{x}_k - v_\epsilon\|_2 + \|v_\epsilon - \tilde{x}\|_2 < 2\epsilon.$$

Taking $\epsilon$ to 0 yields that $\lim_{k \to \infty} \|\tilde{x}_k - \tilde{x}\|_2 = 0$ with $\tilde{x} \in \mathcal{X}^*$.

Therefore, on the event that (B.9) holds, there exists some $x^* \in \mathcal{X}^*$ such that

$$\{\tilde{x}_k\} \text{ converges to } x^* \text{ with probability } 1,$$

which combined with (B.19) yields assertion (ii). This completes the proof of Theorem 2.2.

**Remark B.1.** It is worth noting that the proof technique adopted here differs from that of Proposition 3.8 in (Asi & Duchi, 2019). For the case where all the assumptions in Theorem 2.2 hold, even if the accuracy parameter $\epsilon_k \equiv 0$, i.e., subproblems are solved exactly, we cannot directly apply the argument in the proof of (Asi & Duchi, 2019, Proposition 3.8) to deduce the almost sure convergence of isPPA. The reason for this is that when using (B.2) in Lemma B.3 with $x = \overline{x} \in \mathcal{X}^*$, we can only obtain

$$\mathbb{E}_k \left[ \|x_{k+1} - \overline{x}\|_2^2 \right] \leq \|x_k - \overline{x}\|_2^2 - 2\alpha_k \cdot \mathbb{E}_k \left[ \phi(x_{k+1}) - \phi^* \right] + \rho_{f,m,U} L_F(U)^2 \cdot \alpha_k^2 \quad \text{(B.22)}$$

for all $k \in \mathbb{Z}_+$. However, observing that the term appearing in the right-hand side of (B.22) is $\mathbb{E}_k \left[ \phi(x_{k+1}) - \phi^* \right]$ rather than $\phi(x_k) - \phi^*$, we cannot apply the supermartingale convergence lemma (see Lemma B.4) to conclude that

$$\sum_{k=1}^{\infty} \alpha_k \cdot \left[ \phi(x_k) - \phi^* \right] < \infty \quad \text{with probability } 1,$$

which is the key to deriving the desired convergence results via the proof technique of (Asi & Duchi, 2019, Proposition 3.8). Indeed, the proof technique provided in this paper is inspired by the proof of (Bertsekas, 2011, Proposition 9), with appropriate modifications to deal with the case where subproblems are solved inexactly.

## B.3 Proof of Auxiliary Lemmas

### B.3.1 Proof of Lemma B.1

Note that Assumption 1 and 3 correspond to (Davis & Drusvyatskiy, 2019, (C3)-(C4)) with $\rho = 0$. Lemma B.1 then follows by an argument almost identical to that used for (Davis & Drusvyatskiy, 2019, Lemma 4.1). The details are provided below:

(i) Suppose that Assumption 1 holds. Then by definition, function $\phi$ is proper and closed with $\text{dom}(\phi) = \text{dom}(r)$. It only remains to show that $\phi$ is convex. Pick some $s \in \mathcal{S}$ such that $f(\cdot; s)$ is proper, $\text{dom}(r) \subset \text{dom}(f(\cdot; s))$ and $\varphi(\cdot; s)$ is convex. Then $\text{dom}(r) = \text{dom}(\varphi(\cdot; s))$ and thus $\text{dom}(r)$ is convex, implying the convexity of $\text{dom}(\phi)$. Fix any $x, y \in \text{dom}(\phi) = \text{dom}(r)$ and $\theta \in [0, 1]$. Set $\overline{z} = \theta x + (1 - \theta) y$. We can easily deduce that

$$\phi(\overline{z}) = F(\overline{z}) + r(\overline{z}) = \mathbb{E}_{s \sim P} \left[ f(\overline{z}; s) \right] + r(\overline{z})$$

$$= \int_{\mathcal{S}} f(\overline{z}; s) \mathrm{d}\, P(s) + r(\overline{z}) = \int_{\mathcal{S}} (f(\overline{z}; s) + r(\overline{z})) \, \mathrm{d}\, P(s)$$

$$\leq \int_{\mathcal{S}} \left[ \theta \left( f(x; s) + r(x) \right) + (1 - \theta) \left( f(y; s) + r(y) \right) \right] \mathrm{d}\, P(s)$$

$$= \theta \left( F(x) + r(x) \right) + (1 - \theta) \left( F(y) + r(y) \right)$$

$$= \theta \phi(x) + (1 - \theta) \phi(y),$$

where the inequality uses Assumption 1. Then $\phi$ is convex, which completes the proof of assertion (i).

(ii) Now we assume that Assumption 3 holds. Let $U$ be a bounded open subset of $\mathrm{dom}(r)$ and fix any $x, y \in U$. It holds that

$$
\begin{aligned}
|F(x) - F(y)| &= \left| \int_{\mathcal{S}} [f(x; s) - f(y; s)] \, \mathrm{d}\, P(s) \right| \\
&\leq \int_{\mathcal{S}} |f(x; s) - f(y; s)| \, \mathrm{d}\, P(s) \\
&\leq \int_{\mathcal{S}} L_{f,U}(s) \|x - y\|_2 \mathrm{d}\, P(s) = \mathbb{E}_{s \sim P} \left[ L_{f,U}(s) \right] \cdot \|x - y\|_2 \\
&\leq \sqrt{\mathbb{E}_{s \sim P} \left[ L_{f,U}(s)^2 \right]} \cdot \|x - y\|_2 \leq L_F(U) \|x - y\|_2,
\end{aligned}
$$

where the second last inequality uses Jensen's inequality. This completes the proof of assertion (ii).

### B.3.2 PROOF OF LEMMA B.2

It follows from Assumption 1 that functions $\overline{\varphi}(x; S_k^{1:m})$ and $\overline{\varphi}(x; S_k^{1:m}) + \frac{1}{2\alpha_k} \|x - x_k\|_2^2$ are convex and $\frac{1}{\alpha_k}$-strongly convex with respect to the variable $x$, respectively. Then the definition of $\tilde{x}_{k+1}$ implies

$$
\begin{aligned}
&\left[ \overline{\varphi}\left(x; S_k^{1:m}\right) + \frac{1}{2\alpha_k} \|x - x_k\|_2^2 \right] \\
&\geq \left[ \overline{\varphi}\left(\tilde{x}_{k+1}; S_k^{1:m}\right) + \frac{1}{2\alpha_k} \|\tilde{x}_{k+1} - x_k\|_2^2 \right] + \frac{1}{2\alpha_k} \|x - \tilde{x}_{k+1}\|_2^2 \quad \text{for all } x \in \mathbb{R}^d,
\end{aligned}
$$

which completes the proof of Lemma B.2.

### B.3.3 PROOF OF LEMMA B.3

Inspired from the technique used in (Davis & Drusvyatskiy, 2019, Lemma 4.2), we provide the following proof of Lemma B.3. Recall that the terms $\eta_{f,U}$ and $\rho_{f,m,U}$ are defined as follows:

$$
\eta_{f,U} \triangleq 1 - \frac{\mathrm{Var}\left(L_{f,U}(s)\right)}{\mathbb{E}_{s \sim P}\left[L_{f,U}(s)^2\right]} \quad \text{and} \quad \rho_{f,m,U} \triangleq \left( \sqrt{\frac{1 + (m-1)\eta_{f,U}}{m}} + 1 \right)^2.
$$

In the case where $m = 1$, (B.2) follows directly from (Davis & Drusvyatskiy, 2019, (4.12)) with $\tau = \eta = 0$. To extend this to the minibatch case, we employ the following result whose proof is straightforward and provided at the end of this section.

**Lemma B.5.** *Let $X_1, \cdots, X_m$ be independent and identically distributed random variables with mean $\mathbb{E}[X_i] = \mu$ and variance $\mathrm{Var}(X_i) = \sigma^2$ for each $i = 1, \cdots, m$. If $\mathbb{E}[(X_1)^2] > 0$, then*

$$
\mathbb{E}\left[ \left( \frac{1}{m} \sum_{i=1}^{m} X_i \right)^2 \right] = \frac{1}{m}\sigma^2 + \mu^2 = \frac{1 + (m-1)\eta}{m} \mathbb{E}[(X_1)^2],
$$

*where $\eta \triangleq 1 - \frac{\mathrm{Var}(X_1)}{\mathbb{E}[(X_1)^2]}$.*

*Proof of Lemma B.3.* By assumption, $U$ is a bouned open subset of $\mathrm{dom}(r)$ which contains the sequences $\{x_k\}$ and $\{\tilde{x}_k\}$. Fix any $x \in \mathbb{R}^d$ and $k \in \mathbb{Z}_+$.

(i) Taking the conditional expectation with respect to $\mathcal{F}_{k-1}$ on both sides of inequality (B.1) in Lemma B.2 gives

$$
\begin{aligned}
&\mathbb{E}_k \left[ \|x - \tilde{x}_{k+1}\|_2^2 + \|\tilde{x}_{k+1} - x_k\|_2^2 \right] - \|x - x_k\|_2^2 \\
&\leq 2\alpha_k \cdot \mathbb{E}_k \left[ \overline{f}(x; S_k^{1:m}) + r(x) - \overline{f}(\tilde{x}_{k+1}; S_k^{1:m}) - r(\tilde{x}_{k+1}) \right].
\end{aligned} \tag{B.23}
$$

Let $t_k \triangleq \sqrt{\mathbb{E}_k[\|\tilde{x}_{k+1} - x_k\|_2^2]}$. Then we have

$$\mathbb{E}_k \left[ \|x - \tilde{x}_{k+1}\|_2^2 \right] + t_k^2 - \|x - x_k\|_2^2$$

$$\leq 2\alpha_k \cdot \mathbb{E}_k \left[ \overline{f}(x; S_k^{1:m}) + r(x) - \overline{f}(x_k; S_k^{1:m}) - r(\tilde{x}_{k+1}) \right]$$

$$+ 2\alpha_k \cdot \mathbb{E}_k \left[ \frac{1}{m} \sum_{i=1}^{m} L_{f,U}(S_k^i) \cdot \|\tilde{x}_{k+1} - x_k\|_2 \right]$$

$$\leq 2\alpha_k \cdot \mathbb{E}_k \left[ F(x) + r(x) - F(x_k) - r(\tilde{x}_{k+1}) \right]$$

$$+ 2\alpha_k \sqrt{\mathbb{E}_k \left[ \left( \frac{1}{m} \sum_{i=1}^{m} L_{f,U}(S_k^i) \right)^2 \right]} \cdot \sqrt{\mathbb{E}_k \left[ \|\tilde{x}_{k+1} - x_k\|_2^2 \right]}$$

$$\leq 2\alpha_k \cdot \mathbb{E}_k \left[ F(x) + r(x) - F(x_k) - r(\tilde{x}_{k+1}) \right]$$

$$+ 2\alpha_k \sqrt{\frac{1 + (m-1)\eta_{f,U}}{m}} L_F(U) \cdot t_k,$$

(B.24)

where the first inequality comes from Assumption 3 and the fact that $x_k, \tilde{x}_{k+1} \in U$, the second is due to the Cauchy-Schwarz inequality and the last follows from Assumption 3 and Lemma B.5. By Jensen's inequality, the following assertion holds:

$$\mathbb{E}_k \left[ \|\tilde{x}_{k+1} - x_k\|_2 \right] \leq t_k. \tag{B.25}$$

Using inequality (B.24), we can deduce that

$$\mathbb{E}_k \left[ \|x - \tilde{x}_{k+1}\|_2^2 \right] + t_k^2 - \|x - x_k\|_2^2$$

$$\leq 2\alpha_k \cdot \mathbb{E}_k \left[ F(x) + r(x) - (F(\tilde{x}_{k+1}) - L_F(U)\|\tilde{x}_{k+1} - x_k\|_2) - r(\tilde{x}_{k+1}) \right]$$

$$+ 2\alpha_k \sqrt{\frac{1 + (m-1)\eta_{f,U}}{m}} L_F(U) t_k$$

$$\leq 2\alpha_k \cdot \mathbb{E}_k \left[ F(x) + r(x) - F(\tilde{x}_{k+1}) - r(\tilde{x}_{k+1}) \right] + 2\sqrt{\rho_{f,m,U}} L_F(U) \cdot \alpha_k t_k,$$

where the first inequality uses Lemma B.1(ii) and the fact that $x_k, \tilde{x}_{k+1} \in U$, the second follows from (B.25) and the definition of $\rho_{f,m,U}$. Rearranging the previous result, one can conclude that

$$\mathbb{E}_k \left[ \|x - \tilde{x}_{k+1}\|_2^2 \right] \leq \|x - x_k\|_2^2 - 2\alpha_k \cdot \mathbb{E}_k \left[ \phi(\tilde{x}_{k+1}) - \phi(x) \right]$$

$$+ 2\sqrt{\rho_{f,m,U}} L_F(U) \cdot \alpha_k t_k - t_k^2$$

$$\leq \|x - x_k\|_2^2 - 2\alpha_k \cdot \mathbb{E}_k \left[ \phi(\tilde{x}_{k+1}) - \phi(x) \right]$$

$$+ \max_{t \in \mathbb{R}_+} \left\{ 2\sqrt{\rho_{f,m,U}} L_F(U) \cdot \alpha_k t - t^2 \right\}$$

$$= \|x - x_k\|_2^2 - 2\alpha_k \cdot \mathbb{E}_k \left[ \phi(\tilde{x}_{k+1}) - \phi(x) \right] + \rho_{f,m,U} L_F(U)^2 \cdot \alpha_k^2,$$

which completes the proof of (B.2).

(ii) Inequality (B.3) follows directly from the combination of (B.2) derived in (i) and (B.7) mentioned before.

This completes the proof of Lemma B.3. □

### B.3.4 PROOF OF LEMMA B.5

The first equality is easily established as follows:

$$\mathbb{E} \left[ \left( \frac{1}{m} \sum_{i=1}^{m} X_i \right)^2 \right] = \text{Var} \left( \frac{1}{m} \sum_{i=1}^{m} X_i \right) + \left( \mathbb{E} \left[ \frac{1}{m} \sum_{i=1}^{m} X_i \right] \right)^2$$

$$= \frac{1}{m^2} \text{Var} \left( \sum_{i=1}^{m} X_i \right) + \left( \frac{1}{m} \mathbb{E} \sum_{i=1}^{m} [X_i] \right)^2$$

$$= \frac{1}{m^2} \sum_{i=1}^{m} \text{Var} (X_i) + \left( \frac{1}{m} \mathbb{E} \sum_{i=1}^{m} [X_i] \right)^2 = \frac{1}{m} \sigma^2 + \mu^2,$$

where the second last equality uses the fact that the random variables $X_1, \cdots, X_m$ are independent. Note that $\sigma^2 + \mu^2 = \text{Var}(X_1) + (\mathbb{E}[X_1])^2 = \mathbb{E}[(X_1)^2] > 0$. Then

$$\eta = 1 - \frac{\text{Var}(X_1)}{\mathbb{E}[(X_1)^2]} = 1 - \frac{\sigma^2}{\mathbb{E}[(X_1)^2]} = \frac{\mu^2}{\mathbb{E}[(X_1)^2]}$$

and

$$\frac{1}{m}\sigma^2 + \mu^2 = \frac{1}{m}(1-\eta)\,\mathbb{E}[(X_1)^2] + \eta\mathbb{E}[(X_1)^2] = \frac{1 + (m-1)\eta}{m}\mathbb{E}[(X_1)^2],$$

which proves the second equality and completes the proof of Lemma B.5.

## C    PROOFS FOR THE RESULTS IN SECTION 2.2

In this section, we establish the convergence rate guarantees for isPPA in terms of the expected squared distance to the optimal solution set. Recall that function $\varsigma_\beta$ is defined in (2.3) as follows:

$$\varsigma_\beta(t) \triangleq \begin{cases} \frac{t^\beta - 1}{\beta} & \text{if } \beta \neq 0, \\ \ln(t) & \text{if } \beta = 0, \end{cases}$$

for all $t \in \mathbb{R}_{++}$. To prove the convergence rate results presented in Section 2.2, we need the following three technical lemmas, the proofs of which are deferred to Appendix C.4 for brevity.

**Lemma C.1.** *Consider the scalar sequence $\{\delta_k\}$ satisfying the recursive relation:*

$$\delta_{k+1} \leq a_1 \delta_k + a_2 \quad \text{for all } k \in \mathbb{Z}_+$$

*for some constants $a_1 \in (0, 1)$ and $a_2 \in \mathbb{R}_+$. Then it holds that:*

$$\delta_k \leq a_1^{k-1} \delta_1 + a_2(1-a_1)^{-1} \quad \text{for all } k \in \mathbb{Z}_+.$$

**Lemma C.2.** *Consider the scalar sequence $\{\delta_k\}$ satisfying the recursive relation:*

$$\delta_{k+1} \leq \exp(-a_1\alpha_k)\delta_k + a_2\alpha_k^2 \quad \text{for all } k \in \mathbb{Z}_+$$

*with $\alpha_k = \alpha_0 k^{-\beta}$ for some constants $\beta \in (0,1)$, $\alpha_0 \in \mathbb{R}_{++}$, $a_1 \in \mathbb{R}_{++}$ and $a_2 \in \mathbb{R}_+$. Then it holds that*

$$\delta_k \leq \exp\left(-a_1\alpha_0\varsigma_{1-\beta}(k)\right)\delta_1$$
$$+ \frac{4^\beta a_2\alpha_0^2}{1 - \exp(-a_1\alpha_0)} \cdot \frac{1}{k^\beta} + 4^\beta a_2\alpha_0^2\exp\left(-\frac{a_1\alpha_0}{2}k^{1-\beta}\right)\cdot\varsigma_{1-2\beta}\left(\frac{k}{2}\right)$$

*for all $k \in \mathbb{Z}_+$.*

**Lemma C.3.** *Consider the scalar sequence $\{\delta_k\}$ satisfying the recursive relation:*

$$\delta_{k+1} \leq \frac{1}{1 + a_1\alpha_k}\delta_k + a_2\frac{\alpha_k^2}{1 + a_1\alpha_k} \quad \text{for all } k \in \mathbb{Z}_+$$

*with $\alpha_k = \alpha_0 k^{-1}$ for some constants $\alpha_0 \in \mathbb{R}_{++}$, $a_1 \in \mathbb{R}_{++}$ and $a_2 \in \mathbb{R}_+$. Then it holds that*

$$\delta_k \leq \begin{cases} \left[\frac{4\delta_1}{1+a_1\alpha_0} + \frac{4a_2(2+a_1\alpha_0)\alpha_0^2}{2-a_1\alpha_0}\right]\cdot\left(\frac{1}{k}\right)^{\frac{2a_1\alpha_0}{2+a_1\alpha_0}} & \text{if } a_1\alpha_0 < 2, \\ \frac{4\delta_1}{3}\cdot\frac{1}{k} + 4a_2\alpha_0^2\cdot\frac{\ln(k)}{k} & \text{if } a_1\alpha_0 = 2, \\ \frac{4\delta_1}{1+a_1\alpha_0}\cdot\left(\frac{1}{k}\right)^{\frac{2a_1\alpha_0}{2+a_1\alpha_0}} + \frac{4a_2(2+a_1\alpha_0)\alpha_0^2}{a_1\alpha_0-2}\cdot\frac{1}{k} & \text{if } a_1\alpha_0 > 2. \end{cases}$$

Let $s \in (0,2)$ be an arbitrary scalar. Recall that the term $\tilde{C}_{f,m,U,c_1,\gamma}(\alpha_0)$ is defined as:

$$\tilde{C}_{f,m,U,c_1,\gamma}(\alpha_0) \triangleq \rho_{f,m,U}L_F(U)^2 + \frac{(1 + 2c_1\alpha_0)(1 + sc_1\alpha_0)}{(2-s)c_1}\gamma^2 \quad \text{for } \alpha_0 \in \mathbb{R}_{++}.$$

As mentioned in Section 2.2, if exists, $U$ denotes a bounded open subset of $\text{dom}(r)$ which contains the sequences $\{x_k\}$ and $\{\tilde{x}_k\}$.

## C.1 PROOF OF THEOREM 2.3

On the event that $\sup_{k\in\mathbb{Z}_+}\|x_k\|_2 < \infty$, similar to the proof of Theorem 2.2, we utilize the Cauchy-Schwarz inequality combined with criterion (SCA) and the condition $\epsilon_k = \gamma\alpha_0^{\frac{3}{2}}$ to show that $\sup_{k\in\mathbb{Z}_+}\|\tilde{x}_k\|_2 < \infty$. Consequently, there exists a bounded open subset of $\mathrm{dom}(r)$ containing the sequences $\{x_k\}$ and $\{\tilde{x}_k\}$. We denote this subset by $U$ by default. Let $\delta_k \triangleq \mathbb{E}[\mathrm{dist}(x_k,\mathcal{X}^*)^2]$ for each $k \in \mathbb{Z}_+$. The proof of Theorem 2.3 proceeds by establishing the validity of the following deterministic recursive relation:

**Claim C.1.** *Let Assumption 1-4 hold, and let $\{x_k\}$ be generated by isPPA (Algorithm 1) with stepsizes $\{\alpha_k\}$ and accuracy parameters $\epsilon_k = \gamma\alpha_k^{\frac{3}{2}}$ for all $k \in \mathbb{Z}_+$ with $\gamma \in \mathbb{R}_+$. Suppose that the sequences $\{x_k\}$ and $\{\tilde{x}_k\}$ are contained in some bounded open set $U \subset \mathrm{dom}(r)$. Then for all $k \in \mathbb{Z}_+$, we have*

$$\delta_{k+1} \le \frac{1}{1+sc_1\alpha_k}\delta_k + \tilde{C}_{f,m,U,c_1,\gamma}(\alpha_k) \cdot \frac{\alpha_k^2}{1+sc_1\alpha_k} \quad \text{for all } k \in \mathbb{Z}_+.$$

Assuming that the above assertion holds, we immediately have

$$\delta_{k+1} \le \frac{1}{1+sc_1\alpha_0}\delta_k + \tilde{C}_{f,m,U,c_1,\gamma}(\alpha_0) \cdot \frac{\alpha_0^2}{1+sc_1\alpha_0} \quad \text{for all } k \in \mathbb{Z}_+.$$

Taking $a_1 = (1+sc_1\alpha_0)^{-1}$ and $a_2 = \tilde{C}_{f,m,U,c_1,\gamma}(\alpha_0) \cdot \frac{\alpha_0^2}{1+sc_1\alpha_0}$ in Lemma C.1 leads to inequality (2.8) in Theorem 2.3. The only remaining task is to prove Claim C.1. For brevity, its proof is deferred to Appendix C.2.1.

## C.2 PROOF OF THEOREM 2.4

Use the same notations as in the proof of Theorem 2.3 and set $\delta_k \triangleq \mathbb{E}[\mathrm{dist}(x_k,\mathcal{X}^*)^2]$ for each $k \in \mathbb{Z}_+$. Now we prove the results for isPPA with diminishing stepsizes $\alpha_k = \alpha_0 k^{-\beta}$ where $\beta \in (0,1]$. Then

$$0 < \alpha_k \le \alpha_{k+1} \le \alpha_0 \quad \text{for all } k \in \mathbb{Z}_+. \tag{C.1}$$

On the event that $\sup_{k\in\mathbb{Z}_+}\|x_k\|_2 < \infty$, similar to the proof of Theorem 2.2, the combination of criterion (SCA), the Cauchy-Schwarz inequality, and the condition $\epsilon_k = \gamma\alpha_k^{\frac{3}{2}} \le \gamma\alpha_0^{\frac{3}{2}}$ leads to the conclusion that $\sup_{k\in\mathbb{Z}_+}\|\tilde{x}_k\|_2 < \infty$. We can thus use $U$ to denote a bounded open subset of $\mathrm{dom}(r)$ containing the sequences $\{x_k\}$ and $\{\tilde{x}_k\}$. This confirms that all the assumptions required in Claim C.1 are satisfied. Fix any $k \in \mathbb{Z}_+$. Note that $C_{f,m,U,c_1,\gamma}(\cdot)$ is nondecreasing on $\mathbb{R}_+$ and thus by (C.1), we have

$$\tilde{C}_{f,m,U,c_1,\gamma}(\alpha_k) \le \tilde{C}_{f,m,U,c_1,\gamma}(\alpha_0). \tag{C.2}$$

Then, it follows from Claim C.1 that

$$\begin{aligned}
\delta_{k+1} &\le \frac{1}{1+sc_1\alpha_k}\delta_k + \tilde{C}_{f,m,U,c_1,\gamma}(\alpha_k) \cdot \frac{\alpha_k^2}{1+sc_1\alpha_k} \\
&\le \frac{1}{1+sc_1\alpha_k}\delta_k + \tilde{C}_{f,m,U,c_1,\gamma}(\alpha_0) \cdot \frac{\alpha_k^2}{1+sc_1\alpha_k} \tag{C.3a}\\
&= \left(1 - \frac{sc_1\alpha_k}{1+sc_1\alpha_k}\right)\delta_k + \tilde{C}_{f,m,U,c_1,\gamma}(\alpha_0) \cdot \frac{\alpha_k^2}{1+sc_1\alpha_k} \\
&\le \exp\left(-\frac{sc_1\alpha_k}{1+sc_1\alpha_k}\right)\delta_k + \tilde{C}_{f,m,U,c_1,\gamma}(\alpha_0) \cdot \frac{\alpha_k^2}{1+sc_1\alpha_k} \\
&\le \exp\left(-\frac{sc_1\alpha_k}{1+sc_1\alpha_0}\right)\delta_k + \tilde{C}_{f,m,U,c_1,\gamma}(\alpha_0) \cdot \alpha_k^2, \tag{C.3b}
\end{aligned}$$

where the second inequality uses the relation (C.2), the third follows from (B.11) and the last holds because of inequality (C.1).

**Proof of assertion (i).** If $\beta \in (0, 1)$, by taking $a_1 = sc_1(1 + sc_1\alpha_0)^{-1}$ and $a_2 = \tilde{C}_{f,m,U,c_1,\gamma}(\alpha_0)$ in Lemma C.2, we can easily obtain the estimate (2.9) from inequality (C.3b), thus confirming assertion (i) of Theorem 2.4.

**Proof of assertion (ii).** As for the case where $\beta = 1$, the estimate (2.10) follows from (C.3a) and Lemma C.3 with $a_1 = sc_1$ and $a_2 = \tilde{C}_{f,m,U,c_1,\gamma}$. This verifies assertion (ii) and thus completes the proof of Theorem 2.4.

### C.2.1 Proof of Claim C.1

It only remains to prove Claim C.1. We present the detailed proof as follows: Fix any $k \in \mathbb{Z}_+$. Setting $x = \text{proj}(\tilde{x}_{k+1}, \mathcal{X}^*)$ in inequality (B.7) yields

$$
\begin{aligned}
\text{dist}\,(x_{k+1}, \mathcal{X}^*)^2 &\leq \|x_{k+1} - \text{proj}\,(\tilde{x}_{k+1}, \mathcal{X}^*)\|_2^2 \\
&\leq \left(1 + \frac{1}{t}\right) \|\tilde{x}_{k+1} - \text{proj}\,(\tilde{x}_{k+1}, \mathcal{X}^*)\|_2^2 + (1 + t)\,\epsilon_k^2 \\
&= \left(1 + \frac{1}{t}\right) \text{dist}\,(\tilde{x}_{k+1}, \mathcal{X}^*)^2 + (1 + t)\,\epsilon_k^2,
\end{aligned}
$$

and thus

$$
\mathbb{E}_k\left[\text{dist}\,(x_{k+1}, \mathcal{X}^*)^2\right] \leq \left(1 + \frac{1}{t}\right) \mathbb{E}_k\left[\text{dist}\,(\tilde{x}_{k+1}, \mathcal{X}^*)^2\right] + (1 + t)\,\epsilon_k^2. \tag{C.4}
$$

Note that the sequences $\{x_k\}$ and $\{\tilde{x}_k\}$ are contained in $U$. By setting $\overline{x} = \text{proj}(x_k, \mathcal{X}^*)$ in inequality (B.2) of Lemma B.3, one can conclude that

$$
\begin{aligned}
\mathbb{E}_k\left[\text{dist}\,(\tilde{x}_{k+1}, \mathcal{X}^*)^2\right] &\leq \mathbb{E}_k\left[\|\tilde{x}_{k+1} - \text{proj}(x_k, \mathcal{X}^*)\|_2^2\right] \\
&\leq \|x_k - \text{proj}(x_k, \mathcal{X}^*)\|_2^2 - 2\alpha_k \cdot \mathbb{E}_k\left[\phi\,(\tilde{x}_{k+1}) - \phi^*\right] + \rho_{f,m,U} L_F(U)^2 \cdot \alpha_k^2 \\
&\leq \|x_k - \text{proj}(x_k, \mathcal{X}^*)\|_2^2 - 2c_1\alpha_k \cdot \mathbb{E}_k\left[\text{dist}(\tilde{x}_{k+1}, \mathcal{X}^*)^2\right] + \rho_{f,m,U} L_F(U)^2 \cdot \alpha_k^2 \\
&= \text{dist}\,(x_k, \mathcal{X}^*)^2 - 2c_1\alpha_k \cdot \mathbb{E}_k\left[\text{dist}(\tilde{x}_{k+1}, \mathcal{X}^*)^2\right] + \rho_{f,m,U} L_F(U)^2 \cdot \alpha_k^2,
\end{aligned}
$$

where the last inequality comes from the quadratic growth condition in Assumption 4. Rearranging the preceding inequality yields

$$
\mathbb{E}_k\left[\text{dist}\,(\tilde{x}_{k+1}, \mathcal{X}^*)^2\right] \leq \frac{1}{1 + 2c_1\alpha_k} \text{dist}\,(x_k, \mathcal{X}^*)^2 + \rho_{f,m,U} L_F(U)^2 \cdot \frac{\alpha_k^2}{1 + 2c_1\alpha_k}. \tag{C.5}
$$

Applying (C.5) to inequality (C.4) with $t = \frac{1 + sc_1\alpha_k}{(2-s)c_1\alpha_k}$ and $\epsilon_k = \gamma\alpha_k^{\frac{3}{2}}$, and using the fact that

$$
\left(1 + \frac{1}{t}\right) \cdot \frac{1}{1 + 2c_1\alpha_k} = \frac{1 + 2c_1\alpha_k}{1 + sc_1\alpha_k} \cdot \frac{1}{1 + 2c_1\alpha_k} = \frac{1}{1 + sc_1\alpha_k},
$$

$$
(1 + t) \cdot \alpha_k^3 = \frac{1 + 2c_1\alpha_k}{(2-s)c_1\alpha_k} \cdot \alpha_k^3 = \frac{(1 + 2c_1\alpha_k)(1 + sc_1\alpha_k)}{(2-s)c_1} \cdot \frac{\alpha_k^2}{1 + sc_1\alpha_k},
$$

we have

$$
\begin{aligned}
&\mathbb{E}_k\left[\text{dist}\,(x_{k+1}, \mathcal{X}^*)^2\right] \\
&\leq \frac{1}{1 + sc_1\alpha_k} \cdot \text{dist}\,(x_k, \mathcal{X}^*)^2 + \left[\rho_{f,m,U} L_F(U)^2 + \frac{(1 + 2c_1\alpha_k)(1 + sc_1\alpha_k)}{(2-s)c_1}\gamma^2\right] \cdot \frac{\alpha_k^2}{1 + sc_1\alpha_k}.
\end{aligned}
$$

Taking the expectation of both sides of the previous inequality, applying the law of total expectation and using the definition of $\tilde{C}_{f,m,U,c_1,\gamma}(\cdot)$ defined in (2.7), we can immediately derive Claim C.1.

### C.3 Proof of Corollary 2.5

For the case where $\beta = 1$, the dominant term of the bound (2.10) obtained in Theorem 2.4 can be readily determined. Thus, we only need to check the case where $\beta \in (0, 1)$. Note that the bound (2.9) given in Theorem 2.4 can be written as

$$
\mathbb{E}\left[\text{dist}(x_k, \mathcal{X}^*)^2\right] \leq C_1\exp(-A_1\varsigma_{1-\beta}(k)) + \frac{C_2}{k^\beta} + C_3\exp\left(-A_2 k^{1-\beta}\right) \cdot \varsigma_{1-2\beta}\left(\frac{k}{2}\right), \tag{C.6}
$$

where $C_i \in \mathbb{R}_+$ and $A_j \in \mathbb{R}_{++}$ for $i = 1, 2, 3$ and $j = 1, 2$. Since $\beta \in (0, 1)$, then

$$C_1 \exp(-A_1 \varsigma_{1-\beta}(k)) \leq \mathcal{O}\left(b_1^{k^{1-\beta}}\right) \leq \mathcal{O}\left(k^{-\beta}\right)$$

for sufficiently large $k$, where $b_1 \in (0, 1)$.

   (i) If $\beta \in (0, \frac{1}{2})$, then $\varsigma_{1-2\beta}(\frac{k}{2}) \leq \mathcal{O}\left(k^{1-2\beta}\right)$ by (C.7) (see Lemma C.4(i) below). Note that (B.11) implies $\exp(-t) \leq \frac{1}{1+t}$ for all $t \in \mathbb{R}_+$. Thus we have

$$C_3 \exp\left(-A_2 k^{1-\beta}\right) \cdot \varsigma_{1-2\beta}\left(\frac{k}{2}\right) \leq \mathcal{O}\left(k^{-(1-\beta)} \cdot k^{1-2\beta}\right) = \mathcal{O}\left(k^{-\beta}\right).$$

  (ii) If $\beta = \frac{1}{2}$, then

$$C_3 \exp\left(-A_2 k^{1-\beta}\right) \cdot \varsigma_{1-2\beta}\left(\frac{k}{2}\right) = \mathcal{O}\left(b_2^{\sqrt{k}} \ln(k)\right) \leq \mathcal{O}(k^{-\frac{1}{2}})$$

for sufficiently large $k$, where $b_2 \in (0, 1)$.

 (iii) If $\beta \in (\frac{1}{2}, 1)$, then $\varsigma_{1-2\beta}(\frac{k}{2}) \leq \frac{1}{2\beta-1}$ by (C.7) and thus

$$C_3 \exp\left(-A_2 k^{1-\beta}\right) \cdot \varsigma_{1-2\beta}\left(\frac{k}{2}\right) \leq \mathcal{O}(b_3^{k^{1-\beta}}) \leq \mathcal{O}\left(k^{-\beta}\right)$$

for sufficiently large $k$, where $b_3 \in (0, 1)$.

Combining the previous results with (C.6) verifies the estimates for the case where $\beta \in (0, 1)$, thereby completing the proof of Corollary 2.5.

**Remark C.1.** The convergence rate results in Theorem 2.3 and Theorem 2.4 provide insights into how the minibatch size $m$ affects the performance of isPPA. In general, increasing the minibatch size $m$ improves the convergence rate of isPPA. Specifically, from equation (2.4), we have

$$\eta_{f,U} \in [0, 1] \quad \text{and} \quad \rho_{f,m,U} = (\sqrt{1 + \frac{1 - \eta_{f,U}}{m}} + 1)^2,$$

indicating that $\rho_{f,m,U}$ is a nonincreasing function of $m$. Then, from the definition of $\tilde{C}_{f,m,U,c_1,\gamma}(\cdot)$ in (2.7), $\tilde{C}_{f,m,U,c_1,\gamma}(\alpha)$ decreases with $m$ for a fixed $\alpha$. This leads to the following observations:

- **Neighborhood shrinks when using constant stepsizes.** For isPPA with constant stepsizes, when increasing the minibatch size $m$, the linear convergence of $\{x_k\}$ to a neighborhood of the optimal solution set improves, as the neighborhood shrinks due to the noise term in (2.8) from Theorem 2.3 being proportional to $\tilde{C}_{f,m,U,c_1,\gamma}(\alpha_0)$.

- **Coefficient decreases when using diminishing stepsizes.** For isPPA with diminishing stepsizes, when increasing the minibatch size $m$, the sublinear convergence of $\{x_k\}$ improves in the sense that the coefficient of the dominant term decreases. For instance, as shown in the proof of Corollary 2.5, the dominant term in (2.9) from Theorem 2.4 is the second part $\mathcal{O}(k^{-\beta})$, whose coefficient is proportional to $\tilde{C}_{f,m,U,c_1,\gamma}(\alpha_0)$.

While a larger minibatch size $m$ improves the convergence rate of isPPA, it also increases the computational cost of solving the subproblem. Therefore, a trade-off must be considered when selecting $m$.

## C.4 PROOF OF AUXILIARY LEMMAS

It only remains to prove the auxiliary lemmas presented before. Note that the proofs of Lemma C.2 and Lemma C.3 leverage techniques similar to those employed in the proof of Theorem 1 in (Moulines & Bach, 2011). We first present the following lemma summarizing several useful inequalities to be subsequently employed.

**Lemma C.4.** *The following assertions hold:*

(i) *Let $\beta \in \mathbb{R}$. Then the function $\varsigma_\beta$ satisfies $\varsigma_\beta(1) = 0$,*

$$\varsigma_\beta(t) \leq \begin{cases} \frac{1}{-\beta} & \text{if } \beta < 0, \\ \frac{t^\beta}{\beta} & \text{if } \beta > 0. \end{cases} \tag{C.7}$$

*for all $t \in \mathbb{R}_{++}$, and is nondecreasing over $\mathbb{R}_{++}$.*

(ii) *The following inequality holds:*

$$\frac{k+1}{k} \leq 2 \quad \text{for all } k \in \mathbb{Z}_+. \tag{C.8}$$

*Let $\beta \in \mathbb{R}$ and $j, k \in \mathbb{N}$ with $k \geq j + 1$. Then*

$$\frac{1}{2^\beta} \cdot [\varsigma_{\beta+1}(k+1) - \varsigma_{\beta+1}(j+1)] \leq \sum_{i=j+1}^{k} i^\beta \leq \varsigma_{\beta+1}(k+1) - \varsigma_{\beta+1}(j+1)$$

$$\text{if } \beta \geq 0, \tag{C.9a}$$

$$\varsigma_{\beta+1}(k+1) - \varsigma_{\beta+1}(j+1) \leq \sum_{i=j+1}^{k} i^\beta \leq \frac{1}{2^\beta} \cdot [\varsigma_{\beta+1}(k+1) - \varsigma_{\beta+1}(j+1)]$$

$$\text{if } \beta < 0. \tag{C.9b}$$

*Moreover, if $j, k \in \mathbb{Z}_+$ with $k \geq j + 1$, then*

$$\varsigma_{\beta+1}(k) - \varsigma_{\beta+1}(j) \leq \sum_{i=j+1}^{k} i^\beta \leq (\beta + 2) \cdot [\varsigma_{\beta+1}(k) - \varsigma_{\beta+1}(j)] \quad \text{if } \beta > 0, \tag{C.10a}$$

$$\sum_{i=j+1}^{k} i^\beta \leq \varsigma_{\beta+1}(k) - \varsigma_{\beta+1}(j) \quad \text{if } \beta \leq 0. \tag{C.10b}$$

(iii) *Let $\beta \in (0, 1)$. Then the following assertions hold:*

$$(1+t)^\beta \leq 1 + \beta t \quad \text{for all } t \in [-1, \infty), \tag{C.11a}$$

$$\frac{1}{2}t^\beta \leq \varsigma_\beta(t) - \varsigma_\beta\left(\frac{t}{2}\right) \quad \text{for all } t \in \mathbb{R}_{++}. \tag{C.11b}$$

(iv) *Let $a \in \mathbb{R}_{++}$ and define $h(t) \triangleq \frac{t}{1-\exp(-at)}$ for all $t \in \mathbb{R}_{++}$. Then the function $h$ is nondecreasing over $\mathbb{R}_{++}$.*

An elementary proof of the previous lemma is deferred to Appendix C.4.4 for reference.

### C.4.1  PROOF OF LEMMA C.1

Fix any $k \in \mathbb{Z}_+$. Applying the recursion $k$ times yields that

$$\delta_{k+1} \leq a_1^k \delta_1 + a_2 \sum_{i=0}^{k-1} a_1^i = a_1^k \delta_1 + a_2 \cdot \frac{1 - a_1^k}{1 - a_1}$$

$$\leq a_1^k \delta_1 + a_2 (1 - a_1)^{-1}$$

where the equality is due to $a_1 \neq 1$, and the second inequality follows from the facts that $a_1 \in (0, 1)$ and $a_2 \in \mathbb{R}_+$. This completes the proof of Lemma C.1.

### C.4.2  PROOF OF LEMMA C.2

Fix any $k \in \mathbb{Z}_+$. Applying the recursion $k$ times yields that

$$\delta_{k+1} \leq \exp\left(-a_1 \sum_{i=1}^{k} \alpha_i\right) \delta_1 + a_2 \sum_{i=1}^{k} \alpha_i^2 \exp\left(-a_1 \sum_{j=i+1}^{k} \alpha_j\right). \tag{rBound}$$

Define

$$I_1 \triangleq \exp\left(-a_1 \sum_{i=1}^{k} \alpha_i\right) \quad \text{and} \quad I_2 \triangleq \sum_{i=1}^{k} \alpha_i^2 \exp\left(-a_1 \sum_{j=i+1}^{k} \alpha_j\right). \tag{defTerm}$$

Note that $\alpha_k = \alpha_0 k^{-\beta}$ for all $k \in \mathbb{Z}_+$ with $\beta \in (0,1)$. Then it follows from (C.9b) with $j = 0$ that

$$I_1 = \exp\left(-a_1 \alpha_0 \sum_{i=1}^{k} i^{-\beta}\right) \leq \exp\left(-a_1 \alpha_0 \left[\varsigma_{1-\beta}(k+1) - \varsigma_{1-\beta}(1)\right]\right) \tag{I1Bound}$$

$$= \exp\left(-a_1 \alpha_0 \cdot \varsigma_{1-\beta}(k+1)\right),$$

where the last equality uses the fact that $\varsigma_{1-\beta}(1) = 0$ implied by Lemma C.4(i). It only remains to obtain an upper bound on $I_2$. Set $m \triangleq \lfloor \frac{k-1}{2} \rfloor$. We can easily verify that

$$\frac{k}{2} \leq m + 1 \leq \frac{k+1}{2}. \tag{C.12}$$

For simplicity, we denote by $\sum_{i=j}^{k} \cdot \triangleq 0$ for $j, k \in \mathbb{Z}_+$ with $j > k$. Then by using the definition of $\alpha_k$, the term $I_2$ can be written by

$$I_2 = \sum_{i=1}^{m} \alpha_i^2 \exp\left(-a_1 \alpha_0 \sum_{j=i+1}^{k} j^{-\beta}\right) + \sum_{i=m+1}^{k} \alpha_i^2 \exp\left(-a_1 \sum_{j=i+1}^{k} \alpha_j\right) \triangleq I_2^{(1)} + I_2^{(2)}.$$

By assumption, we have $\beta \in (0,1)$. Then the term $I_2^{(1)}$ can be bounded by

$$I_2^{(1)} \leq \sum_{i=1}^{m} \alpha_i^2 \exp\left(-a_1 \alpha_0 \left(\varsigma_{1-\beta}(k+1) - \varsigma_{1-\beta}(i+1)\right)\right)$$

$$\leq \exp\left(-a_1 \alpha_0 \left(\varsigma_{1-\beta}(k+1) - \varsigma_{1-\beta}(m+1)\right)\right) \sum_{i=1}^{m} \alpha_i^2$$

$$\leq \exp\left(-a_1 \alpha_0 \left(\varsigma_{1-\beta}(k+1) - \varsigma_{1-\beta}\left(\frac{k+1}{2}\right)\right)\right) \sum_{i=1}^{m} \alpha_i^2$$

$$\leq \exp\left(-\frac{a_1 \alpha_0}{2}(k+1)^{1-\beta}\right) \sum_{i=1}^{m} \alpha_i^2$$

$$\leq 4^{\beta} \alpha_0^2 \exp\left(-\frac{a_1 \alpha_0}{2}(k+1)^{1-\beta}\right) \cdot \left[\varsigma_{1-2\beta}(m+1) - \varsigma_{1-2\beta}(1)\right]$$

$$\leq 4^{\beta} \alpha_0^2 \exp\left(-\frac{a_1 \alpha_0}{2}(k+1)^{1-\beta}\right) \cdot \varsigma_{1-2\beta}\left(\frac{k+1}{2}\right),$$

where the first and second last inequalities follow from (C.9b), the second is due to Lemma C.4(i), the third and last are due to Lemma C.4(i) and inequality (C.12), and the fourth comes from (C.11b). For the term $I_2^{(2)}$, we obtain the following bound:

$$I_2^{(2)} = \sum_{i=m+1}^{k} \alpha_i^2 \exp\left(-a_1 \sum_{j=i+1}^{k} \alpha_j\right)$$

$$= \sum_{i=m+1}^{k} \frac{\alpha_i^2}{1 - \exp(-a_1 \alpha_i)} \cdot \left[\exp\left(-a_1 \sum_{j=i+1}^{k} \alpha_j\right) - \exp\left(-a_1 \sum_{j=i}^{k} \alpha_j\right)\right]$$

$$\leq \alpha_{m+1} \cdot \sum_{i=m+1}^{k} \frac{\alpha_i}{1 - \exp(-a_1 \alpha_i)} \cdot \left[\exp\left(-a_1 \sum_{j=i+1}^{k} \alpha_j\right) - \exp\left(-a_1 \sum_{j=i}^{k} \alpha_j\right)\right]$$

$$\leq \alpha_{m+1} \frac{\alpha_0}{1 - \exp(-a_1 \alpha_0)} \cdot \sum_{i=m+1}^{k} \left[\exp\left(-a_1 \sum_{j=i+1}^{k} \alpha_j\right) - \exp\left(-a_1 \sum_{j=i}^{k} \alpha_j\right)\right]$$

$$= \alpha_{m+1} \frac{\alpha_0}{1 - \exp\left(-a_1\alpha_0\right)} \cdot \left[ 1 - \exp\left( -a_1 \sum_{j=m+1}^{k} \alpha_j \right) \right]$$

$$\leq \alpha_{m+1} \frac{\alpha_0}{1 - \exp\left(-a_1\alpha_0\right)} = \frac{\alpha_0^2}{1 - \exp\left(-a_1\alpha_0\right)}(m+1)^{-\beta}$$

$$\leq \frac{\alpha_0^2}{1 - \exp\left(-a_1\alpha_0\right)} \left(\frac{k}{2}\right)^{-\beta} = \frac{\alpha_0^2}{1 - \exp\left(-a_1\alpha_0\right)} \left(\frac{k+1}{2}\right)^{-\beta} \cdot \left(\frac{k}{k+1}\right)^{-\beta}$$

$$\leq \frac{\alpha_0^2}{1 - \exp\left(-a_1\alpha_0\right)} \left(\frac{k+1}{4}\right)^{-\beta},$$

where the first inequality is due to (C.1), the second follows from Lemma C.4(iv) and (C.1), the second last comes from (C.12), and the last holds by (C.8). Thus,

$$I_2 \leq 4^\beta \alpha_0^2 \exp\left( -\frac{a_1\alpha_0}{2}(k+1)^{1-\beta} \right) \cdot \varsigma_{1-2\beta} \left( \frac{k+1}{2} \right)$$
$$+ \frac{4^\beta \alpha_0^2}{1 - \exp\left(-a_1\alpha_0\right)} \cdot (k+1)^{-\beta}. \tag{I2Bound-l1}$$

Combining (rBound) with (defTerm), (I1Bound) and (I2Bound-l1) proves Lemma C.2.

### C.4.3 PROOF OF LEMMA C.3

The proof of Lemma C.3 differs from the proof of Lemma C.2 because now $\beta = 1$. Applying the recursion $k$ times yields that

$$\delta_{k+1} \leq \left( \prod_{i=1}^{k} \frac{1}{1 + a_1\alpha_i} \right) \delta_1 + a_2 \sum_{i=1}^{k} \frac{\alpha_i^2}{1 + a_1\alpha_i} \left( \prod_{j=i+1}^{k} \frac{1}{1 + a_1\alpha_j} \right). \tag{rBound$'$}$$

Define

$$I_1 \triangleq \prod_{i=1}^{k} \frac{1}{1 + a_1\alpha_i} \quad \text{and} \quad I_2 \triangleq \sum_{i=1}^{k} \frac{\alpha_i^2}{1 + a_1\alpha_i} \left( \prod_{j=i+1}^{k} \frac{1}{1 + a_1\alpha_j} \right). \tag{defTerm$'$}$$

Note that $\alpha_k = \alpha_0 k^{-1}$ for all $k \in \mathbb{Z}_+$ and that

$$a_1\alpha_0 \leq \frac{1}{2}a_1\alpha_0 i \quad \text{for all } i \in \mathbb{Z}_+ \text{ and } i \geq 2. \tag{C.13}$$

With the previous observation, we can establish the following bound on $I_1$:

$$\frac{1}{1 + a_1\alpha_i} = \frac{1}{1 + a_1\alpha_0 i^{-1}} = \frac{i}{i + a_1\alpha_0} = 1 - \frac{a_1\alpha_0}{i + a_1\alpha_0}$$
$$\leq 1 - \frac{2a_1\alpha_0}{2i + a_1\alpha_0 i} = 1 - \frac{2a_1\alpha_0}{2 + a_1\alpha_0} i^{-1} \tag{C.14}$$
$$\leq \exp\left( -\frac{2a_1\alpha_0}{2 + a_1\alpha_0} i^{-1} \right) \quad \text{for all } i \in \mathbb{Z}_+ \text{ and } i \geq 2,$$

where the first inequality follows from (C.13) and the second is due to (B.11). Using (C.14) above, we can derive that

$$I_1 = \frac{1}{1 + a_1\alpha_0} \left( \prod_{i=2}^{k} \frac{1}{1 + a_1\alpha_i} \right) \leq \frac{1}{1 + a_1\alpha_0} \exp\left( -\frac{2a_1\alpha_0}{2 + a_1\alpha_0} \sum_{i=2}^{k} i^{-1} \right)$$

$$\leq \frac{1}{1 + a_1\alpha_0} \exp\left( -\frac{2a_1\alpha_0}{2 + a_1\alpha_0} [\varsigma_0(k+1) - \varsigma_0(2)] \right)$$

$$= \frac{1}{1 + a_1\alpha_0} \exp\left( -\frac{2a_1\alpha_0}{2 + a_1\alpha_0} \ln\left(\frac{k+1}{2}\right) \right) = \frac{1}{1 + a_1\alpha_0} \left(\frac{2}{k+1}\right)^{\frac{2a_1\alpha_0}{2 + a_1\alpha_0}} \tag{I1Bound$'$}$$

$$\leq \frac{4}{1 + 2a_1\alpha_0} \left(\frac{1}{k+1}\right)^{\frac{2a_1\alpha_0}{2 + a_1\alpha_0}},$$

where the second inequality is due to (C.9b) with $j = 1$, and the last uses the fact that

$$\frac{2a_1\alpha_0}{2 + a_1\alpha_0} \le 2. \tag{C.15}$$

It remains to obtain an upper bound on $I_2$. By using (C.14) again, one can easily verify that

$$
\begin{aligned}
\prod_{j=i+1}^{k} \frac{1}{1 + a_1\alpha_j} &\le \exp\left(-\frac{2a_1\alpha_0}{2 + a_1\alpha_0} \sum_{j=i+1}^{k} j^{-1}\right) \\
&\le \exp\left(-\frac{2a_1\alpha_0}{2 + a_1\alpha_0} \left[\varsigma_0(k+1) - \varsigma_0(i+1)\right]\right) \\
&= \exp\left(-\frac{2a_1\alpha_0}{2 + a_1\alpha_0} \ln\left(\frac{k+1}{i+1}\right)\right) = \left(\frac{i+1}{k+1}\right)^{\frac{2a_1\alpha_0}{2+a_1\alpha_0}}
\end{aligned} \tag{C.16}
$$

for $i = 1, \cdots, k$,

where the second inequality comes from (C.9b) with $j = i$. Then

$$
\begin{aligned}
I_2 = \sum_{i=1}^{k} \frac{\alpha_i^2}{1 + a_1\alpha_i} \left(\prod_{j=i+1}^{k} \frac{1}{1 + a_1\alpha_j}\right) &\le \sum_{i=1}^{k} \alpha_i^2 \left(\frac{i+1}{k+1}\right)^{\frac{2a_1\alpha_0}{2+a_1\alpha_0}} \\
&= \alpha_0^2 (k+1)^{-\frac{2a_1\alpha_0}{2+a_1\alpha_0}} \sum_{i=1}^{k} i^{-2} (i+1)^{\frac{2a_1\alpha_0}{2+a_1\alpha_0}} \\
&= \alpha_0^2 (k+1)^{-\frac{2a_1\alpha_0}{2+a_1\alpha_0}} \sum_{i=1}^{k} (i+1)^{\frac{2a_1\alpha_0}{2+a_1\alpha_0}-2} \left(\frac{i+1}{i}\right)^2 \\
&\le 4\alpha_0^2 (k+1)^{-\frac{2a_1\alpha_0}{2+a_1\alpha_0}} \sum_{i=1}^{k} (i+1)^{\frac{2a_1\alpha_0}{2+a_1\alpha_0}-2} = 4\alpha_0^2 (k+1)^{-\frac{2a_1\alpha_0}{2+a_1\alpha_0}} \sum_{i=2}^{k+1} i^{\frac{2a_1\alpha_0}{2+a_1\alpha_0}-2} \\
&\le 4\alpha_0^2 (k+1)^{-\frac{2a_1\alpha_0}{2+a_1\alpha_0}} \left[\varsigma_{\frac{2a_1\alpha_0}{2+a_1\alpha_0}-1}(k+1) - \varsigma_{\frac{2a_1\alpha_0}{2+a_1\alpha_0}-1}(1)\right] \\
&= 4\alpha_0^2 (k+1)^{-\frac{2a_1\alpha_0}{2+a_1\alpha_0}} \varsigma_{\frac{2a_1\alpha_0}{2+a_1\alpha_0}-1}(k+1)
\end{aligned} \tag{C.17}
$$

where the first inequality follows from (C.16), the second last is due to (C.8), and the last comes from (C.15) and (C.10b) with $j = 1$. Using the previous inequality and the fact that

$$
\begin{aligned}
\frac{2a_1\alpha_0}{2 + a_1\alpha_0} &< 1 \quad \Leftrightarrow \quad a_1\alpha_0 < 2, \\
\frac{2a_1\alpha_0}{2 + a_1\alpha_0} &= 1 \quad \Leftrightarrow \quad a_1\alpha_0 = 2, \\
\frac{2a_1\alpha_0}{2 + a_1\alpha_0} &> 1 \quad \Leftrightarrow \quad a_1\alpha_0 > 2,
\end{aligned} \tag{C.18}
$$

we can deduce from the relation (C.7) in Lemma C.4(i) and the definition of function $\varsigma_{\frac{2a_1\alpha_0}{2+a_1\alpha_0}}(\cdot)$ given in (2.3) that the following assertion holds:

$$
I_2 \le \begin{cases}
\frac{4(2+a_1\alpha_0)\alpha_0^2}{2-a_1\alpha_0} \cdot \left(\frac{1}{k+1}\right)^{\frac{2a_1\alpha_0}{2+a_1\alpha_0}} & \text{if } a_1\alpha_0 < 2, \\
4\alpha_0^2 \frac{\ln(k+1)}{k+1} & \text{if } a_1\alpha_0 = 2, \\
\frac{4(2+a_1\alpha_0)\alpha_0^2}{a_1\alpha_0-2} \cdot \frac{1}{k+1} & \text{if } a_1\alpha_0 > 2.
\end{cases} \tag{I2Bound-e1$'$}
$$

Combining (rBound$'$) with (defTerm$'$), (I1Bound$'$) and (I2Bound-e1$'$) proves Lemma C.3.

### C.4.4 PROOF OF LEMMA C.4

By using the definition of the function $\varsigma_\beta$ given in (2.3), Lemma C.4 can be derived from elementary calculations, as shown below:

**Proof of assertion (i).** Fix any $\beta \in \mathbb{R}$ and $t \in \mathbb{R}_{++}$. Using the definition of $\varsigma_\beta$ given in (2.3), we can easily verify that $\varsigma_\beta(1) = 0$ and that

$$
\varsigma_\beta(t) = \begin{cases} \frac{t^\beta - 1}{\beta} = \frac{1 - t^\beta}{-\beta} \le \frac{1}{-\beta} & \text{if } \beta < 0, \\ \frac{t^\beta - 1}{\beta} \le \frac{t^\beta}{\beta} & \text{if } \beta > 0. \end{cases}
$$

Note that the derivative $\varsigma'_\beta(t) = t^{\beta-1} > 0$ for all $t \in \mathbb{R}_{++}$. Thus, the function $\varsigma_\beta(\cdot)$ is nondecreasing over $\mathbb{R}_{++}$.

**Proof of assertion (ii).** Note that (C.8) holds since $1 + \frac{1}{k} \le 2$ for all $k \in \mathbb{Z}_+$. Let $j, k \in \mathbb{N}$ such that $k \ge j + 1$.

- Suppose that $\beta \ge 0$. Then

$$
\sum_{i=j+1}^{k} i^\beta = \sum_{i=j+1}^{k} (i+1)^\beta \cdot \left(\frac{i}{i+1}\right)^\beta \ge \frac{1}{2^\beta} \sum_{i=j+1}^{k} (i+1)^\beta
$$

$$
= \frac{1}{2^\beta} \sum_{i=j+1}^{k} \int_i^{i+1} (i+1)^\beta \mathrm{d}t \ge \frac{1}{2^\beta} \sum_{i=j+1}^{k} \int_i^{i+1} t^\beta \mathrm{d}t = \frac{1}{2^\beta} \int_{j+1}^{k+1} t^\beta \mathrm{d}t
$$

$$
= \frac{1}{2^\beta} \left[\varsigma_{\beta+1}(k+1) - \varsigma_{\beta+1}(j+1)\right],
$$

where the first inequality follows from (C.8) and the fact that $\beta \ge 0$. Similarly,

$$
\sum_{i=j+1}^{k} i^\beta = \sum_{i=j+1}^{k} \int_i^{i+1} i^\beta \mathrm{d}t \le \sum_{i=j+1}^{k} \int_i^{i+1} t^\beta \mathrm{d}t
$$

$$
= \int_{j+1}^{k+1} t^\beta \mathrm{d}t = \varsigma_{\beta+1}(k+1) - \varsigma_{\beta+1}(j+1),
$$

which combined with the previous inequality implies (C.9a).

- Now consider the case where $\beta < 0$. We can obtain

$$
\sum_{i=j+1}^{k} i^\beta = \sum_{i=j+1}^{k} \int_i^{i+1} i^\beta \mathrm{d}t \ge \sum_{i=j+1}^{k} \int_i^{i+1} t^\beta \mathrm{d}t = \varsigma_{\beta+1}(k+1) - \varsigma_{\beta+1}(j+1),
$$

and

$$
\sum_{i=j+1}^{k} i^\beta = \sum_{i=j+1}^{k} (i+1)^\beta \cdot \left(\frac{i}{i+1}\right)^\beta \le \frac{1}{2^\beta} \sum_{i=j+1}^{k} (i+1)^\beta
$$

$$
= \frac{1}{2^\beta} \sum_{i=j+1}^{k} \int_i^{i+1} (i+1)^\beta \mathrm{d}t \le \frac{1}{2^\beta} \sum_{i=j+1}^{k} \int_i^{i+1} t^\beta \mathrm{d}t
$$

$$
= \frac{1}{2^\beta} \left[\varsigma_{\beta+1}(k+1) - \varsigma_{\beta+1}(j+1)\right],
$$

where the first inequality follows from (C.8) and the fact that $\beta < 0$. This completes the proof of (C.9b).

Now we assume that $j, k \in \mathbb{Z}_+$ with $k \ge j + 1$.

- If $\beta \le 0$, then we have

$$
\sum_{i=j+1}^{k} i^\beta = \sum_{i=j+1}^{k} \int_{i-1}^{i} i^\beta \mathrm{d}t \le \sum_{i=j+1}^{k} \int_{i-1}^{i} t^\beta \mathrm{d}t = \int_j^k t^\beta \mathrm{d}t = \varsigma_{\beta+1}(k) - \varsigma_{\beta+1}(j),
$$

that is, (C.10b) holds.

- If $\beta > 0$, then

$$\sum_{i=j+1}^{k} i^\beta = \sum_{i=j}^{k-1} i^\beta + k^\beta - j^\beta \leq \varsigma_{\beta+1}(k) - \varsigma_{\beta+1}(j) + k^\beta - j^\beta, \qquad \text{(C.19)}$$

where the inequality follows from (C.9a). The fact that $\beta > 0$ and $k \geq j+1 \geq 2$ implies

$$\begin{aligned}
&(\beta+1) \cdot [\varsigma_{\beta+1}(k) - \varsigma_{\beta+1}(j)] - (k^\beta - j^\beta) \\
&= (k^{\beta+1} - k^\beta) - (j^{\beta+1} - j^\beta) = k^\beta (k-1) + j^\beta (1-j) \\
&\geq k^\beta (k-1) + j^\beta (1-k) = (k^\beta - j^\beta)(k-1) \\
&\geq 0,
\end{aligned}$$

that is,

$$k^\beta - j^\beta \leq (\beta+1) \cdot [\varsigma_{\beta+1}(k) - \varsigma_{\beta+1}(j)].$$

Combining the preceding inequality with (C.19) yields (C.10a) and completes the proof of assertion (ii).

**Proof of assertion (iii).** Note that $\beta \in (0,1)$. We first show (C.11a) holds. If $t = -1$ then it holds trivially. Thus we consider the case where $t \in (-1, \infty)$. Define $g(t) \triangleq (1+t)^\beta$ for any $t \in (-1, \infty)$. Then $g$ is twice continuously differentiable with

$$g'(t) = \beta(1+t)^{\beta-1} \quad \text{and} \quad g''(t) = \beta(\beta-1)(1+t)^{\beta-2} \leq 0$$

for any $t \in (-1, \infty)$, which implies $g$ is concave over $(-1, \infty)$. Thus, we have

$$(1+t)^\beta = g(t) \leq g(0) + g'(0) \cdot t = 1 + \beta t \quad \text{for all } t \in (-1, \infty),$$

which completes the proof of (C.11a). Then we prove (C.11b) by using this result. Fix any $t \in \mathbb{R}_{++}$. It follows from the definition of $\varsigma_\beta$ given in (2.3) that

$$\varsigma_\beta(t) - \varsigma_\beta\left(\frac{t}{2}\right) = \frac{1}{\beta}\left[t^\beta - \left(\frac{t}{2}\right)^\beta\right] = \frac{t^\beta}{\beta}\left[1 - \left(1 - \frac{1}{2}\right)^\beta\right] \geq \frac{t^\beta}{\beta}\left[1 - \left(1 - \frac{\beta}{2}\right)\right] = \frac{t^\beta}{2},$$

where the inequality comes from (C.11a). This completes the proof of assertion (iii).

**Proof of assertion (iv).** This result follows immediately from the fact that $h$ is differentiable over $\mathbb{R}_{++}$ and

$$h'(t) = \frac{1 - \exp(-at) - t \cdot (a\exp(-at))}{(1 - \exp(-at))^2} = \frac{1 - (1+at)\exp(-at)}{(1 - \exp(-at))^2} \geq 0$$

for all $t \in \mathbb{R}_{++}$, where the inequality uses (B.11) with $t = at$. This completes the proof of Lemma C.4.

# D  PROOFS FOR THE RESULTS IN SECTION 2.3

In this section, we prove the main results on the convergence rate in terms of the KKT residual outlined in Section 2.3.

## D.1  PROOF OF LEMMA 2.6

Fix any $x \in \mathbb{R}^d$. Recall that $\mathcal{X}^* = \arg\min_{x \in \mathbb{R}^d} \phi(x)$. By assumption, we have $\phi$ is proper and closed convex. Then by using the characterization of the proximal mapping $\text{prox}_\phi(\cdot)$ (Rockafellar, 1976b, (2.2)), one can derive the following relation:

$$\bar{x} \in \mathcal{X}^* \quad \Leftrightarrow \quad \bar{x} - \bar{x} \in \partial\phi(x^*) \quad \Leftrightarrow \quad \bar{x} = \text{prox}_\phi(\bar{x}) \quad \Leftrightarrow \quad \bar{x} - \text{prox}_\phi(\bar{x}) = 0.$$

Thus, combining the preceding equivalence relation with the triangle inequality and the nonexpansiveness of the proximal mapping $\mathrm{prox}_\phi(\cdot)$ (Rockafellar, 1976b, Proposition 1(c)) gives

$$
\begin{aligned}
\left\| x - \mathrm{prox}_\phi(x) \right\|_2 &= \left\| x - \mathrm{prox}_\phi(x) - \left( \overline{x} - \mathrm{prox}_\phi(\overline{x}) \right) \right\|_2 \\
&\leq \left\| x - \overline{x} \right\|_2 + \left\| \mathrm{prox}_\phi(x) - \mathrm{prox}_\phi(\overline{x}) \right\|_2 \\
&\leq 2 \left\| x - \overline{x} \right\|_2 \quad \text{for all } \overline{x} \in \mathcal{X}^*,
\end{aligned}
$$

which implies

$$
\left\| x - \mathrm{prox}_\phi(x) \right\|_2 \leq 2 \, \mathrm{dist}\left( x, \mathcal{X}^* \right)
$$

and thus completes the proof of Lemma 2.6.

## D.2 PROOF OF COROLLARY 2.7

Fix any $k \in \mathbb{Z}_+$. It follows directly from the Jensen's inequality and Lemma 2.6 that

$$
\mathbb{E}\left[ \left\| x_k - \mathrm{prox}_\phi(x_k) \right\|_2 \right] \leq \sqrt{ \mathbb{E}\left[ \left\| x_k - \mathrm{prox}_\phi(x_k) \right\|_2^2 \right] } \leq 2 \sqrt{ \mathbb{E}\left[ \mathrm{dist}\left( x_k, \mathcal{X}^* \right) \right] }.
$$

Using this bound on the expected KKT residual, all results in Corollary 2.7 follows immediately from Corollary 2.5 proved earlier.

# E PROOFS FOR THE RESULTS IN SECTION 3

In this section, we present the proof of Proposition 3.1 shown in Section 3.

## E.1 PROOF OF PROPOSITION 3.1

**Proof of assertion (i).** Denote the sample space $\mathcal{S} \triangleq [n]$. For each $s \in \mathcal{S}$, define

$$
f(x; s) \triangleq \| x - p_s \|_2^2 \quad \text{for all } x \in \mathbb{R}^d.
$$

Then, the function $F(x)$ can be expressed as the expectation $\mathbb{E}_{s \sim P}[f(x; s)]$, where $P$ denotes the discrete uniform distribution over $\mathcal{S}$.

- For each $s \in \mathcal{S}$, function $f(\cdot; s)$ is real-valued, locally Lipschitz and 2-strongly convex, which implies $F$ is locally Lipschitz and Assumption 3 holds.

- Note that function $r(\cdot)$ is real-valued and $\lambda$-strongly convex. Then $\varphi(\cdot; s)$ is real-valued and $(2 + \lambda)$-strongly convex for each $s \in \mathcal{S}$, leading to the $(2 + \lambda)$-strong convexity of $\phi$ over $\mathbb{R}^d$. Thus, Assumption 1 and 4 hold.

- The existence and uniqueness of $x^*$ follow from the strong convexity of $\phi$. Additionally, it can be readily inferred from the optimality condition for differentiable convex functions that

$$
\left[ 0 = \nabla \phi\left( x^* \right) = \frac{2}{n} \sum_{i=1}^{n} \left( x^* - p_i \right) + \lambda x^* \right] \quad \Rightarrow \quad \left[ x^* = \frac{2}{2 + \lambda} \cdot \frac{1}{n} \sum_{i=1}^{n} p_i \right].
$$

Then, the optimal solution set $\mathcal{X}^* = \{ x^* \}$ is nonempty, and

$$
\mathbb{E}_{s \sim P}\left[ \left\| \nabla \varphi(x^*; s) \right\|_2^2 \right] = \frac{4}{n} \sum_{i=1}^{n} \left\| x^* - p_i \right\|_2^2 \triangleq \sigma_\phi < \infty,
$$

which proves the first and second conditions in Assumption 2. By Remark 1.1, we have (1.2) holds and thus the last condition in Assumption 2 holds.

This completes the proof of assertion (i).

**Proof of assertion (ii).** Fix any $k \in \mathbb{Z}_+$. By the definition of $x_{k+1}$ and the optimality condition of differentiable convex functions, we can obtain

$$x_{k+1} = \arg\min_{x \in \mathbb{R}^d} \left\{ \frac{1}{m} \sum_{i=1}^{m} \left\| x - p_{S_k^i} \right\|_2^2 + \frac{\lambda}{2} \|x\|_2^2 + \frac{1}{2\alpha_k} \|x - x_k\|_2^2 \right\}$$

$$= \frac{2\alpha_k}{(2+\lambda)\alpha_k + 1} \cdot \frac{1}{m} \sum_{i=1}^{m} p_{S_k^i} + \frac{1}{(2+\lambda)\alpha_k + 1} x_k. \tag{E.1}$$

Recall that $\Sigma = \max_{i \in [n]} \|p_i\|_2$. It follows from (E.1) and the triangle inequality that

$$\|x_{k+1}\|_2 \leq \frac{2\alpha_k}{(2+\lambda)\alpha_k + 1} \cdot \frac{1}{m} \sum_{i=1}^{m} \left\| p_{S_k^i} \right\|_2 + \frac{1}{(2+\lambda)\alpha_k + 1} \cdot \|x_k\|_2$$

$$\leq \frac{2\alpha_k}{(2+\lambda)\alpha_k + 1} \cdot \Sigma + \frac{1}{(2+\lambda)\alpha_k + 1} \cdot \|x_k\|_2.$$

Applying the recursion $k$ times yields that

$$\|x_{k+1}\|_2 \leq \left( \prod_{l=1}^{k} \frac{1}{(2+\lambda)\alpha_l + 1} \right) \cdot \|x_1\|_2 + \left( \sum_{l=1}^{k} 2\alpha_l \prod_{m=l}^{k} \frac{1}{(2+\lambda)\alpha_m + 1} \right) \cdot \Sigma$$

$$\leq \|x_1\|_2 + \left( \sum_{l=1}^{k} 2\alpha_l \prod_{m=l}^{k} \frac{1}{(2+\lambda)\alpha_m + 1} \right) \cdot \Sigma$$

$$= \|x_1\|_2 + \frac{2}{2+\lambda} \left( 1 - \prod_{m=1}^{k} \frac{1}{(2+\lambda)\alpha_m + 1} \right) \cdot \Sigma \tag{E.2}$$

$$\leq \|x_1\|_2 + \frac{2\Sigma}{2+\lambda} \leq \|x_1\|_2 + \Sigma,$$

where the equality comes from the following relation:

$$\sum_{l=1}^{k} 2\alpha_l \prod_{m=l}^{k} \frac{1}{(2+\lambda)\alpha_m + 1} = \sum_{l=1}^{k} \frac{2\alpha_l}{(2+\lambda)\alpha_l + 1} \prod_{m=l+1}^{k} \frac{1}{(2+\lambda)\alpha_m + 1}$$

$$= \sum_{l=1}^{k} \frac{2\alpha_l}{(2+\lambda)\alpha_l + 1} \cdot \frac{1}{1 - \frac{1}{(2+\lambda)\alpha_l + 1}} \left( \prod_{m=l+1}^{k} \frac{1}{(2+\lambda)\alpha_m + 1} - \prod_{m=l}^{k} \frac{1}{(2+\lambda)\alpha_m + 1} \right)$$

$$= \frac{2}{2+\lambda} \cdot \sum_{l=1}^{k} \left( \prod_{m=l+1}^{k} \frac{1}{(2+\lambda)\alpha_m + 1} - \prod_{m=l}^{k} \frac{1}{(2+\lambda)\alpha_m + 1} \right)$$

$$= \frac{2}{2+\lambda} \cdot \left( 1 - \prod_{m=1}^{k} \frac{1}{(2+\lambda)\alpha_m + 1} \right).$$

Assertion (ii) follows directly from (E.2) and the trivial observation that $\|x_1\|_2 \leq \|x_1\|_2 + \Sigma$.

**Proof of assertion (iii).** Recall that $\sigma^2 = \frac{1}{n} \sum_{j=1}^{n} \|p_j - \frac{1}{n} \sum_{i=1}^{n} p_i\|_2^2$. Then $\sigma^2 \geq 0$ and

$$\sigma^2 = 0 \quad \Leftrightarrow \quad p_j = \frac{1}{n} \sum_{i=1}^{n} p_i \quad \text{for all } j \in [n] \quad \Leftrightarrow \quad p_i = p_j \quad \text{for all } i, j \in [n],$$

from which we have $\sigma^2 > 0$ according to the assumption that $p_{i_0} \neq p_{j_0}$ for some $i_0 \neq j_0 \in [n]$. Fix any $k \in \mathbb{Z}_+$. It follows from (E.1) that

$$x_{k+1} - x^* = \frac{2\alpha_k}{(2+\lambda)\alpha_k + 1} \cdot \frac{1}{m} \sum_{i=1}^{m} p_{S_k^i} + \frac{1}{(2+\lambda)\alpha_k + 1} x_k - x^*,$$

which implies

$$\mathbb{E}_k\left[\|x_{k+1}-x^*\|_2^2\right]=\mathbb{E}_k\left[\langle x_{k+1}-x^*,x_{k+1}-x^*\rangle\right]$$

$$=\frac{4\alpha_k^2}{[(2+\lambda)\alpha_k+1]^2}\mathbb{E}_k\left[\left\|\frac{1}{m}\sum_{i=1}^m p_{S_k^i}\right\|_2^2\right]+\frac{2\alpha_k}{[(2+\lambda)\alpha_k+1]^2}\mathbb{E}_k\left[\left\langle\frac{1}{m}\sum_{i=1}^m p_{S_k^i},x_k\right\rangle\right]$$

$$-\frac{2\alpha_k}{(2+\lambda)\alpha_k+1}\mathbb{E}_k\left[\left\langle\frac{1}{m}\sum_{i=1}^m p_{S_k^i},x^*\right\rangle\right] \qquad\qquad\text{(E.3)}$$

$$+\frac{2\alpha_k}{[(2+\lambda)\alpha_k+1]^2}\mathbb{E}_k\left[\left\langle x_k,\frac{1}{m}\sum_{i=1}^m p_{S_k^i}\right\rangle\right]+\frac{\|x_k\|_2^2}{[(2+\lambda)\alpha_k+1]^2}-\frac{\langle x_k,x^*\rangle}{(2+\lambda)\alpha_k+1}$$

$$-\frac{2\alpha_k}{(2+\lambda)\alpha_k+1}\mathbb{E}_k\left[\left\langle x^*,\frac{1}{m}\sum_{i=1}^m p_{S_k^i}\right\rangle\right]-\frac{\langle x^*,x_k\rangle}{(2+\lambda)\alpha_k+1}+\|x^*\|_2^2.$$

Note that

$$\mathbb{E}_k\left[p_{S_k^i}\right]=\frac{1}{n}\sum_{j=1}^n p_j\quad\text{for all }i\in[m]\quad\text{and}\quad\mathbb{E}_k\left[\frac{1}{m}\sum_{i=1}^m p_{S_k^i}\right]=\frac{1}{n}\sum_{j=1}^n p_j=\frac{2+\lambda}{2}x^*.\quad\text{(E.4)}$$

Then we can prove the following two assertions:

(a) It follows from (E.4) that

$$\mathbb{E}_k\left[\left\langle\frac{1}{m}\sum_{i=1}^m p_{S_k^i},x_k\right\rangle\right]=\left\langle\frac{1}{n}\sum_{j=1}^n p_j,x_k\right\rangle=\left\langle\frac{2+\lambda}{2}x^*,x_k\right\rangle=\frac{2+\lambda}{2}\langle x_k,x^*\rangle,$$

$$\mathbb{E}_k\left[\left\langle\frac{1}{m}\sum_{i=1}^m p_{S_k^i},x^*\right\rangle\right]=\left\langle\frac{1}{n}\sum_{j=1}^n p_j,x^*\right\rangle=\left\langle\frac{2+\lambda}{2}x^*,x^*\right\rangle=\frac{2+\lambda}{2}\|x^*\|_2^2,$$

which combined with (E.3) yields that

$$\mathbb{E}_k\left[\|x_{k+1}-x^*\|_2^2\right]$$

$$=\frac{4\alpha_k^2}{[(2+\lambda)\alpha_k+1]^2}\mathbb{E}_k\left[\left\|\frac{1}{m}\sum_{i=1}^m p_{S_k^i}\right\|_2^2\right]+\frac{2}{(2+\lambda)\alpha_k+1}\left(\frac{(2+\lambda)\alpha_k}{(2+\lambda)\alpha_k+1}-1\right)\langle x_k,x^*\rangle$$

$$+\frac{\|x_k\|_2^2}{[(2+\lambda)\alpha_k+1]^2}+\left(1-\frac{2(2+\lambda)\alpha_k}{(2+\lambda)\alpha_k+1}\right)\|x^*\|_2^2$$

$$=\frac{\|x_k\|_2^2-2\langle x_k,x^*\rangle+\|x^*\|_2^2}{[(2+\lambda)\alpha_k+1]^2}+\frac{4\alpha_k^2}{[(2+\lambda)\alpha_k+1]^2}\mathbb{E}_k\left[\left\|\frac{1}{m}\sum_{i=1}^m p_{S_k^i}\right\|_2^2\right]$$

$$+\left(1-\frac{2(2+\lambda)\alpha_k}{(2+\lambda)\alpha_k+1}-\frac{1}{[(2+\lambda)\alpha_k+1]^2}\right)\|x^*\|_2^2$$

$$=\frac{\|x_k-x^*\|_2^2}{[(2+\lambda)\alpha_k+1]^2}+\frac{4\alpha_k^2}{[(2+\lambda)\alpha_k+1]^2}\left(\mathbb{E}_k\left[\left\|\frac{1}{m}\sum_{i=1}^m p_{S_k^i}\right\|_2^2\right]-\left\|\frac{2+\lambda}{2}x^*\right\|_2^2\right).$$

(b) Using (E.4) again, one can derive that

$$\mathbb{E}_k\left[\left\|\frac{1}{m}\sum_{i=1}^m p_{S_k^i}\right\|_2^2\right]-\left\|\frac{2+\lambda}{2}x^*\right\|_2^2=\mathbb{E}_k\left[\left\|\frac{1}{m}\sum_{i=1}^m p_{S_k^i}\right\|_2^2\right]-\left\|\mathbb{E}_k\left[\frac{1}{m}\sum_{i=1}^m p_{S_k^i}\right]\right\|_2^2$$

$$= \mathbb{E}_k \left[ \left\| \frac{1}{m} \sum_{i=1}^m p_{S_k^i} - \mathbb{E}_k \left[ \frac{1}{m} \sum_{i=1}^m p_{S_k^i} \right] \right\|_2^2 \right] = \mathbb{E}_k \left[ \left\| \frac{1}{m} \sum_{i=1}^m \left( p_{S_k^i} - \mathbb{E}_k \left[ p_{S_k^i} \right] \right) \right\|_2^2 \right]$$

$$= \frac{1}{m^2} \sum_{i=1}^m \mathbb{E}_k \left[ \left\| p_{S_k^i} - \mathbb{E}_k \left[ p_{S_k^i} \right] \right\|_2^2 \right] = \frac{1}{m^2} \sum_{i=1}^m \mathbb{E}_k \left[ \left\| p_{S_k^i} - \frac{1}{n} \sum_{j=1}^n p_j \right\|_2^2 \right] = \frac{\sigma^2}{m},$$

where the third last equality holds because the random variables $S_k^1, \cdots, S_k^m$ are independent and identically distributed, and the last comes from the definition of $\sigma$.

Assertion (a) together with assertion (b) above leads us to the result that

$$\mathbb{E}_k \left[ \|x_{k+1} - x^*\|_2^2 \right] = \frac{\|x_k - x^*\|_2^2}{[(2+\lambda)\alpha_k + 1]^2} + \frac{4\alpha_k^2}{[(2+\lambda)\alpha_k + 1]^2} \cdot \frac{\sigma^2}{m}.$$

Taking the expectation of both sides of the previous inequality and applying the law of total expectation gives

$$\mathbb{E} \left[ \|x_{k+1} - x^*\|_2^2 \right] = \frac{\mathbb{E} \left[ \|x_k - x^*\|_2^2 \right]}{[(2+\lambda)\alpha_k + 1]^2} + \frac{4\alpha_k^2}{[(2+\lambda)\alpha_k + 1]^2} \cdot \frac{\sigma^2}{m}. \tag{E.5}$$

Denote by $\delta_1 \triangleq \|x_1 - x^*\|_2^2$. Applying the recursion $k$ times yields that

$$\mathbb{E} \left[ \|x_{k+1} - x^*\|_2^2 \right]$$
$$= \left( \prod_{l=1}^k \frac{1}{[(2+\lambda)\alpha_l + 1]^2} \right) \cdot \delta_1 + \frac{4\sigma^2}{m} \sum_{l=1}^k \alpha_l^2 \cdot \left( \prod_{m=l}^k \frac{1}{[(2+\lambda)\alpha_m + 1]^2} \right) \tag{E.6}$$

holds for all $k \in \mathbb{Z}_+$.

- First consider the case in which $\alpha_k \geq \alpha_0$ for some $\alpha_0 \in \mathbb{R}_{++}$. Define $h(t) \triangleq \frac{t}{(2+\lambda)t+1}$ for all $t \in \mathbb{R}_{++}$. Then $h$ is differentiable over $\mathbb{R}_{++}$ and

$$h'(t) = \frac{(2+\lambda)t + 1 - (2+\lambda)t}{[(2+\lambda)t + 1]^2} = \frac{1}{[(2+\lambda)t + 1]^2} > 0 \quad \text{for all } t \in \mathbb{R}_{++},$$

that is, $h$ is nondecreasing over $\mathbb{R}_{++}$. We can thus obtain from (E.5) that

$$\mathbb{E} \left[ \|x_{k+1} - x^*\|_2^2 \right] \geq \frac{4\alpha_k^2}{[(2+\lambda)\alpha_k + 1]^2} \cdot \frac{\sigma^2}{m} = \left( \frac{\alpha_k}{(2+\lambda)\alpha_k + 1} \right)^2 \cdot \frac{4\sigma^2}{m}$$

$$\geq \left( \frac{\alpha_0}{(2+\lambda)\alpha_0 + 1} \right)^2 \cdot \frac{4\sigma^2}{m} = \frac{4\sigma^2 \alpha_0^2}{m \left[ (2+\lambda)\alpha_0 + 1 \right]^2}$$

holds for all $k \in \mathbb{Z}_+$.

- Now we prove the assertion for the case where $\alpha_k = \alpha_0 k^{-\beta}$ for some $\alpha_0 \in \mathbb{R}_{++}$ and $\beta \in (0, 1]$. Observe that

$$\sum_{l=1}^k \alpha_l^2 \cdot \left( \prod_{m=l}^k \frac{1}{[(2+\lambda)\alpha_m + 1]^2} \right)$$

$$= \sum_{l=1}^k \frac{\alpha_l^2}{[(2+\lambda)\alpha_l + 1]^2} \cdot \left( \prod_{m=l+1}^k \frac{1}{[(2+\lambda)\alpha_m + 1]^2} \right)$$

$$\geq \frac{1}{[(2+\lambda)\alpha_0 + 1]^3} \sum_{l=1}^k \alpha_l^2 \cdot \left( \prod_{m=l+1}^k \frac{1}{(2+\lambda)\alpha_m + 1} \right),$$

where the inequality uses the fact that $\alpha_k \leq \alpha_0$ for all $k \in \mathbb{Z}_+$. Moreover, the last term is bounded below by

$$
\sum_{l=1}^{k} \alpha_l^2 \cdot \left( \prod_{m=l+1}^{k} \frac{1}{(2+\lambda)\alpha_m + 1} \right)
$$

$$
= \sum_{l=1}^{k} \alpha_l^2 \cdot \frac{1}{1 - \frac{1}{(2+\lambda)\alpha_l + 1}} \left( \prod_{m=l+1}^{k} \frac{1}{(2+\lambda)\alpha_m + 1} - \prod_{m=l}^{k} \frac{1}{(2+\lambda)\alpha_m + 1} \right)
$$

$$
= \sum_{l=1}^{k} \frac{(2+\lambda)\alpha_l^2 + \alpha_l}{2+\lambda} \left( \prod_{m=l+1}^{k} \frac{1}{(2+\lambda)\alpha_m + 1} - \prod_{m=l}^{k} \frac{1}{(2+\lambda)\alpha_m + 1} \right)
$$

$$
\geq \sum_{l=1}^{k} \frac{\alpha_l}{2+\lambda} \left( \prod_{m=l+1}^{k} \frac{1}{(2+\lambda)\alpha_m + 1} - \prod_{m=l}^{k} \frac{1}{(2+\lambda)\alpha_m + 1} \right)
$$

$$
\geq \frac{\alpha_k}{2+\lambda} \sum_{l=1}^{k} \left( \prod_{m=l+1}^{k} \frac{1}{(2+\lambda)\alpha_m + 1} - \prod_{m=l}^{k} \frac{1}{(2+\lambda)\alpha_m + 1} \right)
$$

$$
= \frac{\alpha_k}{2+\lambda} \left( 1 - \prod_{m=1}^{k} \frac{1}{(2+\lambda)\alpha_m + 1} \right) \geq \frac{\alpha_k}{2+\lambda} \left( 1 - \frac{1}{(2+\lambda)\alpha_1 + 1} \right)
$$

$$
= \frac{\alpha_k}{2+\lambda} \left( 1 - \frac{1}{(2+\lambda)\alpha_0 + 1} \right) = \frac{\alpha_0^2}{(2+\lambda)\alpha_0 + 1} \cdot k^{-\beta},
$$

where the second inequality is due to the fact that $\alpha_l \geq \alpha_k$ for all $l \in \mathbb{Z}_+$ with $l \leq k$, and the second last equality comes from the fact that $\alpha_1 = \alpha_0$ implied by the definition of the stepsizes. Combining (E.6) with the previous two relations leads us to the conclusion that

$$
\mathbb{E}\left[ \|x_{k+1} - x^*\|_2^2 \right] \geq \frac{4\sigma^2}{m} \sum_{l=1}^{k} \alpha_l^2 \cdot \left( \prod_{m=l}^{k} \frac{1}{[(2+\lambda)\alpha_m + 1]^2} \right)
$$

$$
\geq \frac{4\sigma^2}{m} \cdot \frac{1}{[(2+\lambda)\alpha_0 + 1]^3} \cdot \frac{\alpha_0^2}{(2+\lambda)\alpha_0 + 1} \cdot k^{-\beta}
$$

$$
= \frac{4\sigma^2 \alpha_0^2}{m [(2+\lambda)\alpha_0 + 1]^4} \cdot k^{-\beta}
$$

holds for all $k \in \mathbb{Z}_+$.

This completes the proof of assertion (iii) and thus proves Proposition 3.1.

## F  DETAILS OF NUMERICAL EXPERIMENTS

In this section, we provide the essential details required to implement the numerical experiments described in Section 4.

### F.1  LASSO LINEAR REGRESSION MODEL

Recall the Lasso linear regression model defined as:

$$
\min_{x \in \mathbb{R}^d} \psi_{\text{Lasso}}(x) \triangleq \frac{1}{2} \|Ax - b\|_2^2 + \lambda_1 \|x\|_1 .
$$

**Solve inner-loop subproblems.**  Denote the sample space $\mathcal{S}$ by the finite discrete domain $[n]$. Then, when using isPPA to solve Lasso linear regression model, the inner-loop subproblem (1.3) at iteration $k$ can be formulated as

$$
\min_{x \in \mathbb{R}^d} \left\{ \frac{1}{2m} \|A_{S_k^{1:m},:}x - b_{S_k^{1:m}}\|_2^2 + \frac{\lambda_1}{n} \|x\|_1 + \frac{1}{2\alpha_k} \|x - x_k\|_2^2 \right\} \tag{F.1}
$$

where $A_{S_k^{1:m},:}$ denotes the submatrix of $A$ consisting of rows indexed by $S_k^{1:m}$ and $b_{S_k^{1:m}}$ denotes the subvector generated by the elements of $b$ indexed by $S_k^{1:m}$. Set $\tilde{\lambda}_1 \triangleq \frac{m}{n}\lambda_1$ and $\tilde{\alpha}_k \triangleq \frac{\alpha_k}{m}$. Then (F.1) can be written as

$$\min_{x \in \mathbb{R}^d} \left\{ H\left(A_{S_k^{1:m},:}x\right) + p(x) + \frac{1}{2\tilde{\alpha}_k}\|x - x_k\|_2^2 \right\} \tag{F.2}$$

$$\text{where} \quad H(y) \triangleq \frac{1}{2}\left\|y - b_{S_k^{1:m}}\right\|_2^2 \text{ for all } y \in \mathbb{R}^m \quad \text{and} \quad p(x) \triangleq \tilde{\lambda}_1\|x\|_1 \text{ for all } x \in \mathbb{R}^d.$$

In an effort to obtain the next iterate $x_{k+1}$ satisfying the stopping criterion (SCA), we propose the following claim and defer its proof to the end of this section for reference.

**Claim F.1.** *Consider the subproblem (F.2) at iteration $k$ of isPPA (Algorithm 1) with stepsizes $\{\alpha_k\}$ and accuracy parameters $\{\epsilon_k\}$. Define*

$$\Psi(\xi) \triangleq H^\star(\xi) + p^\star\left(\text{prox}_{\frac{p^\star}{\tilde{\alpha}_k}}\left(\frac{x_k}{\tilde{\alpha}_k} - A_{S_k^{1:m},:}^\top\xi\right)\right)$$

$$+ \frac{\tilde{\alpha}_k}{2}\left\|\frac{x_k}{\tilde{\alpha}_k} - A_{S_k^{1:m},:}^\top\xi - \text{prox}_{\frac{p^\star}{\tilde{\alpha}_k}}\left(\frac{x_k}{\tilde{\alpha}_k} - A_{S_k^{1:m},:}^\top\xi\right)\right\|_2^2 \quad \text{for all } \xi \in \mathbb{R}^m,$$

*where $H^\star$ and $p^\star$ are the Fenchel conjugate functions of $H$ and $p$, respectively. Fix any $k \in \mathbb{Z}_+$. If $x_{k+1}$ is obtained via the following update rule:*

$$u_{k+1} = \text{prox}_{\frac{p^\star}{\tilde{\alpha}_k}}\left(\frac{x_k}{\tilde{\alpha}_k} - A_{S_k^{1:m},:}^\top\xi_{k+1}\right) \quad \text{with}$$

$$\xi_{k+1} \approx \arg\min_{\xi \in \mathbb{R}^m} \Psi(\xi) \quad \text{where} \quad \|\nabla\Psi(\xi_{k+1})\|_2 \le \frac{\epsilon_k}{\sqrt{\tilde{\alpha}_k}}, \tag{F.3}$$

$$x_{k+1} = x_k - \tilde{\alpha}_k\left(A_{S_k^{1:m},:}^\top\xi_{k+1} + u_{k+1}\right) = \text{prox}_{\tilde{\alpha}_k p}\left(x_k - \tilde{\alpha}_k A_{S_k^{1:m},:}^\top\xi_{k+1}\right),$$

*then $x_{k+1}$ satisfies the stopping criterion (SCA), i.e., $\left\|x_{k+1} - \text{prox}_{\alpha_k \overline{\varphi}(\cdot;S_k^{1:m})}(x_k)\right\|_2 \le \epsilon_k$.*

Note that

$$H^\star(\xi) = \sup_{y \in \mathbb{R}^m} \{\langle y, \xi\rangle - H(y)\} = \frac{1}{2}\|\xi\|_2^2 + \left\langle b_{S_k^{1:m}}, \xi\right\rangle \quad \text{for all } \xi \in \mathbb{R}^m. \tag{F.4}$$

Then the function $\Psi$ defined in Claim F.1 is 1-strongly convex and continuously differentiable on $\mathbb{R}^m$ with

$$\nabla\Psi(\xi) = \nabla H^\star(\xi) - \tilde{\alpha}_k A_{S_k^{1:m},:}\left(\frac{x_k}{\tilde{\alpha}_k} - A_{S_k^{1:m},:}^\top\xi - \text{prox}_{\frac{p^\star}{\tilde{\alpha}_k}}\left(\frac{x_k}{\tilde{\alpha}_k} - A_{S_k^{1:m},:}^\top\xi\right)\right)$$

$$= \xi + b_{S_k^{1:m}} - A_{S_k^{1:m},:}\text{prox}_{\tilde{\alpha}_k p}\left(x_k - \tilde{\alpha}_k A_{S_k^{1:m},:}^\top\xi\right) \quad \text{for all } \xi \in \mathbb{R}^m.$$

Therefore, utilizing Claim F.1, we can obtain $\xi_{k+1}$ by applying the semismooth Newton (SSN) method (Li et al., 2018, Algorithm SSN) to solve the minimization subproblem in (F.3).

**Obtain an approximate optimal solution.** For each data set, we first apply the semismooth Newton augmented Lagrangian (SSNAL) method (Li et al., 2018) to solve problem (4.1) and obtain an approximate solution $\tilde{x}^*$ satisfying the relative KKT residual $\eta_{\text{rel,Lasso}} < 10^{-8}$, where

$$\eta_{\text{rel,Lasso}} \triangleq \frac{\left\|\tilde{x}^* - \text{prox}_{\lambda_1\|\cdot\|_1}\left(\tilde{x}^* - A^\top(A\tilde{x}^* - b)\right)\right\|_2}{1 + \|\tilde{x}^*\|_2 + \|A^\top(A\tilde{x}^* - b)\|_2}.$$

Then $\psi_{\text{Lasso}}(\tilde{x}^*)$ can be taken as an approximation of the optimal function value. For each trial, we run isPPA to solve (4.1) and obtain an approximate solution $\hat{x}^*$ such that the relative gap $\delta_{\text{rel,Lasso}} < 10^{-5}$, where

$$\delta_{\text{rel,Lasso}} \triangleq \frac{|\psi_{\text{Lasso}}(\hat{x}^*) - \psi_{\text{Lasso}}(\tilde{x}^*)|}{1 + |\psi_{\text{Lasso}}(\tilde{x}^*)|}.$$

Then $\hat{x}^*$ is the desired approximation of the optimal solution, to which $\{x_k\}$ converges almost surely.

**Numerical results on real data sets.** The real data sets are taken from the UCI data repository (Chang & Lin, 2011) and the original features of the obtained data are expeneded by following the settings in (Li et al., 2018). We test on real data sets `abalone7` with $(n, d) = (4177, 6435)$ and `space_ga9` with $(n, d) = (3107, 5005)$, and set the regurlarization parameter $\lambda_1 = \lambda_c \left\| A^\top b \right\|_\infty$ with $\lambda_c = 10^{-2}$. The KKT residual with respect to the Lasso model is defined by

$$\eta_{\text{Lasso}}(x) \triangleq \left\| x - \text{prox}_{\lambda_1 \| \cdot \|_1} \left( x - A^\top (Ax - b) \right) \right\|_2 \quad \text{for all } x \in \mathbb{R}^d. \tag{F.5}$$

Given that $\overline{x} \in \mathcal{X}^*$ if and only if $\overline{x} = \text{prox}_{\lambda_1 \| \cdot \|_1} (\overline{x} - A^\top (A\overline{x} - b))$, it follows that $\eta_{\text{Lasso}}(x) \leq (2 + \|A\|_2^2)\text{dist}(x, \mathcal{X}^*)$, ensuring the applicability of all results from Corollary 2.7 to $\eta_{\text{Lasso}}$. As illustrated in the top row of Figure 3, the convergence curves are analogous to those for the synthetic data, where the asymptotic convergence rates for the squared distance to the optimal solution set and the KKT residual are $\mathcal{O}(k^{-\beta})$ and $\mathcal{O}(k^{-\beta/2})$, respectively.

### F.2 ELASTIC NET LINEAR REGRESSION MODEL

The elastic net linear regression model is presented as follows:

$$\min_{x \in \mathbb{R}^d} \psi_{\text{elastic}}(x) \triangleq \frac{1}{2} \|Ax - b\|_2^2 + \lambda_1 \|x\|_1 + \frac{\lambda_2}{2} \|x\|_2^2.$$

**Solve inner-loop subproblems.** Use the same notations as in Appendix F.1 and set

$$\tilde{\lambda}_2 \triangleq \frac{m}{n} \lambda_2, \quad \hat{x}_k \triangleq \frac{1}{1 + \tilde{\lambda}_2 \tilde{\alpha}_k} x_k \quad \text{and} \quad \hat{\alpha}_k \triangleq \frac{\tilde{\alpha}_k}{1 + \tilde{\lambda}_2 \tilde{\alpha}_k}.$$

Then we can formulate the inner-loop subproblem (1.3) at iteration $k$ as follows:

$$\min_{x \in \mathbb{R}^d} \left\{ H\left( A_{S_k^{1:m},:} x \right) + p(x) + \frac{\tilde{\lambda}_2}{2} \|x\|_2^2 + \frac{1}{2\tilde{\alpha}_k} \|x - x_k\|_2^2 \right\}$$

$$\Leftrightarrow \quad \min_{x \in \mathbb{R}^d} \left\{ H\left( A_{S_k^{1:m},:} x \right) + p(x) + \frac{1 + \tilde{\lambda}_2 \tilde{\alpha}_k}{2\tilde{\alpha}_k} \left\| x - \frac{1}{1 + \tilde{\lambda}_2 \tilde{\alpha}_k} x_k \right\|_2^2 \right\} \tag{F.6}$$

$$\Leftrightarrow \quad \min_{x \in \mathbb{R}^d} \left\{ H\left( A_{S_k^{1:m},:} x \right) + p(x) + \frac{1}{2\hat{\alpha}_k} \|x - \hat{x}_k\|_2^2 \right\}$$

where $H(y) \triangleq \frac{1}{2} \left\| y - b_{S_k^{1:m}} \right\|_2^2$ for all $y \in \mathbb{R}^m$ and $p(x) \triangleq \tilde{\lambda}_1 \|x\|_1$ for all $x \in \mathbb{R}^d$.

By using the equivalent formulation (F.6) described above and following the proof argument of Claim F.1 for the Lasso model, one can easily check the validity of the claim below:

**Claim F.2.** *Consider the subproblem (F.6) at iteration $k$ of isPPA (Algorithm 1) with stepsizes $\{\alpha_k\}$ and accuracy parameters $\{\epsilon_k\}$. Define*

$$\Psi(\xi) \triangleq H^\star(\xi) + p^\star \left( \text{prox}_{\frac{p^\star}{\hat{\alpha}_k}} \left( \frac{\hat{x}_k}{\hat{\alpha}_k} - A_{S_k^{1:m},:}^\top \xi \right) \right)$$

$$+ \frac{\hat{\alpha}_k}{2} \left\| \frac{\hat{x}_k}{\hat{\alpha}_k} - A_{S_k^{1:m},:}^\top \xi - \text{prox}_{\frac{p^\star}{\hat{\alpha}_k}} \left( \frac{\hat{x}_k}{\hat{\alpha}_k} - A_{S_k^{1:m},:}^\top \xi \right) \right\|_2^2 \quad \text{for all } \xi \in \mathbb{R}^m,$$

*where $H^\star$ and $p^\star$ are the Fenchel conjugate functions of $H$ and $p$, respectively. Fix any $k \in \mathbb{Z}_+$. If $x_{k+1}$ is obtained via the following update rule:*

$$u_{k+1} = \text{prox}_{\frac{p^\star}{\hat{\alpha}_k}} \left( \frac{\hat{x}_k}{\hat{\alpha}_k} - A_{S_k^{1:m},:}^\top \xi_{k+1} \right) \quad \text{with}$$

$$\xi_{k+1} \approx \arg\min_{\xi \in \mathbb{R}^m} \Psi(\xi) \quad \text{where} \quad \|\nabla \Psi(\xi_{k+1})\|_2 \leq \frac{\epsilon_k}{\sqrt{\hat{\alpha}_k}}, \tag{F.7}$$

$$x_{k+1} = \hat{x}_k - \hat{\alpha}_k \left( A_{S_k^{1:m},:}^\top \xi_{k+1} + u_{k+1} \right) = \text{prox}_{\hat{\alpha}_k p} \left( \hat{x}_k - \hat{\alpha}_k A_{S_k^{1:m},:}^\top \xi_{k+1} \right),$$

*then $x_{k+1}$ satisfies the stopping criterion (SCA), i.e., $\left\| x_{k+1} - \text{prox}_{\alpha_k \overline{\varphi}(\cdot; S_k^{1:m})}(x_k) \right\|_2 \leq \epsilon_k$.*

Similar to the Lasso model, the iterates $\{x_k\}$ generated by isPPA are computed by employing the SSN method to solve the minimization subproblem specified in (F.7).

**Obtain an approximate optimal solution.** For each data set, we employ the SSNAL method to solve problem (4.2) and obtain an approximate solution $\tilde{x}^*$ satisfying the relative KKT residual $\eta_{\text{rel,elastic}} < 10^{-8}$, where

$$\eta_{\text{rel,elastic}} \triangleq \frac{\left\| \tilde{x}^* - \text{prox}_{\lambda_1 \|\cdot\|_1} \left( \tilde{x}^* - A^\top \left( A\tilde{x}^* - b \right) - \lambda_2 \tilde{x}^* \right) \right\|_2}{1 + \|\tilde{x}^*\|_2 + \|A^\top \left( A\tilde{x}^* - b \right) + \lambda_2 \tilde{x}^*\|_2}.$$

Then $\tilde{x}^*$ is the desired approximation of the optimal solution to (4.2).

**Test setup for synthetic and real data and numerical results for real data sets.** For numerical experiments on synthetic data, we use $n = 10000$ and $d = 1000$, setting $\sigma$ to zero for noiseless conditions and to $10^{-2}$ otherwise. The regularization parameters, $\lambda_1$ and $\lambda_2$, are set to $\lambda_{c_1} \left\| A^\top b \right\|_\infty$ and $\lambda_{c_2} \left\| A^\top b \right\|_\infty$, respectively, with $\lambda_{c_1} = 5 \times 10^{-2}$ and $\lambda_{c_2} = 5 \times 10^{-2}$. For test on real data sets, specifically the data sets `abalone7` and `space_ga9`, we set the regularization parameters to $(\lambda_1, \lambda_2) = (\lambda_{c_1} \left\| A^\top b \right\|_\infty, \lambda_{c_2} \left\| A^\top b \right\|_\infty)$, with $(\lambda_{c_1}, \lambda_{c_2}) = (10^{-1}, 10^{-1})$ for `abalone7` and $(\lambda_{c_1}, \lambda_{c_2}) = (10^{-2}, 10^{-2})$ for `space_ga9`, respectively. The accuracy parameter $\epsilon_k$ is defined as $\gamma \alpha_k^2$ with $\gamma = 10^{-2}$, and the minibatch size $m$ is fixed at 32. The KKT residual for the elastic net Lasso model is defined as:

$$\eta_{\text{elastic}}(x) \triangleq \left\| x - \text{prox}_{\lambda_1 \|\cdot\|_1} \left( x - A^\top \left( Ax - b \right) - \lambda_2 x \right) \right\|_2, \tag{F.8}$$

which is upper bounded by $(2 + \lambda_2 + \|A\|_2^2)\text{dist}(x, \mathcal{X}^*)$, affirming that all the results from Corollary 2.7 apply to $\eta_{\text{elastic}}$. The performance of isPPA with diminishing stepsizes $\alpha_k = \alpha_0 k^{-\beta}$ for various stepsize exponents $\beta \in \{0.55, 0.75, 0.9, 1\}$, starting with an initial stepsize $\alpha_0 = 50$, is displayed in the bottom row of Figure 3. The observed convergence results align with those tested on synthetic data, thus validating Corollary 2.5 and 2.7.

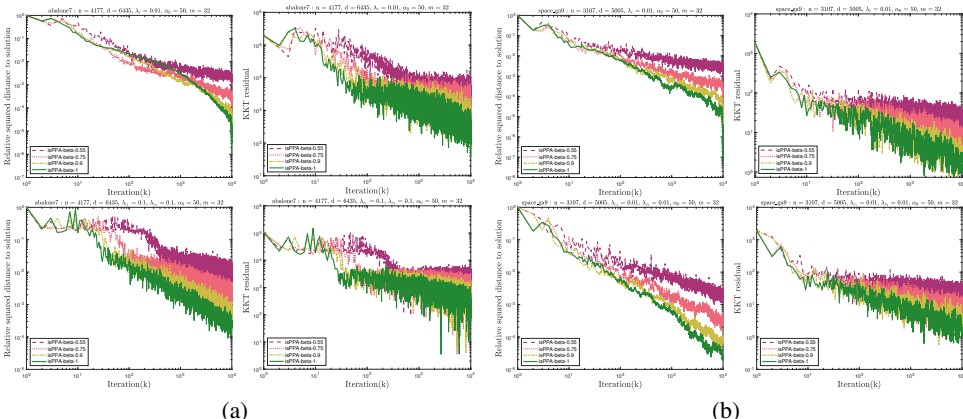

(a)                    (b)

Figure 3: Performance of isPPA in solving the linear regression models (Lasso - top and elastic net - bottom) for four values of stepsize exponents $\beta = 0.55, 0.75, 0.9$ and $1$. The legend "isPPA-beta-$\beta$" denotes isPPA with diminishing stepsizes $\alpha_k = \alpha_0 k^{-\beta}$. (a) The test on data set `abalone7`. (b) The test on data set `space_ga9`.

### F.3    PROOF OF CLAIM F.1

The proof of Claim F.1 are based on the following result. Here we still use $H^\star$ and $p^\star$ to denote the Fenchel conjugate functions of $H$ and $p$, respectively.

**Claim F.3.** *Consider the following composite optimization problem*

$$\min_{x \in \mathbb{R}^d} \Phi(x) \triangleq H(Ax) + p(x), \tag{F.9}$$

*where $A \in \mathbb{R}^{m \times d}$, $H \colon \mathbb{R}^m \to (-\infty, +\infty]$ and $p \colon \mathbb{R}^d \to (-\infty, +\infty]$ are proper and closed convex functions. Fix any $\overline{x} \in \mathbb{R}^d$, $\alpha \in \mathbb{R}_{++}$ and $\epsilon \in \mathbb{R}_+$. Define*

$$\Psi(\xi) \triangleq H^\star(\xi) + p^\star \left( \mathrm{prox}_{\frac{p^\star}{\alpha}} \left( \frac{\overline{x}}{\alpha} - A^\top \xi \right) \right)$$

$$+ \frac{\alpha}{2} \left\| \frac{\overline{x}}{\alpha} - A^\top \xi - \mathrm{prox}_{\frac{p^\star}{\alpha}} \left( \frac{\overline{x}}{\alpha} - A^\top \xi \right) \right\|_2^2 \quad \textit{for all } \xi \in \mathbb{R}^m.$$

*Suppose that $H^\star$ is $\eta_H$-strongly convex for some $\eta_H \in \mathbb{R}_{++}$. If*

$$\tilde{u} \triangleq \mathrm{prox}_{\frac{p^\star}{\alpha}} \left( \frac{\overline{x}}{\alpha} - A^\top \tilde{\xi} \right) \quad \textit{with}$$

$$\tilde{\xi} \approx \arg\min_{\xi \in \mathbb{R}^m} \ \Psi(\xi) \quad \textit{where} \quad \left\| \nabla \Psi \left( \tilde{\xi} \right) \right\|_2 \le \sqrt{\frac{\eta_H}{\alpha}} \epsilon, \tag{F.10}$$

*then $\tilde{x} \triangleq \overline{x} - \alpha(A^\top \tilde{\xi} + \tilde{u})$ satisfies*

$$\tilde{x} \approx \arg\min_{x \in \mathbb{R}^d} \left\{ H(Ax) + p(x) + \frac{1}{2\alpha} \|x - \overline{x}\|_2^2 \right\} \quad \textit{where} \quad \|\tilde{x} - \mathrm{prox}_{\alpha \Phi}(\overline{x})\|_2 \le \epsilon.$$

By using Claim F.3 with $A = A_{S_k^{1:m}, :}$ and functions $H$ and $p$ defined in (F.2), we can deduce from the 1-strong convexity of $H^\star$ (see (F.4)) that

$$\left\| x_{k+1} - \mathrm{prox}_{\alpha_k \overline{\varphi}(\cdot; S_k^{1:m})}(x_k) \right\|_2 = \left\| x_{k+1} - \mathrm{prox}_{\tilde{\alpha}_k \Phi}(x_k) \right\|_2 \le \epsilon_k,$$

where $\Phi(x) \triangleq H(A_{S_k^{1:m}, :} x) + p(x)$ for all $x \in \mathbb{R}^d$. It only remains to prove Claim F.3.

*Proof of Claim F.3.* Let $\hat{x} \triangleq \mathrm{prox}_{\alpha \Phi}(\overline{x})$. We assume that $(\tilde{\xi}, \tilde{u}) \in \mathbb{R}^m \times \mathbb{R}^d$ satisfy (F.10) and set $\tilde{x} \triangleq \overline{x} - \alpha \left( A^\top \tilde{\xi} + \tilde{u} \right)$. By following an argument nearly identical to that in the proof of (Rockafellar, 1976a, Proposition 6), we can obtain

$$\|\hat{x} - \tilde{x}\|_2^2 \le 2\alpha \cdot \left( \Psi \left( \tilde{\xi} \right) - \inf_{\xi \in \mathbb{R}^m} \Psi \left( \xi \right) \right). \tag{F.11}$$

Given that $H^\star$ is $\eta_H$-strongly convex, $\Psi$ is also $\eta_H$-strongly convex. Thus,

$$\inf_{\xi \in \mathbb{R}^m} \Psi \left( \xi \right) \ge \inf_{\xi \in \mathbb{R}^m} \left\{ \Psi \left( \tilde{\xi} \right) + \left\langle \nabla \Psi \left( \tilde{\xi} \right), \xi - \tilde{\xi} \right\rangle + \frac{\eta_H}{2} \left\| \xi - \tilde{\xi} \right\|_2^2 \right\} = \Psi \left( \tilde{\xi} \right) - \frac{1}{2\eta_H} \left\| \nabla \Psi \left( \tilde{\xi} \right) \right\|_2^2.$$

This property together with (F.11) implies

$$\|\hat{x} - \tilde{x}\|_2 \le \sqrt{2\alpha \left( \Psi \left( \tilde{\xi} \right) - \inf_{\xi \in \mathbb{R}^m} \Psi \left( \xi \right) \right)} \le \sqrt{2\alpha \cdot \frac{1}{2\eta_H} \left\| \nabla \Psi \left( \tilde{\xi} \right) \right\|_2^2} \le \epsilon,$$

where the last inequality follows from the definition of $\tilde{\xi}$ in (F.10). This completes the proof of Claim F.3. For completeness, we provide a detailed proof of (F.11) below. Let $L_\alpha$ denote the augmented Lagrangian function:

$$L_\alpha(\xi, u, x) \triangleq H^\star(\xi) + p^\star(u) - \left\langle A^\top \xi + u, x \right\rangle + \frac{\alpha}{2} \left\| A^\top \xi + u \right\|_2^2$$

$$\text{for all } (\xi, u, x) \in \mathbb{R}^m \times \mathbb{R}^d \times \mathbb{R}^d.$$

Note that

$$\nabla_x L_\alpha(\tilde{\xi}, \tilde{u}, \overline{x}) = - \left( A^\top \tilde{\xi} + \tilde{u} \right) = \frac{\tilde{x} - \overline{x}}{\alpha}.$$

Then, the concavity of $L_\alpha(\tilde{\xi}, \tilde{u}, \cdot)$ leads us to

$$L_\alpha \left( \tilde{\xi}, \tilde{u}, \overline{x} \right) + \alpha^{-1} \left\langle \tilde{x} - \overline{x}, x - \overline{x} \right\rangle \ge L_\alpha \left( \tilde{\xi}, \tilde{u}, x \right) \ge \inf_{(\xi, u) \in \mathbb{R}^m \times \mathbb{R}^d} L_\alpha \left( \xi, u, x \right). \tag{F.12}$$

Fix any $x \in \mathbb{R}^d$. It is easily verified that the maximization problem

$$\max_{y \in \mathbb{R}^d} -H^{**}(Ay) - p^{**}(y) - \frac{1}{2\alpha} \|y - x\|_2^2$$

is the dual of the following minimization problem:

$$\min_{(\xi, u) \in \mathbb{R}^m \times \mathbb{R}^d} H^\star(\xi) + p^\star(u) - \langle A^\top \xi + u, x \rangle + \frac{\alpha}{2} \|A^\top \xi + u\|_2^2$$

$$\Leftrightarrow \quad \min_{(\xi, u, w) \in \mathbb{R}^m \times \mathbb{R}^d \times \mathbb{R}^d} H^\star(\xi) + p^\star(u) - \langle w, x \rangle + \frac{\alpha}{2} \|w\|_2^2 \tag{F.13}$$

$$\text{s.t.} \quad A^\top \xi + u - w = 0$$

Note that the objective function of problem (F.13) is proper, closed and strongly convex. Then, it is bounded below over a nonempty feasible set, ensuring that the optimal value of problem (F.13) is finite. By combining this property with the fact that problem (F.13) is convex and satisfies Slater's condition, we conclude that strong duality holds, i.e.,

$$\inf_{(\xi, u) \in \mathbb{R}^m \times \mathbb{R}^d} L_\alpha(\xi, u, x) = \sup_{y \in \mathbb{R}^d} \left\{ -H^{**}(Ay) - p^{**}(y) - \frac{1}{2\alpha} \|y - x\|_2^2 \right\}.$$

Based on this result, and given that both $H$ and $p$ are proper and closed convex, we deduce that

$$\inf_{(\xi, u) \in \mathbb{R}^m \times \mathbb{R}^d} L_\alpha(\xi, u, x) = \sup_{y \in \mathbb{R}^d} \left\{ -H(Ay) - p(y) - \frac{1}{2\alpha} \|y - x\|_2^2 \right\}$$

$$= \sup_{y \in \mathbb{R}^d} \left\{ -\Phi(y) - \frac{1}{2\alpha} \|y - x\|_2^2 \right\} \quad \text{for all } x \in \mathbb{R}^d. \tag{F.14}$$

Combining (F.12) with (F.14) yields

$$L_\alpha \left( \tilde{\xi}, \tilde{u}, \overline{x} \right) \geq -\Phi(\hat{x}) - \frac{1}{2\alpha} \|\hat{x} - x\|_2^2 - \alpha^{-1} \langle \tilde{x} - \overline{x}, x - \overline{x} \rangle \quad \text{for all } x \in \mathbb{R}^d. \tag{F.15}$$

Similarly, we can obtain

$$\inf_{(\xi, u) \in \mathbb{R}^m \times \mathbb{R}^d} L_\alpha(\xi, u, \overline{x}) = \sup_{y \in \mathbb{R}^d} \left\{ -\Phi(y) - \frac{1}{2\alpha} \|y - \overline{x}\|_2^2 \right\}$$

$$= -\inf_{y \in \mathbb{R}^d} \left\{ \Phi(y) + \frac{1}{2\alpha} \|y - \overline{x}\|_2^2 \right\} \tag{F.16}$$

$$= -\Phi(\hat{x}) - \frac{1}{2\alpha} \|\hat{x} - \overline{x}\|_2^2,$$

where the first equality comes from (F.14) with $x = \overline{x}$, and the last follows from the definition of $\hat{x}$. On the other hand, it follows from the definition of $\Psi$ and $\tilde{u}$ that

$$\inf_{(\xi, u) \in \mathbb{R}^m \times \mathbb{R}^d} L_\alpha(\xi, u, \overline{x})$$

$$= \inf_{\xi \in \mathbb{R}^m} \inf_{u \in \mathbb{R}^d} \left\{ H^\star(\xi) + p^\star(u) - \langle A^\top \xi + u, \overline{x} \rangle + \frac{\alpha}{2} \|A^\top \xi + u\|_2^2 \right\}$$

$$= \inf_{\xi \in \mathbb{R}^m} \inf_{u \in \mathbb{R}^d} \left\{ H^\star(\xi) + p^\star(u) + \frac{\alpha}{2} \left\| A^\top \xi + u - \frac{\overline{x}}{\alpha} \right\|_2^2 - \frac{1}{2\alpha} \|\overline{x}\|_2^2 \right\}$$

$$= \inf_{\xi \in \mathbb{R}^m} \left\{ H^\star(\xi) + p^\star \left( \text{prox}_{\frac{p^\star}{\alpha}} \left( \frac{\overline{x}}{\alpha} - A^\top \xi \right) \right) + \frac{\alpha}{2} \left\| A^\top \xi + \text{prox}_{\frac{p^\star}{\alpha}} \left( \frac{\overline{x}}{\alpha} - A^\top \xi \right) - \frac{\overline{x}}{\alpha} \right\|_2^2 \right\}$$

$$- \frac{1}{2\alpha} \|\overline{x}\|_2^2$$

$$= \inf_{\xi \in \mathbb{R}^m} \Psi(\xi) - \frac{1}{2\alpha} \|\overline{x}\|_2^2$$

and

$$L_\alpha \left( \tilde{\xi}, \tilde{u}, \overline{x} \right) = H^\star(\tilde{\xi}) + p^\star(\tilde{u}) + \frac{\alpha}{2} \left\| A^\top \tilde{\xi} + \tilde{u} - \frac{\overline{x}}{\alpha} \right\|_2^2 - \frac{1}{2\alpha} \|\overline{x}\|_2^2 = \Psi(\tilde{\xi}) - \frac{1}{2\alpha} \|\overline{x}\|_2^2.$$

Then, we can deduce from the previous two equations that

$$\Psi\left(\tilde{\xi}\right) - \inf_{\xi \in \mathbb{R}^m} \Psi\left(\xi\right) = L_\alpha\left(\tilde{\xi}, \tilde{u}, \overline{x}\right) - \inf_{(\xi,u)\in\mathbb{R}^m\times\mathbb{R}^d} L_\alpha\left(\xi, u, \overline{x}\right),$$

which combined with (F.15) and (F.16) implies

$$\Psi\left(\tilde{\xi}\right) - \inf_{\xi \in \mathbb{R}^m} \Psi\left(\xi\right)$$

$$\geq \left[-\Phi\left(\hat{x}\right) - \frac{1}{2\alpha}\|\hat{x} - x\|_2^2 - \alpha^{-1}\langle\tilde{x} - \overline{x}, x - \overline{x}\rangle\right] - \left[-\Phi(\hat{x}) - \frac{1}{2\alpha}\|\hat{x} - \overline{x}\|_2^2\right]$$

$$= \frac{1}{2\alpha}\left(\|\hat{x} - \overline{x}\|_2^2 - \|\hat{x} - x\|_2^2 - 2\langle\tilde{x} - \overline{x}, x - \overline{x}\rangle\right) \quad \text{for all } x \in \mathbb{R}^d.$$

Hence, the inequality (F.11) can be derived from the aforementioned results as follows:

$$\Psi\left(\tilde{\xi}\right) - \inf_{\xi \in \mathbb{R}^m} \Psi\left(\xi\right) \geq \sup_{x\in\mathbb{R}^d}\left\{\frac{1}{2\alpha}\left(\|\hat{x} - \overline{x}\|_2^2 - \|\hat{x} - x\|_2^2 - 2\langle\tilde{x} - \overline{x}, x - \overline{x}\rangle\right)\right\}$$

$$= \frac{1}{2\alpha}\left(\|\hat{x} - \overline{x}\|_2^2 - \|\tilde{x} - \overline{x}\|_2^2 - 2\langle\tilde{x} - \overline{x}, \hat{x} - \tilde{x}\rangle\right)$$

$$= \frac{1}{2\alpha}\|\hat{x} - \tilde{x}\|_2^2.$$

$\square$

