# OpenReview forum: "A Tight Convergence Analysis of Inexact Stochastic Proximal Point Algorithm for Stochastic Composite Optimization Problems"
_ICLR.cc/2025/Conference — ICLR 2025 Poster_

### Official Review · Reviewer_eTPG · 2024-11-01

**Soundness:** 4
**Presentation:** 3
**Contribution:** 4
**Rating:** 8
**Confidence:** 4

**Summary:**

This paper proposes an inexact stochastic proximal point algorithm. It gives conditions that the inexact subproblem solutions should satisfy, and analyzes the convergence of the algorithm based on the conditions being satisfied. Specifically, the paper shows the convergence rate of the distance of current iterate to the optimal solution set is O(k^{-\beta}), and the convergence rate of the residual of current iterate to the exact proximal solution is O(k^{-\beta/2}), where \beta \in (0,1] defines the diminishing stepsize. The experimental results align with the theoretical convergence rates for different values of \beta.

**Strengths:**

The contribution seems very nice and, as far as I can see, the analysis appears to be sound.

**Weaknesses:**

1) The paper does not present a general algorithm for solving the proximal subproblem inexactly to satisfy SCA for arbitrary problems. Although Appendix F.1 provides an algorithm, it is specifically designed for problem (4.1).
2) There is no computation cost and convergence comparison between sPPA and isPPA.

**Questions:**

1) Line 219: equation (2.2). Why does Criterion SCB imply Criterion SCA?
2) \sup_k ||x_k||_2 < \infinity is a condition in Theorem 2.3 where constant stepsize is used. I’m curious what is the challenge of proving sup_k ||x_k||_2 < \infinity when using constant stepsize.
3) In Section 2.3, why is the distance ||x - prox_\phi (x)||_2 referred to as the KKT residual?
4) In the experiment setting Line 479, for the  accuracy parameter \epsilon_k = \gamma \alpha_k^2, what is the practical rule for tuning \gamma?
5) Theorem 2.4 uses \epsilon_k = \gamma \alpha_k^{3/2}, which is different from the experiment setting. Does it make a difference? Does Theorem 2.4 specifically rely on the exponent 3/23/2?

---

> ### Author Response · Authors · 2024-11-26
> **Response to Weakness 1 of Reviewer eTPG**
>
> We thank the reviewer for acknowledging the soundness, quality of
> presentation and significance of the contributions of this work. In what
> follows, we address the weaknesses and answer the questions raised by
> the reviewer.
>
> ------
>
> # Weaknesses:
>
> ------
>
> > **W1:** _a general algorithm for solving the proximal subproblem inexactly to satisfy (SCA) for arbitrary problems._
>
> Thank you for your comment. We would like to give some of
>     our thoughts on this matter.
>
>   -   **Depend on the problem structure and user preferences.** This
>         concern raised by the reviewer is indeed important. However, the
>         choice of an algorithm for solving the subproblem often depends
>         on the specific problem structure and user preferences. Thus, we
>         believe it is more appropriate to leave this choice to the user
>         rather than prescribing a general algorithm. Additionally, we
>         note that under our assumptions, the subproblem is strongly
>         convex, allowing a wide range of efficient algorithms to be
>         employed.
>
>   -   **How to obtain an approximate solution satisfying (SCA).** By
>         (2.2) in Line 219 of the revised version, even if the explicit
>         formulation of
>         $\mathrm{prox}_{\alpha_k \overline{\varphi}(\cdot;S_k^{1:m})}(x_k)$
>         is unavailable (a common scenario in practice), an approximate
>         solution can still be obtained by finding an iterate that
>         satisfies Criterion (SCB) or Criterion (SCC).
>
>       -   **Use Criterion (SCB).** To achieve an iterate $x_{k+1}$
>             satisfying (SCB), if the value
>             $\Phi\_{\alpha_k,x_k,S\_k\^{1:m}}\^{\*}$ is unknown, we can use
>             the dual of the subproblem to compute a lower bound on
>             $\Phi\_{\alpha_k,x_k,S\_k\^{1:m}}^{\*}$, denoted as
>             $\hat{\Phi}^{\*}\_{\alpha_k,x_k,S_k^{1:m}}$. Then, using
>             (2.2), it suffices to obtain an iterate $x_{k+1}$ satisfying
>             $$\Phi\_{\alpha_k,x_k}(x_{k+1};S_k^{1:m}) - \hat{\Phi}^{*}_{\alpha_k,x_k,S_k^{1:m}} \leq \frac{\epsilon_k^2}{2\alpha_k}.$$
>
>       -   **Use Criterion (SCC).** Similarly, if the subdifferential
>             $\partial \Phi_{\alpha_k,x_k}(\cdot;S_k^{1:m})$ is
>             accessible, an iterate $x_{k+1}$ satisfying (SCA) can be
>             obtained by ensuring (SCC) is met.
>
>    Finally, we emphasize that the main focus of this paper is on
>     analyzing the convergence rate of isPPA. The specific algorithmic
>     details for solving the subproblem are therefore beyond the scope of
>     this work.

---

> ### Author Response · Authors · 2024-11-26
> **Response to Weakness 2 of Reviewer eTPG**
>
> ------
>
> > **W2:** _computation cost and convergence comparison between sPPA and isPPA._
>
> Thank you for your comment. Below is the comparison
>     between sPPA and isPPA.
>
>    -   **An additional term $O(\gamma^2)$ involved in the dominant term
>         of the convergence rate for isPPA.** Note that our derived
>         results apply to sPPA as a special case by setting $\gamma = 0$.
>
>         -   For isPPA with constant stepsizes, the noise term in
>             inequality (2.8) (Line 289 in the revised version) is
>             proportional to $\tilde{C}_{f,m,U,c_1,\gamma}(\alpha_0)$.
>
>         -   For isPPA with diminishing stepsizes, as demonstrated in the
>             proof of Corollary 2.5 in Appendix C.3, the dominant term in
>             inequality (2.9) (Lines 300-306 in the revised version) is
>             the second part $\mathcal{O}(k^{-\beta})$, whose coefficient
>             is also proportional to
>             $\tilde{C}_{f,m,U,c_1,\gamma}(\alpha_0)$.
>
>         Using the definition of $\tilde{C}_{f,m,U,c_1,\gamma}(\alpha)$
>         in equation (2.7) (Line 279 in the revised version), we can
>         conclude from the discussion above that, compared to sPPA, an
>         additional term $O(\gamma^2)$ is introduced into the constant of
>         the asymptotic bound on the convergence rate for isPPA.
>
>    -   **Comparable asymptotic overall complexity when subproblem can
>         be solved efficiently.** Now, suppose that the subproblem at
>         iteration $k$ is solved by an algorithm $\mathcal{M}$ satisfying
>         $$\Phi_{\alpha_k,x_k}(x_k^{(t)};S_k^{1:m}) - \Phi_{\alpha_k,x_k,S_k^{1:m}}^{*}\leq C_1 (1-\rho\_{\mathcal{M}}\^{(k)})^t$$
>         for some constant $C_1$ and $\rho\_{\mathcal{M}}\^{(k)}\in (0,1)$
>         (see Line 215 in the revised version for the definition of
>         $\Phi_{\alpha_k,x_k}(\cdot;S_k^{1:m})$).
>
>         -   **Overall complexity
>             $\mathcal{O}((\frac{1}{\epsilon})^{\frac{1}{\beta}}\log(\frac{1}{\epsilon}))$
>             for isPPA.** Denote by $\alpha_k = \alpha_0 k^{-\beta}$ ($\alpha_0 > 0$,
>             $\beta\in (0,1])$) and
>             $\epsilon_k = \gamma \alpha_k^{\delta}$ ($\gamma > 0$,
>             $\delta\geq \frac{3}{2}$). Let $N_{\mathrm{inner}}(k)$ denote the
>             inner-loop complexity bound of $\mathcal{M}$ for solving the
>             subproblem at iteration $k$. By (SCB), we have
>             $$\begin{aligned}
>                             N_{\mathrm{inner}}(k) \leq \inf\left\lbrace  t\in \mathbb{Z}_{+}\ \vert\ C_1 e^{-\rho\_{\mathcal{M}}\^{(k)} t} \leq \frac{\epsilon_k^2}{2\alpha_k}\right\rbrace = \left\lceil \frac{1}{\rho\_{\mathcal{M}}\^{(k)}} \log\left(\frac{2C_1}{\gamma^2} \alpha_k^{1-2\delta}\right) \right\rceil = \left\lceil \frac{1}{\rho\_{\mathcal{M}}\^{(k)}} \log\left(\frac{2C_1}{\gamma^2\alpha_0^{2\delta - 1}} k^{(2\delta-1)\beta}\right) \right\rceil.
>                         \end{aligned}$$
>
>               - Assume that
>             $\rho\_{\mathcal{M}}\^{(k)} \geq C_2 \alpha_k^{-\eta}$ for
>             some $C_2 > 0$ and $\eta\in [0,1]$ (e.g., $\eta = 1$ if
>             $\mathcal{M}$ is the gradient descent (GD) method, assuming
>             the problem satisfies the conditions outlined in Common
>             Response 3). Then, the iteration complexity for solving the
>             subproblem at iteration $k$ is upper bounded by
>             $$\begin{aligned}
>                             N_{\mathrm{inner}}(k) &\leq \left\lceil A_1 k^{-\eta\beta} + A_2 k^{-\eta\beta} \log(k)\right\rceil
>                         \end{aligned}$$
>                   where $A_1 \triangleq C_2^{-1}\alpha_0^{\eta} (\log(2C_1\gamma^{-2}) + (1-2\delta) \log(\alpha_0)$ and
>             $A_2 \triangleq C_2^{-1}\alpha_0^{\eta} (2\delta - 1) \beta$.
>               - Let $N_{\mathrm{outer}}$ denote the number of iterations of
>             isPPA required to generate an iterate
>             $x_{N_{\mathrm{outer}}}$ such that
>             $\mathbb{E}[\mathrm{dist}(x_k,\mathcal{X}^{*})^2] \leq \epsilon$
>             for some $\epsilon>0$. Under the setting of Corollary 2.5,
>             we have
>             $$N_{\mathrm{outer}} \leq \mathcal{O}\left(\left(\frac{1}{\epsilon}\right)^{\frac{1}{\beta}}\right).$$
>             Using inequalities (C.7) and (C.9b) from Lemma C.4 in our
>             submitted manuscript, we can deduce that the overall
>             complexity $N_{\mathrm{total}}$ satisfies $$\begin{aligned}
>                             N_{\mathrm{total}} \leq \sum_{k=1}^{N_{\mathrm{outer}}} N_{\mathrm{inner}}(k) \leq 2\zeta_{1-\eta\beta}(N_{\mathrm{outer}}+1)\cdot \left(\frac{A_1}{2^{\eta\beta}} + \frac{A_2}{2^{\eta\beta}}\log\left(N_{\mathrm{outer}}\right)\right) \leq \mathcal{O}\left(N_{\mathrm{outer}}\log\left(N_{\mathrm{outer}}\right)\right) \leq \mathcal{O}\left(\left(\frac{1}{\epsilon}\right)^{\frac{1}{\beta}}\log\left(\frac{1}{\epsilon}\right)\right).
>                         \end{aligned}$$

---

> > ### Author Response · Authors · 2024-11-26
> > **Response to Weakness 2 of Reviewer eTPG (continued)**
> >
> > ------
> >
> > -   **Comparable asymptotic overall complexity when subproblem can
> >         be solved efficiently.** *(continued)*
> >
> >        -   **Overall complexity
> >             $\mathcal{O}((\frac{1}{\epsilon})^{\frac{1}{\beta}})$ for
> >             sPPA.** Since sPPA assumes that the exact solution is
> >             obtained for each subproblem, its overall complexity reduces
> >             to the iteration complexity of the outer loop. By Corollary
> >             2.5, the overall complexity satisfies
> >             $$N_{\mathrm{overall}} \leq \mathcal{O}\left(\left(\frac{1}{\epsilon}\right)^{\frac{1}{\beta}}\right).$$
> >
> >        Leveraging the fact that $\log(k) = o(k^\xi)$ for any $\xi > 0$,
> >         we conclude that the overall complexity of isPPA is
> >         asymptotically comparable to that of sPPA.
> >
> > **Trade-off when selecting $\gamma$.** While decreasing $\gamma$
> >     improves the convergence rate of isPPA, obtaining exact solutions or
> >     solutions with high accuracy (i.e., small $\gamma$) can
> >     significantly increase the computational cost of solving
> >     subproblems. Therefore, a trade-off must be carefully considered in
> >     practice.

---

> ### Author Response · Authors · 2024-11-26
> **Response to Questions 1-3 of Reviewer eTPG**
>
> ------
>
> # Question:
>
> ------
>
> > **Q1:** _why does Criterion SCB imply Criterion SCA?_
>
> Thank you for your question. Denote by
>     $\tilde{x}\_{k+1} \triangleq \mathrm{prox}\_{\alpha_k\overline{\varphi}(\cdot;S_k^{1:m})}(x_k)$.
>     Under Assumption 1, $\tilde{x}\_{k+1}$ is well-defined and the
>     function $\Phi_{\alpha_k,x_k}(\cdot;S_k^{1:m})$ is
>     $\frac{1}{\alpha_k}$-strongly convex. Consequently, for any
>     $x\in \mathbb{R}^d$, we have $$\begin{aligned}
>             \Phi_{\alpha_k,x_k}(x;S_k^{1:m}) \geq \Phi_{\alpha_k,x_k}(\tilde{x}\_{k+1};S_k^{1:m}) + \langle0, x - \tilde{x}\_{k+1}\rangle + \frac{1}{2\alpha_k}\Vert x - \tilde{x}\_{k+1} \Vert_2^2 = \Phi_{\alpha_k,x_k,S_k^{1:m}}^{\*} + \frac{1}{2\alpha_k}\Vert x - \tilde{x}\_{k+1} \Vert_2^2,
>         \end{aligned}$$ where the inequality follows from
>     $0\in \partial \Phi_{\alpha_k,x_k}(\tilde{x}\_{k+1};S_k^{1:m})$
>     together with the strong convexity of
>     $\Phi_{\alpha_k,x_k}(\cdot;S_k^{1:m})$, and the equality holds
>     because
>     $\Phi_{\alpha_k,x_k}(\tilde{x}\_{k+1};S_k^{1:m}) = \Phi_{\alpha_k,x_k,S_k^{1:m}}^{\*}$.
>
>  -   **Criterion (SCB) $\Rightarrow$ Criterion (SCA).** If $x_{k+1}$
>         satisfies (SCB), it follows from the above inequality that
>         $$\Vert x_{k+1} - \tilde{x}\_{k+1} \Vert_2 \leq \sqrt{2\alpha_k \left(\Phi_{\alpha_k,x_k}(x_{k+1};S_k^{1:m}) - \Phi_{\alpha_k,x_k,S_k^{1:m}}^{\*}\right)} \leq \epsilon_k.$$
>         Thus, by definition, $x_{k+1}$ also satisfies (SCA).
>
> ------
>
> > **Q2:** _challenge of proving $\sup_k \Vert x_k\Vert_2 < \infty$ when using constant stepsize._
>
> Thank you for your question. The almost sure boundedness
>     established in Theorem 2.1 relies on the supermartingale convergence
>     lemma, which requires the stepsize sequence $\lbrace \alpha_k\rbrace$ to be
>     square summable. Therefore, when constant stepsizes are used for
>     isPPA, the supermartingale convergence lemma is no longer
>     applicable. Consequently, the proof techniques employed in this
>     paper are insufficient to establish the almost sure boundedness of
>     $\lbrace x_k\rbrace$ under the proposed assumptions in such cases.
>
>  ------
>
> > **Q3:** _why is the distance $\Vert x - \mathrm{prox}\_\phi (x)\Vert_2$ referred to as the KKT residual?_
>
> Thank you for your question. Consider the setting of Lemma 2.6.
>
> -   **KKT conditions.** The minimization problem $\min \phi(x)$ can
>         be reformulated as $$\begin{aligned}
>                         \min\ \phi(y)\quad \mbox{s.t.}\quad y = x,\ x,y\in \mathbb{R}^d,
>                     \end{aligned}$$ with the associated KKT system given
>         by $$\begin{aligned}
>                     \hspace{0em}v = 0,\quad 0 \in \partial \phi(y) -v,\quad x = y\quad
>                     \Leftrightarrow\quad v = 0,\quad 0\in \partial \phi(y) + y - (y+v),\quad x = y
>                 \end{aligned}$$ $$\begin{aligned}
>                     \hspace{14.7em}\overset{(\star)}{\Leftrightarrow}\quad v = 0,\quad 0\in \partial \left(\phi(\cdot) + \frac{1}{2}\Vert \cdot - (y+v) \Vert_2^2\right)(y),\quad x = y
>                 \end{aligned}$$ $$\begin{aligned}
>                     \hspace{14.7em}\Leftrightarrow\quad v = 0,\quad y = \mathrm{prox}\_{\phi}(y+v),\quad x = y,
>                 \end{aligned}$$ where $(\star)$ uses the assumption that
>         $\phi$ is proper closed and convex.
>
> -   **KKT residual.** Define mappings
>         $$\Theta\_{KKT}(x,y,v) \triangleq \begin{pmatrix}
>                     v & y - \mathrm{prox}\_\phi(y+v) & x - y
>                     \end{pmatrix}\quad \forall\ (x,y,v)\in \mathbb{R}^d \times \mathbb{R}^d \times \mathbb{R}^d,$$
>         and
>         $$\mathrm{Res}(x) \triangleq x - \mathrm{prox}\_\phi(x)\quad \forall\ x\in \mathbb{R}^d.$$
>         The KKT conditions imply that
>         $$\overline{x}\in \arg\min_{x\in \mathbb{R}^d}\phi(x)\quad \Leftrightarrow\quad \Theta_{KKT}(\overline{x},\overline{y},\overline{v}) = 0\ \mathrm{for\ some}\ \overline{y},\overline{v}\in \mathbb{R}^d\quad \Leftrightarrow\quad \mathrm{Res}(\overline{x}) = 0,$$
>         Hence, we refer to *$\Vert x - \mathrm{prox}_\phi (x)\Vert_2$*
>         as the *KKT residual*.

---

> ### Author Response · Authors · 2024-11-26
> **Response to Questions 4-5 of Reviewer eTPG**
>
> ------
>
> > **Q4:** _practical rule for tuning $\gamma$ regarding accuracy parameter $\epsilon_k = \gamma \alpha_k^2$._
>
> Thank you for your question. As noted in response to ***W2***, a trade-off
>     must be considered between the cost of solving the subproblem and
>     the convergence rate of the outer loop. Since this work primarily
>     focuses on theoretical convergence analysis, we have not explored
>     this issue in detail. However, we believe this is a valuable
>     direction for future research.
>
> ------
>
> > **Q5:** _accuracy parameter setting in numerical experiments and exponent $\frac{3}{2}$ in Theorem 2.4._
>
> Thank you for your insightful question regarding the
>     choice of $\epsilon_k$.
>
> -   **Theorem 2.4 holds for $\epsilon_k = \gamma \alpha_k^{\delta}$
>         with $\gamma>0$ and $\delta\geq \frac{3}{2}$.** Fix any
>         $\delta\geq \frac{3}{2}$, and let $\lbrace x_k\rbrace$ be generated by
>         isPPA with diminishing stepsizes
>         $\alpha_k = \alpha_0 k^{-\beta}$ and parameter
>         $\epsilon_k = \gamma \alpha_k^{\delta}$, where $\alpha_0>0$,
>         $\beta\in (0,1]$ and $\gamma\geq 0$. Denote by
>         $\tilde{x}\_{k+1} \triangleq \mathrm{prox}\_{\alpha_k\overline{\varphi}(\cdot;S_k^{1:m})}(x_k)$.
>         Since $\alpha_k \leq \alpha_0$, it follows that
>         $$\Vert x_{k+1} - \tilde{x}\_{k+1}\Vert_2 \leq \gamma \alpha_k^{\frac{3}{2}}\cdot \alpha_k^{\delta - \frac{3}{2}} \leq  \gamma\alpha_0^{\delta - \frac{3}{2}}\cdot \alpha_k^{\frac{3}{2}}\quad \forall\ k\in \mathbb{Z}_{+}.$$
>         Therefore, $\lbrace x_k\rbrace$ can be regarded as the iterates generated
>         by isPPA with parameter
>         $\tilde{\epsilon}_k = \tilde{\gamma} \alpha_k^{\frac{3}{2}}$,
>         where
>         $\tilde{\gamma} \triangleq \gamma\alpha_0^{\delta - \frac{3}{2}}$.
>         The results of Theorem 2.4 then apply. Under the condition
>         $\sup_k \Vert x_k \Vert_2 < \infty$, the exponent $\frac{3}{2}$
>         specified in Theorem 2.4 serves as a *lower bound* for the decay
>         rate of $\epsilon_k$.
>
> -   **Reason for setting $\delta = 2$ in numerical experiments.**
>         The choice $\epsilon_k = \gamma \alpha_k^2$ in the numerical
>         experiments ensures compliance with the conditions required for
>         the almost sure boundedness established in Theorem 2.1. This
>         boundedness is also used in the proof of Theorem 2.4 when the
>         statement $\sup_k \Vert x_k\Vert_2 < \infty$ is removed.
>         Additionally, for bounded stepsizes, a similar argument shows
>         that Theorem 2.1 remains valid for
>         $\epsilon_k = \gamma \alpha_k^{\delta}$ with $\gamma > 0$ and
>         $\delta \geq 2$. Thus, for square summable diminishing
>         stepsizes, any exponent $\delta \geq 2$ can be used in numerical
>         experiments.

---

### Official Review · Reviewer_KLe5 · 2024-11-02

**Soundness:** 2
**Presentation:** 3
**Contribution:** 2
**Rating:** 5
**Confidence:** 4

**Summary:**

The paper presents the convergence of an inexact stochastic proximal point algorithm for solving stochastic composite optimization problems. The main contributions lies in weakening the assumptions on the function class and illustrating the tightness of the established bounds.

**Strengths:**

The problem setting, assumptions, and theoretical results are clearly stated, and necessary comparisons with the existing literature are provided.

**Weaknesses:**

1. The applicability of the proposed proximal point algorithm is limited. It requires two loops, and the complexity of the inner loop can also be prohibitive. The complexity of solving the inner problems is not considered in the paper.
2. Deriving the stability result from existing exact PPA algorithms in the literature does not seem particularly challenging. Additionally, given coercivity and the assumption that $\|x_k\|$ is bounded, it would be more natural to use local Lipschitz continuity rather than global Lipschitz continuity. It’s essential to clearly state the technical difficulties involved.

**Questions:**

1. Remark 2.1: The conditions in Remark 1.1 are insufficient to ensure Assumption 2. For example, consider the indicator function of $x \geq 0$; its subgradient at $x = 0$ is a cone, and thus an unbounded set.
2. Theorem 2.3: Is there an achievable high-probability bound on the distance to the optimal set?
3. Could the assumption ${\rm sup}\\|x_k\\|_2 < \infty$ be removed to obtain a cleaner theoretical result?
4. It would be helpful comment on the impact of minibatch size $m$.
5. Numerical experiments: The applications considered are somewhat outdated, and the datasets used are synthetic. Since the theory targets a more general setting where $f$ and the regularizer are not necessarily convex, it would be valuable to see results on more general machine learning problems with this structure.
6. Numerical experiments: The current experiments resemble an ablation study. It would be helpful to include comparisons with other state-of-the-art algorithms for these problems to demonstrate practical applicability.

---

> ### Author Response · Authors · 2024-11-26
> **Response to Weakness 1 of Reviewer KLe5**
>
> We would like to thank the reviewer for the time spent evaluating our
> work and for the detailed feedback. In what follows, we address the
> weaknesses and answer the concerns raised by the reviewer.
>
> ------
>
> # Weaknesses:
>
> ------
>
> > **W1(a):** _applicability and overall efficiency of double-loop algorithm._
>
> Thank you for your comment. Please refer to ***Common
>     Response 2*** and ***Common Response 3*** for details.
>
>  In Common Response 2, we explain our motivation for analyzing the
>     inexact stochastic PPA (isPPA). The primary goal is to establish
>     bounds on the convergence rate of isPPA. This work also highlights a
>     fundamental discrepancy between the deterministic and stochastic
>     versions of the PPA (see Lines 416-435 in the colorbluerevised
>     version).
>
> In Common Response 3, we provide the iteration complexity for the
>     algorithm when using the gradient descent algorithm to solve the
>     subproblem. Additionally, we outline general principles for
>     designing algorithms to solve the subproblem while maintaining the
>     validity of the overall complexity. Since the manuscript is already
>     over 40 pages, we plan to explore this perspective further in future
>     work.
>
> ------
>
> > **W1(b):** _iteration complexity of solving the subproblem not considered._
>
> Thank you for your comment. Please refer to ***Common
>     Response 3***, where we analyze the overall complexity of isPPA. Due to
>     the inclusion of an additional squared $\ell_2$-norm term, each
>     subproblem is strongly convex. This strong convexity significantly
>     facilitates the analysis and ensures efficient algorithms for the
>     inner loop, contributing to the overall complexity of isPPA.

---

> ### Author Response · Authors · 2024-11-26
> **Response to Weakness 2 of Reviewer KLe5**
>
> ------
> > **W2:** _essential to clearly state the technical difficulties involved._
>
> Thank you for your comment.
>
> -   **Technical difficulty in establishing the stability of isPPA.**
>         The stability analysis of isPPA presents unique challenges
>         compared to deterministic inexact PPA. This is because the
>         subproblem in isPPA incorporates the noise term
>         $\overline{\varphi}(\cdot;S_k^{1:m})$ instead of the exact
>         objective function $\phi(\cdot)$. Consequently,
>
>       -   The function value sequence $\lbrace \phi(x_k)\rbrace$ (or
>             $\lbrace \mathbb{E}[\phi(x_k)]\rbrace$) no longer exhibits a
>             nonincreasing property. As a result, the coercivity of
>             $\phi$ alone is insufficient to guarantee the boundedness of
>             iterates $\lbrace x_k\rbrace$.
>
>       -   Unlike the proximal operator
>             $\mathrm{prox}\_{\alpha_k \phi}(\cdot)$ in the deterministic
>             inexact PPA, the operator
>             $\mathrm{prox}\_{\alpha_k \overline{\varphi}(\cdot;S_k^{1:m})}(\cdot)$
>             in isPPA does not generally satisfy the fixed-point property
>             on the optimal solution set. Specifically, there is no
>             guarantee that
>             $\mathrm{prox}\_{\alpha\_k \overline{\varphi}(\cdot;S_k^{1:m})}(x\^{\star})= x\^{\star}$
>             holds for all
>             $x\^{\star}\in \mathcal{X}\^{*}=\arg\min\_{x\in \mathbb{R}\^d} \phi(x)$.
>
>        These observations necessitate the use of techniques distinct
>         from those employed in deterministic settings to establish the
>         almost sure boundedness of the iterates generated by isPPA.
>
> -   **Our contributions extend beyond stability.** In addition to
>         proving the stability of isPPA, the main contributions of this
>         paper include:
>
>        -   **Derivation of upper and lower bounds on the convergence
>             rate of isPPA.**
>
>            -   We provide upper bounds on the convergence rate of isPPA
>                 under some reasonable conditions.
>
>            -   We also derive a matching lower bound using the
>                 regularized Fréchet mean problem, as presented in
>                 Proposition 3.1. Consequently, the derived upper bounds
>                 are shown to be tight up to constant factors.
>
>        -   **Challenges in deriving the lower bound in Proposition
>             3.1.** The key difficulty in deriving Proposition 3.1 lies
>             not in proving it but in constructing an example that
>             simultaneously satisfies the proposed assumptions and
>             ensures that isPPA achieves a lower bound that matches the
>             derived upper bound.
>
>        -   **Insights from Proposition 3.1.** The derived lower bound
>             highlights a gap between the convergence rates of the
>             deterministic PPA and its stochastic variant. Specifically:
>
>            -   Deterministic inexact PPA can achieve linear convergence
>                 under the quadratic growth condition when stepsizes with
>                 positive lower bound are used.
>
>            -   For isPPA, we observed a slower convergence rate in
>                 Theorem 2.3 and 2.4. Initially, it was unclear whether
>                 this discrepancy stemmed from the limitations of isPPA
>                 itself or from the conservativeness of the proposed
>                 upper bound. Proposition 3.1, based on the regularized
>                 Fréchet mean problem (3.1), confirms that this gap
>                 indeed exists.
>
>            These findings provide new insights into the performance
>             differences between deterministic and stochastic PPA.

---

> ### Author Response · Authors · 2024-11-26
> **Response to Questions of Reviewer KLe5**
>
> # Questions:
>
> ------
>
> > **Q1:** _conditions in Remark 1.1 insufficient to ensure Assumption 2._
>
> Thank you for pointing it out. The following ***revised
>     version*** has been included in our manuscript:
>
> -   For the finite-sum regularized regression model with real-valued
>         nonnegative convex component functions and a real-valued convex
>         coercive regularizer, Remark 1.1, combined with the compactness
>         of the subdifferential of a real-valued convex function, implies
>         that Assumption 1 and 2 hold.
>
> ------
>
> > **Q2:** _Theorem 2.3: achievable high-probability bound on the distance to the optimal set._
>
> Thank you for your insightful question. While the
>     theoretical analysis in this paper focuses on convergence guarantees
>     in the expectation sense (which is standard in the literature on
>     stochastic algorithms), establishing high-probability bounds on the
>     rate of convergence is both possible and important. We will explore
>     this problem in our future work.
>
> ------
>
> > **Q3:** _remove assumption $\sup\Vert x_k\Vert_2 < \infty$ to obtain a cleaner theoretical result._
>
> Thank you for your comment.
>
>   -   **Remove condition $\sup_k \Vert x_k\Vert_2 < \infty$ for
>         diminishing stepsizes.** As noted in the discussion following
>         Corollary 2.5, for isPPA with diminishing stepsizes
>         $\alpha_k = \alpha_0 k^{-\beta}$ ($\beta\in (\frac{1}{2},1]$)
>         and $\epsilon_k = \gamma \alpha_k^2$, Theorem 2.1 guarantees the
>         almost sure boundedness of $\lbrace x_k\rbrace$ under Assumption 1 and 2.
>         Consequently, all results in Theorem 2.4 and Corollary 2.5 hold
>         with probability 1, even if the condition
>         $\sup_k \Vert x_k \Vert_2 < \infty$ in Line 296 of the revised
>         version is removed.
>
>   -   **Future work for constant stepsizes.** For isPPA with constant
>         stepsizes, determining whether the almost sure boundedness of
>         $\lbrace x_k\rbrace$ can be established under reasonable assumptions, and
>         how to derive this property, remains an open question for future
>         research.
>
> ------
>
> > **Q4:** _comment on the impact of minibatch size $m$._
>
> Thank you for your question. Roughly speaking, increasing
>     the minibatch size $m$ improves the convergence rate of isPPA.
>     Specifically, from equation (2.4), $\eta_{f,U}\in [0,1]$ and
>     $\rho_{f,m,U} = (\sqrt{1 + \frac{1-\eta_{f,U}}{m}}+1)^2$, making
>     $\rho_{f,m,U}$ a nonincreasing function of $m$. From the definition
>     of $\tilde{C}\_{f,m,U,c_1,\gamma}(\cdot)$ in (2.7), it follows that
>     $\tilde{C}\_{f,m,U,c_1,\gamma}(\alpha)$ decreases with $m$ for a
>     fixed $\alpha$.
>
>    -   **Neighborhood shrinks when using constant stepsizes.** For
>         isPPA with constant stepsizes, this implies that the linear
>         convergence of $\lbrace x_k\rbrace$ to a neighborhood of the optimal
>         solution set improves, as the neighborhood shrinks due to the
>         noise term in (2.8) being proportional to
>         $\tilde{C}\_{f,m,U,c_1,\gamma}(\alpha_0)$.
>
>    -   **Coefficient decreases when using diminishing stepsizes.** For
>         isPPA with diminishing stepsizes, the sublinear convergence of
>         $\lbrace x_k\rbrace$ improves in the sense that the coefficient of the
>         dominant term decreases. For instance, as shown in the proof of
>         Corollary 2.5 in Appendix C.3, the dominant term in (2.9) is the
>         second part $\mathcal{O}(k^{-\beta})$, whose coefficient is
>         proportional to $\tilde{C}\_{f,m,U,c_1,\gamma}(\alpha_0)$.
>
>    **Trade-off when selecting $m$.** We also note that increasing the
>     minibatch size $m$ can lead to higher computational costs for
>     solving the subproblem. Therefore, a trade-off must be considered
>     when selecting $m$.
>
> ------
>
> > **Q5-Q6:** _numerical experiments conducted in this paper._
>
> Thank you for your comments. The primary goal of this
>     paper is to derive a tight convergence rate analysis for isPPA, and
>     the numerical experiments are designed to validate our theoretical
>     findings. Specifically, Proposition 3.1 highlights the gap between
>     the convergence rates of deterministic PPA and its stochastic
>     variants. This is a notable observation, especially considering the
>     common practice in the literature of extending deterministic
>     algorithms to stochastic settings. The numerical experiments
>     presented in this paper demonstrate the convergence estimates
>     derived in Section 3.
>
> We agree that investigating the numerical performance of isPPA on
>     more general machine learning problems, particularly those involving
>     nonconvex component functions or regularizers, would be highly
>     valuable. However, given the length of this manuscript (over 40
>     pages), we plan to explore this direction in future work.

---

### Official Review · Reviewer_Lrjb · 2024-11-03

**Soundness:** 3
**Presentation:** 3
**Contribution:** 2
**Rating:** 6
**Confidence:** 3

**Summary:**

In this paper, the authors consider inexact stochastic proximal point algorithm (isPPA) for solving stochastic composite optimization problems. Under mild conditions, the authors demonstrate the stability and almost sure convergence of this method. By further assuming a local Lipschitz condition on component functions and a quadratic growth condition on the objective function, the authors establish convergence rate guarantees. Empirical validation from numerical experiments supports these theoretical analyses.

**Strengths:**

In this submission, the authors prove the stability of isPPA under mild conditions, extending the stability result for exact sPPA. The authors provided non-asymptotic last-iterate convergence rates for isPPA in terms of the expected squared distance to the optimal solution set. They also provide lower bounds on the convergence rates of isPPA for solving problem under the assumptions proposed in this paper, demonstrating that the derived convergence rate guarantees are tight up to constant factors. All the theoretical analysis are proved by detailed mathematical analysis and empirical results from the numerical experiments are consistent with the theoretical analysis.

**Weaknesses:**

The only weakness of this submission is that the authors should explain more about how the assumptions provided in this paper is "mild" or less restrictive compared with the previous results. And how exactly the results of this paper differ from the results of the previous works. How exactly the assumptions one to five are more general compared with the assumptions in the previous results and how this impact the theoretical proof and analysis.

**Questions:**

The authors should make more remarks about how the assumptions in this submission is weaker than the assumptions from previous papers and compared more about the theoretical analysis in this work with the previous works.

---

> ### Author Response · Authors · 2024-11-26
> **Response to Weakness 1 of Reviewer Lrjb**
>
> Thank you for your time and effort in reviewing our paper and for your
> positive evaluation of the soundness and presentation of our work. In
> what follows, we address the weaknesses and answer the concerns that
> were brought up.
>
> ------
>
> # Weaknesses:
>
> ------
>
> > **W1:** _How the assumptions provided in this paper is \"mild\" or less restrictive compared with the previous results._
>
> Thank you for your comment. A comparison of the
>     assumptions is summarized below, and for a simplified version, we
>     refer to the table provided in the ***Common Response 1***.
>
> -   **Remark on mild conditions.** The statement that the proposed
>         assumptions in this paper are **mild** refers to the fact that
>         the convergence analysis of isPPA is established under
>         conditions without requiring Lipschitz smoothness, strong
>         convexity, or global Lipschitz continuity. This makes the
>         framework and the corresponding convergence guarantees
>         applicable to a wide range of machine learning problems.
>
>   -   **Comparison with the previous results.** Below is a detailed
>         comparison of the assumptions used in the convergence analysis
>         of isPPA with those required by previous results. See Table 1 in
>         Appendix A for additional details.
>
>         -   **Lipschitz smoothness.** Lipschitz smoothness (i.e.,
>             differentiable with Lipschitz continuous gradient) condition
>             on each $f(\cdot;s)$ is required in (Ryu & Boyd, 2014,
>             Theorem 4; Yuan & Li, 2023). This condition is violated by
>             nonsmooth loss functions such as $\Vert Ax - b\Vert_1$.
>
>         -   **Strong convexity.** Results in (Ryu & Boyd, 2014, Theorem
>             4 and 7) require each $f(\cdot;s)$ to satisfy
>             $M_s$-restricted strong convexity with
>             $\lambda\_{\min}(\mathbb{E}\_s[M_s])>0$. In this case, by (Ryu
>             & Boyd, 2014, Lemma 5), $F$ is
>             $\lambda\_{\min}(\mathbb{E}\_s[M_s])$-strongly convex.
>             Similarly, (Patrascu & Necoara, 2018) assumes
>             $\sigma_{f,s}$-strong convexity for $f(\cdot;s)$, where
>             $\sigma_{f,s} \geq 0$, $\mathbb{E}\_s[\sigma_{f,s}] >0$ and
>             $\sup_{s\in \mathcal{S}} \sigma_{f,s} < \infty$. Under this
>             condition, using (Ryu & Boyd, 2014, Lemma 5) and setting
>             $M_s=\sigma_{f,s}I_d$, $F$ is
>             $\mathbb{E}\_s[\sigma\_{f,s}]$-strongly convex. In (Davis &
>             Drusvyatskiy, 2019, Theorem 4.5), $f(\cdot;s)+r(\cdot)$ is
>             required to be $\mu$-strongly convex for some $\mu>0$, which
>             implies $\phi$ is $\mu$-strongly convex. These conditions
>             are violated, for instance, by the quadratic loss function
>             $\frac{1}{2}\Vert Ax - b\Vert_2^2$ when
>             $\mathrm{rank}(A) < d$.
>
>         -   **Global Lipschitz continuity on the entire domain.**
>             Lipschitz continuity on $f(\cdot;s)$ over an open convex set
>             $U$ containing $\mathrm{dom}(r)$ is assumed in (Davis &
>             Drusvyatskiy, 2019, Theorem 4.3, 4.4 and 4.5). The
>             convergence results in (Yuan & Li, 2023, Theorem 10 and 13)
>             for inexact minibatch sPPA require Lipschitz continuity
>             condition on the regularizer $r$ over the entire domain
>             $\mathrm{dom}(\phi)$. These conditions are violated by
>             regularized regression models with quadratic loss
>             $F(x) = \frac{1}{2}\Vert Ax - b\Vert_2^2$ and elastic net
>             regularizer
>             $r(x) = \lambda_1 \Vert x\Vert_1 + \frac{\lambda_2}{2}\Vert x \Vert_2^2$,
>             where $\mathrm{dom}(r) = \mathrm{dom}(\phi) = \mathbb{R}^d$.
>
>         -   **Easy optimization problems.** Linear convergence rate
>             derived in (Asi & Duchi, 2018, Proposition 4.3 and 4.4;
>             Patrascu, 2021, Corollary 4.5) require that all (composite)
>             component functions share global minimizers, i.e., $\phi$ is
>             \"easy to optimize\". This condition is violated by the
>             regularized Freéchet mean problem (4.1) if there are two
>             distinct points $p_{i_0}$ and $p_{j_0}$ for
>             $i_0\neq j_0\in [n]$ (see Line 407 in the revised version).
>             In addition, this condition is also violated by most
>             regularized regression models due to the variability between
>             data instances.
>
>         -   **Subproblem solved exactly.** Convergence analysis in (Ryu
>             & Boyd, 2014; Toulis et al, 2016; Patrascu & Necoara, 2018;
>             Asi & Duchi, 2019; Davis & Drusvyatskiy, 2019; Patrascu,
>             2021; Yuan & Li, 2023, Theorem 1 and 5) assumes exact
>             solutions for subproblems.

---

> ### Author Response · Authors · 2024-11-26
> **Response to Weakness 1 of Reviewer Lrjb (continued)**
>
> The convergence analysis of isPPA in this paper relies on the
>         following assumptions, which are mild and more practical
>         compared to prior works.
>
>   -  **Assumption 1 and 2.** These are satisfied by commonly used
>             (finite-sum) regression models with coercive regularizers
>             (see Remark 2.1 in our manuscript),
>
> -   **Assumption 3.** The local Lipschitz condition on component
>             functions $f(\cdot;s)$ is weaker than Lipschitz smoothness
>             and global Lipschitz condition.
>
> -   **Assumption 4 and 5.** The quadratic growth condition on
>             $\phi$ is weaker than strong convexity.
>
> -   **Inexact subproblem solving.** Each subproblem is solved
>             inexactly according to Criterion (SCA), which is more
>             practical than assuming exact solutions for subproblems.
>
> Based on the discussion above, the assumptions made in the
>         convergence analysis for isPPA in this paper are reasonable and
>         mild.
>
> ------
>
> **References**
>
>    Hilal Asi and John C Duchi. Stochastic (approximate) proximal point methods: Convergence, optimality, and adaptivity. *SIAM Journal on Optimization*, 29(3):2257-2290, 2019.
>
>    Damek Davis and Dmitriy Drusvyatskiy. Stochastic model-based minimization of weakly convex functions. *SIAM Journal on Optimization*, 29(1):207-239, 2019.
>
>    Andrei Patrascu. New nonasymptotic convergence rates of stochastic proximal point algorithm for stochastic convex optimization. *Optimization*, 70(9):1891-1919, 2021.
>
>    Andrei Patrascu and Ion Necoara. Nonasymptotic convergence of stochastic proximal point methods for constrained convex optimization. *Journal of Machine Learning Research*, 18(198):1-42, 2018.
>
>    Ernest K Ryu and Stephen Boyd. Stochastic proximal iteration: a non-asymptotic improvement upon stochastic gradient descent. *Author website*, 2014. URL https://web.stanford.edu/ boyd/papers/spi.html.
>
>    Panos Toulis, Dustin Tran, and Edo Airoldi. Towards stability and optimality in stochastic gradient descent. In *Artificial Intelligence and Statistics*, pp. 1290-1298. PMLR, 2016.
>
>    Xiao-Tong Yuan and Ping Li. Sharper analysis for minibatch stochastic proximal point methods: Stability, smoothness, and deviation. *Journal of Machine Learning Research*, 24(270):1-52, 2023.

---

> ### Author Response · Authors · 2024-11-26
> **Response to Weakness 2 of Reviewer Lrjb**
>
> ------
>
> > **W2:** _How exactly the results of this paper differ from the results of the previous works._
>
>  Thank you for your comments. Please refer to ***Common
>     Response 1*** for a detailed comparison with existing works, including
>     examples that highlight the differences in assumptions and
>     requirements for the subproblem in convergence analysis.
>
> In addition, we would like to highlight that a key contribution of
>     this paper is the derivation of a lower bound on the convergence
>     rate of isPPA, as presented in Proposition 3.1. As stated in Lines
>     416-435 of the revised version, this lower bound indicates that
>     there is a gap in terms of convergence rate between PPA and isPPA,
>     and confirms that the derived upper bound for isPPA with diminishing
>     stepsizes is tight up to constant factors.

---

> ### Author Response · Authors · 2024-11-26
> **Response to Weakness 3 of Reviewer Lrjb**
>
> ------
>
> > **W3:** _How exactly the assumptions one to five are more general compared with the assumptions in the previous results and how this impact the theoretical proof and analysis._
>
> Thank you for your comments.
> - Please refer to ***Common
>     Response 1*** for a detailed comparison of the assumptions with
>     existing works.
>
> Below, we outline the roles of our assumptions in
>     the proof and analysis:
>
>  -   **Implications of Assumption 1-5 for theoretical analysis.**
>         Based on the discussion in W1, Assumption 1-5 are more general
>         than those conditions required in previous results. Below, we
>         summarize their implications for the theoretical analysis.
>
>         -   **Assumption 1 and 2 $\Rightarrow$ Theorem 2.1.** Assumption
>             1 and 2 are made to obtain the almost sure boundedness of
>             iterates $\lbrace x_k\rbrace$ in Theorem 2.1.
>
>             -   **(obtain ineqaulity (B.1) from Assumption 1).**
>                 Assumption 1 is used to derive inequality (B.1) (Line
>                 763 in the revised version), which can be seen as a
>                 stochastic analog of the three-term inequality for
>                 deterministic PPA (see Bertsekas, 2015, Proposition
>                 5.1.2).
>
>             -   **(obtain inequality (2.5) from Assumption 1 and 2).**
>                 Together, Assumption 1 and 2 yield inequality (2.5) in
>                 Theorem 2.1. The coefficient
>                 $\frac{\sigma_\phi^2}{m}+\gamma^2$ in the second term
>                 arises from the variance of subgradients of the
>                 composite component function $\varphi(\cdot;s)$ in the
>                 optimal solution set, the minibatch size $m$, and the
>                 accuracy parameter $\epsilon_k = \gamma \alpha_k^2$ used
>                 for solving subproblem.
>
>             The stability of isPPA is established by applying the
>             supermartingale convergence lemma to inequality (2.5).
>
>         -   **Assumption 1-3 $\Rightarrow$ Theorem 2.2.** Assumption 3
>             imposes a local Lipschitz condition on $f(\cdot;s)$, which,
>             combined with Assumption 1 and 2, leads to the almoset sure
>             convergence of $\lbrace x_k\rbrace$ to an optimal solution $x^{*}$ in
>             Theorem 2.2.
>
>             -   **Almost sure boundedness and local Lipschitz
>                 condition.** By Theorem 2.1, Assumption 1 and 2 ensure
>                 $\sup_{k}\Vert x_k\Vert_2 < \infty$ with probability 1.
>                 Then, it suffices to prove the result of Theorem 2.2 on
>                 the event that $\sup_{k}\Vert x_k\Vert_2 < \infty$
>                 holds, allowing us to assume only the local Lipschitz
>                 condition (Assumption 3) instead of a global Lipschitz
>                 condition.
>
>             -   **Obtain inequality (B.10).** On the event that
>                 $\sup_{k}\Vert x_k\Vert_2 < \infty$ holds, Assumption 1
>                 and 3 imply inequality (B.3) (Lines 772-774 in the
>                 revised version), which further leads to inequality
>                 (B.10) (Lines 882-884 in the revised version). The
>                 coefficient $\rho_{f,m,U} L_F(U)^2$ in this inequality
>                 arises from the minibatch size $m$ and the local
>                 Lipschitz condition on $f(\cdot;s)$.
>
>             The remaining proof of Theorem 2.2 is based on inequality
>             (B.10) and follows a similar argument to the proof of
>             (Bertsekas, 2011, Proposition 9).

---

> ### Author Response · Authors · 2024-11-26
> **Response to Weakness 3 of Reviewer Lrjb (continued)**
>
> -   **Implications of Assumption 1-5 for theoretical analysis.** _(continued)_
>
>        -   **Assumption 1-4 $\Rightarrow$ Theorem 2.3, Theorem 2.4,
>             Corollary 2.5 and Corollary 2.7.** The convergence rate
>             bounds in Theorem 2.3 and 2.4 are derived using Lemma
>             C.1-C.3 and Claim C.1 in Appendix C.
>
>             -   **Obtain Claim C.1.** Assumption 1, 3 and 4 lead to the
>                 inequality in Line 1256 of the revised version. Here,
>                 the coefficient $\tilde{C}_{f,m,U,c_1,\gamma}(\alpha_k)$
>                 depends on the local Lipschitz condition in Assumption
>                 3, the minibatch size $m$, the quadtratic growth
>                 condition in Assumption 4, the accuracy parameter
>                 $\epsilon_k = \gamma \alpha_k^{\frac{3}{2}}$, and the
>                 stepsize $\alpha_k$. The term $\frac{1}{1+sc_1\alpha_k}$
>                 is related to the quadtratic growth condition on $\phi$.
>
>             -   **Remove condition $\sup_k \Vert x_k\Vert_2 < \infty$.**
>                 As noted in Lines 334-340 of the revised version, for
>                 isPPA with square summable diminishing stepsizes
>                 $\alpha_k$ and accuracy parameter
>                 $\epsilon_k = \gamma \alpha_k^2$, Theorem 2.1 ensures
>                 the almost sure boundedness of $\lbrace x_k\rbrace$ under
>                 Assumption 1 and 2. Consequently, all the results in
>                 Theorem 2.4 and Corollary 2.5 hold almost surely, even
>                 if the statement $\sup_k \Vert x_k\Vert_2 < \infty$
>                 (Line 296 in the revised version) is removed.
>
>             -   **Replace quadratic growth condition with its localized
>                 version.** As noted in Lines 340-344 of the revised
>                 version, under Assumption 1-3, Theorem 2.2 ensures
>                 $\mathrm{dist}(x_k,\mathcal{X}^{*}) \to 0$ almost
>                 surely. This allows the quadratic growth condition
>                 (Assumption 4) to be replaced with its localized version
>                 (Assumption 5) while preserving the validity of the
>                 results in Theorem 2.4 and Corollary 2.5 for
>                 sufficiently large $k$.
>
>             By focusing on dominant terms, Corollary 2.5 is deduced from
>             Theorem 2.4, and Corollary 2.7 follows immediately from
>             Corollary 2.5 and Lemma 2.6.
>
> ------
>
> **Reference**
>
> Dimitri Bertsekas. *Convex optimization algorithms*. Athena Scientific, Belmont, Massachusetts, 2015.

---

> ### Author Response · Authors · 2024-11-26
> **Response to Questions of Reviewer Lrjb**
>
> ------
> #  Questions:
> ------
>
> > **Q1:** _remarks on how the assumptions in
>     this submission is weaker than the assumptions from previous papers._
>
> Thank you for this question and please refer to ***W1*** for a
>     detailed comparison.
>
> ------
> > **Q2:** _compare more about the theoretical analysis in this work with the previous works._
>
> We appreciate your feedback. Please refer to ***W2*** for
>     further details.

---

### Official Review · Reviewer_XLfK · 2024-11-03

**Soundness:** 3
**Presentation:** 3
**Contribution:** 3
**Rating:** 6
**Confidence:** 4

**Summary:**

The paper considers the inexact stochastic proximal point algorithm (isPPA) for solving convex stochastic composite optimization problems.

The authors relax the convergence assumptions by removing the requirements for smoothness and strong convexity, instead imposing a local Lipschitz condition and a quadratic growth condition on the objective function.

They establish non-ergodic (last-iterate) convergence rates for the algorithm.

Finally, the authors present empirical results to show the asymptotic convergence rates for the distance to the solution set and the KKT residual.

**Strengths:**

**S1**. This paper proposes isPPA to solve nonsmooth convex stochastic optimization problems under weaker assumptions of **a local Lipschitz continuity condition** and **a quadratic growth condition**, broadening its applicability in nonsmooth settings.

**S2**. This paper provides a tight analysis of the last-iterate convergence rates for the distance to the optimal solution set and the KKT residual, as shown in Theorems 2.4 and 2.7.

**S3**. Using the simple regularized Frechet mean problem, the authors establish the lower bounds for the convergence rate of the distance to the optimal solution set. By comparing the lower and upper bounds, they show that the asymptotic convergence rate is tight up to constant factors.

**Weaknesses:**

**W1.** Algorithm 1 is a double-loop algorithm, as it relies on another optimization algorithm to solve each subproblem. Such algorithms are rarely used; instead, single-loop algorithms, where each iteration precisely solves the subproblem, are more common. Although the algorithm specifies the required accuracy for each subproblem, these accuracy estimates tend to be conservative and are seldom used in practice, as they can significantly reduce the overall efficiency of the algorithm.

**W2.** For general non-smooth convex optimization problems, the paper does not provide the required iteration complexity for solving the subproblems.

**W3.** The authors claim that the algorithm requires weaker conditions than all existing methods. To support this claim, it would be beneficial to include a practical machine learning example where other methods fail, but the proposed method succeeds. The two examples provided in the experiments do not fully support the claim of requiring weaker assumptions.

**W4.** The authors mention in Lines 68-69 that the Lipschitz continuity condition is violated by the elastic net regularizer. I think this statement is inaccurate; a function that is continuously differentiable on a closed compact set is indeed Lipschitz continuous. Given that the iterative sequence is assumed to be bounded (see Lines 95 and 296), the elastic net regularizer should also be globally Lipschitz.


**W5.** Typo. Line 440: It should be "iteration(k)".

**Questions:**

I would like to ask two questions regarding the nonconvex scenario, as it is more general and widely applicable.

**Q1.** Referring to Lines 84-85, why is the proposed methodology more readily extendable to nonconvex settings?

**Q2.** How does the proposed algorithm compare to the proximal point method in the cited paper *Davis, D., & Drusvyatskiy, D. (2019). Stochastic model-based minimization of weakly convex functions*, which is designed for minimizing weakly convex functions?

---

> ### Author Response · Authors · 2024-11-26
> **Response to Weaknesses 1-5 of Reviewer XLfK**
>
> We thank the reviewer for acknowledging the soundness, quality of
> presentation and significance of the contributions of this work. In what
> follows, we address the weaknesses and answer the concerns raised by the
> reviewer.
>
> ------
>
> # Weakness:
>
> ------
>
> > **W1:** _double-loop algorithm and overall efficiency._
>
> Thank you for your comments. While the PPA is a
>     double-loop algorithm, it holds significant interest in optimization
>     and has many potential applications in machine learning. Please see
>     the ***Common Response 2*** for details.
>
> ------
>
> > **W2:** _iteration complexity for solving the subproblem._
>
> Thank you for your comments. Please refer to ***Common
>     Response 3*** for details. As an example, we explicitly present the
>     overall iteration complexity when using the gradient descent
>     algorithm. Additionally, we provide general guidelines for designing
>     algorithms to solve the subproblem while ensuring that the iteration
>     complexity remains valid.
>
> ------
>
> > **W3:** _mild assumptions and examples provided in the experiments._
>
> Thank you for your comments. Please refer to ***Common
>     Response 1*** for details. In that reply, we provide a comparison with
>     existing works, including several examples that highlight
>     differences in assumptions and requirements for the subproblem in
>     convergence analysis.
>
> ------
>
> > **W4:** _local Lipschitz condition and boundedness assumption._
>
> Thank you for your comments and suggestions. We agree
>     with your statement on the Lipschitz continuity on a compact set
>     (contained in the domain of the objective function). We want to
>     clarify the arguments in Lines 68-69 of the revised version. Here,
>     we mean that the elastic net regularizer is not globally Lipschitz
>     continuous on its entire domain. Regarding the boundedness
>     assumption, we have established in Theorem 2.1 that if
>     $\sum_{k}\alpha_k^2 < \infty$, the boundedness assumption (see Line
>     95 and 296 in the revised version) holds almost surely.
>     Consequently, under this condition, our results in Theorem 2.4
>     remain valid almost surely without requiring the global Lipschitz
>     continuity condition and the boundedness assumption of the sequence.
>     It is also worth mentioning that for other choices of $\alpha_k$
>     (e.g., constant step sizes), additional techniques would be required
>     to relax this boundedness requirement. We will revise this section
>     by including a remark to clarify this point.
>
> ------
>
> > **W5:** _typo in Line 440._
>
> Thank you for pointing it out. We will modify it in the revision.

---

> ### Author Response · Authors · 2024-11-26
> **Response to Question 1 of Reviewer XLfK**
>
> ------
>
> # Questions:
>
> ------
>
> > **Q1:** _proposed methodology more readily extendable to nonconvex settings._
>
> Thank you for your question.
> - While some existing works
>     focus on ergodic convergence, our result establishes last-iterate
>     convergence.
> - ***Ergodic convergence*** analyzes the behavior of the
>     averaged iterate, defined as
>     $\overline{x}\_k \triangleq \sum_{i=1}^k w_i x_i$, where
>     $w_i\in [0,1]$ and $\sum_{i=1}^k w_i = 1$. A standard approach for
>     establishing the ergodic convergence rate relies on Jensen's
>     inequality:
>     $$\phi(\sum_{i=1}^k w_i x_i) \leq \sum_{i=1}^k w_i \phi(x_i).$$
>     However, this inequality may fail in nonconvex settings, limiting
>     the applicability of ergodic convergence results.
>
> - In contrast, proving ***last-iterate convergence*** does not depend on
>     such inequalities. This independence makes last-iterate convergence
>     results easier to generalize to nonconvex settings compared to those
>     for ergodic convergence.

---

> ### Author Response · Authors · 2024-11-26
> **Response to Question 2 of Reviewer XLfK**
>
> ------
>
> > **Q2**: _comparison between isPPA and (Davis & Drusvyatskiy, 2019)._
>
> Thank you for your question. Our method (isPPA) differs
>     from the work in (Davis & Drusvyatskiy, 2019) from the following
>     perspectives.
>
>  -   **Solving subproblem: exact in (Davis & Drusvyatskiy, 2019) vs.
>         inexact in isPPA.** In (Davis & Drusvyatskiy, 2019), the
>         convergence analysis assumes that subproblems are solved
>         exactly, which is satisfied by the phase retrieval and blind
>         deconvolution problems presented in their numerical experiments.
>         Conversely, the proposed algorithm isPPA in our paper allows
>         subproblems to be solved inexactly under reasonable stopping
>         criteria, acknowledging the practical difficulty of solving
>         subproblems exactly.
>
>   -   **Algorithm scope: model-based algorithms in (Davis &
>         Drusvyatskiy, 2019) vs. isPPA.** (Davis & Drusvyatskiy, 2019)
>         analyzes to a wide range of stochastic algorithms-including
>         stochastic proximal point method, stochastic proximal
>         subgradient method, and stochastic prox-linear method. In
>         contrast, our paper focuses specifically on generalizing the
>         standard sPPA (i.e., exact stochastic PPA with minibatch size
>         $m=1$) to the inexact stochastic PPA with minibatch size
>         $m\in \mathbb{Z}_{+}$, and providing convergence guarantees.
>
> -   **Convergence rate: ergodic in (Davis & Drusvyatskiy, 2019) vs.
>         last-iterate for isPPA.** The results in (Davis & Drusvyatskiy,
>         2019, Theorems 4.4 and 4.5) provide nonasymptotic bounds on the
>         ergodic convergence rate, in terms of the expected function
>         value gap, for the stochastic proximal point method introduced
>         in (Davis & Drusvyatskiy, 2019, Section 4.2) (referring to the
>         exact sPPA with minibatch size one under our definition). In
>         contrast, our work focuses on the last-iterate convergence rate,
>         deriving nonasymptotic and asymptotic bounds measured by the
>         expected squared distance to the optimal solution set and the
>         expected KKT residual.
>
>   -   **Analysis based on different assumptions.**
>
>       -   **Weak convexity vs. convexity.** The theoretical guarantee
>             in (Davis & Drusvyatskiy, 2019, Theorem 4.3) requires only
>             weak convexity, allowing their results to apply to a broader
>             class of machine learning problems compared to the convexity
>             condition (Assumption 1) in our analysis. However, they
>             assume a (global) Lipschitz condition on each $f(\cdot; s)$
>             over a convex open set containing $\mathrm{dom}(r)$\-a
>             condition violated by the Lasso regression model. Our paper
>             relaxes this requirement to a local Lipschitz condition
>             (Assumption 3) (for isPPA with summable diminishing
>             stepsizes, as mentioned earlier, the statement
>             $\sup_k \Vert x_k\Vert_2 < \infty$ in Theorem 2.4 and
>             Corollary 2.5 can be removed).
>
>       -   **Additional Assumption 2 for isPPA.** In addition, we
>             impose Assumption 2 to ensure the stability of isPPA. While
>             this assumption is not required in (Davis & Drusvyatskiy,
>             2019), it is generally satisfied by finite-sum regularized
>             regression models (see Remark 1.1).
>
>       -   **Strong convexity vs. quadratic growth condition.**
>             Finally, (Davis & Drusvyatskiy, 2019, Theorem 4.5) assumes
>             conditions leading to strong convexity of $\phi$, which is
>             stricter than the quadratic growth conditions (Assumptions 4
>             and 5) used in our work.
>
>    In summary, (Davis & Drusvyatskiy, 2019) explores a broad family of
>     stochastic model-based algorithms, while our isPPA generalizes the
>     stochastic proximal point method within this family. the results in
>     these two papers involve different forms of convergence rate
>     (ergodic convergence in (Davis & Drusvyatskiy, 2019) vs.
>     last-iterate convergence in our paper) and are based on different
>     assumptions. These contributions are complementary. In the future,
>     we will consider extending the convexity condition (in Assumption 1)
>     to weak convexity, enabling isPPA to address a wider range of
>     optimization problems.

---

### Author Response · Authors · 2024-11-26
**Common Response**

We sincerely thank the reviewers for their valuable comments and suggestions. Based on the feedback provided, we have revised our manuscript accordingly. In the uploaded revised version, all changes are highlighted in **blue** for clarity. When referring to specific line numbers in our responses (e.g., Line 6), these correspond to the line numbers in the **revised version** of the manuscript. Below, we outline three responses to address commonly raised concerns.

---

> ### Author Response · Authors · 2024-11-26
> **Common Response 1: comparison with existing results**
>
> The following table summarizes the assumptions made in the existing work
> for the convergence analysis of sPPA-type schemes:
>
> |          Literature          |  Assumptions |              |                                 |                                |              |              |              |              |     Form     |              |                    Bound                    |
> |:----------------------------:|:------------:|:------------:|:-------------------------------:|:------------------------------:|:------------:|:------------:|:------------:|:------------:|:------------:|:------------:|:-------------------------------------------:|
> |                              |  Condition1  |              |                                 |                                |  Condition2 |              |  Condition3 |              |     last     |    ergodic   |                                             |
> |                              |    Lip.sm    |    $C^2$     |               Lip               |             loc.Lip            |     s.cvx    |    qd.grow   |     exact    |    inexact   |              |              |                                             |
> |    (R\&B, 2014) Theorem 4    | $\checkmark$ |              |                                 |                                | $\checkmark$ |              | $\checkmark$ |              | $\checkmark$ |              |          $\mathcal{O}(\frac{1}{k})$         |
> |     (T, 2016)  Theorem 1     |              | $\checkmark$ | $\checkmark$  (on $f(\cdot;s)$) |                                |              |              | $\checkmark$ |              | $\checkmark$ |              |          $\mathcal{O}(\frac{1}{k})$         |
> |    (P\&N, 2018) Theorem 14   |              |              |                                 |                                | $\checkmark$ |              | $\checkmark$ |              | $\checkmark$ |              |          $\mathcal{O}(\frac{1}{k})$         |
> | (A\&D, 2019) Proposition 5.3 |              |              |                                 |                                | $\checkmark$ |              | $\checkmark$ |              | $\checkmark$ |              |       $\mathcal{O}(\frac{\log(k)}{k})$      |
> |   (D\&D, 2019) Theorem 4.5   |              |              |  $\checkmark$ (on $f(\cdot;s)$) |                                | $\checkmark$ |              | $\checkmark$ |              |              | $\checkmark$ |  $\mathcal{O}(\frac{1}{k^2} + \frac{1}{k})$ |
> |     (P, 2021) Theorem 4.3    |              |              |                                 |                                |              |              | $\checkmark$ |              | $\checkmark$ |              |          $\mathcal{O}(\frac{1}{k})$         |
> |    (Y\&L, 2023) Theorem 1    | $\checkmark$ |              |                                 |                                |              | $\checkmark$ | $\checkmark$ |              |              | $\checkmark$ | $\mathcal{O}(\frac{1}{k^2} + \frac{1}{mk})$ |
> |    (Y\&L, 2023) Theorem 10   | $\checkmark$ |              |      $\checkmark$  (on $r$)     |                                |              | $\checkmark$ |              | $\checkmark$ |              | $\checkmark$ | $\mathcal{O}(\frac{1}{k^2} + \frac{1}{mk})$ |
> |           Our paper          |              |              |                                 | $\checkmark$ (on $f(\cdot;s)$) |              | $\checkmark$ |              | $\checkmark$ | $\checkmark$ |              |          $\mathcal{O}(\frac{1}{k})$         |

---

> ### Author Response · Authors · 2024-11-26
> **Common Response 1: comparison with existing results (continued)**
>
> **Assumptions comparison.** Below, we provide clarifications regarding the assumptions required in this paper compared to those in prior works. The following table illustrates the relationship between the assumptions required for the convergence analysis of stochastic PPA.
>
> |  Assumptions |            Relation           |                                                                                                 Example                                                                                                |
> |:------------:|:-----------------------------|:-------------------------------------------------------------------------------------------------------------------------------------------------------------------------------------------------------------|
> | Condition 1  | Lip.sm $\Rightarrow$ loc.Lip | $\Vert Ax-b\Vert_1 + \frac{\lambda}{2}\Vert x\Vert_2^2$  - not Lipschitz smooth - satisfies local Lipschitz condition                                                                                         |
> |              | $C^2$ $\Rightarrow$ loc.Lip   | $\Vert Ax-b\Vert_1 + \frac{\lambda}{2}\Vert x\Vert_2^2$  - not $C^2$  - satisfies local Lipschitz condition                                                                                                   |
> |              | Lip $\Rightarrow$ loc.Lip     | $\frac{1}{2}\Vert Ax - b\Vert_2^2+\frac{\lambda}{2}\Vert x\Vert_2^2$  - not globally Lipschitz on $f(\cdot;s)$ & not globally Lipschitz on $r$ - satisfies local Lipschitz condition on $f(\cdot;s)$ and $r$ |
> |  Condition 2 | s.cvx $\Rightarrow$ qd.grow   | $\frac{1}{2}\Vert Ax - b\Vert_2^2+\lambda\Vert x\Vert_1$ with $\mathrm{rank}(A) < d$ ($A\in \mathbb{R}^{n\times d}$)  - not strongly convex  - satisfies local quadratic growth condition                     |
> |  Condition 3 | exact $\Rightarrow$ inexact   | $\frac{1}{2}\Vert Ax - b\Vert_2^2+\lambda\Vert x\Vert_1$  - solving each subproblem exactly difficult to achieve in practice - each subproblem can be solved inexactly (Claim F.1 in Appendix F)              |
>
>
> Here, we use the following abbreviations:
>
> -   $k$: iteration count; $m$: minibatch size;
>
> -   Condition 1:
>
>     -   Lip.sm: Lipschitz smoothness (i.e., Lipschitz continuous
>         gradient);
>
>     -   $C^2$: twice continuously differentiable;
>
>     -   Lip: (global) Lipschitz continuity over the entire domain;
>
>     -   loc.Lip: local Lipschitz continuity;
>
> -   Condition 2
>
>     -   s.cvx: (restricted) strong convexity;
>
>     -   qd.grow: quadratic growth condition;
>
> -   Condition 3
>
>     -   exact: each subproblem solved exactly;
>
>     -   inexact: each subproblem solved inexactly;
>
> -   Form:
>
>     -   last: last-iterate convergence rate;
>
>     -   ergodic: ergodic convergence rate;
>
> -   $\checkmark$ under \"Assumptions\": the assumption is required to
>     derive the bound on the rate of convergence;
>
> -   $\checkmark$ under \"Form\": the bound is provided for last-iterate
>     convergence (last) or ergodic convergence (ergodic).
>
> Note that
>
> -   By Remark 2.1 of our submitted manuscript, Assumption 1 and
>     Assumption 2 in our paper are satisfied by commonly used finite-sum
>     regularized regression models with real-valued nonnegative convex
>     component functions and real-valued convex coercive regularizers.
>
> -   Assumption 3 in our paper imposes only a local Lipschitz condition
>     on component functions $f(\cdot;s)$.
>
> -   Assumption 4 in our paper assumes the quadratic growth condition on
>     $\phi$, which can be replaced with its localized version
>     (Assumption 5) when using square summable stepsizes
>     $\alpha_k = \alpha_0 k^{-\beta}$ ($\beta\in (\frac{1}{2},1]$) (see
>     Lines 334-344 in the revised version).
>
> Based on the above discussion, one can see that the proposed assumptions
> are reasonable and mild.

---

> ### Author Response · Authors · 2024-11-26
> **References**
>
> ------
>
> (A&D, 2019) Hilal Asi and John C Duchi. Stochastic (approximate) proximal point methods: Convergence, optimality, and adaptivity. *SIAM Journal on Optimization*, 29(3):2257-2290, 2019.
>
> (D&D, 2019) Damek Davis and Dmitriy Drusvyatskiy. Stochastic model-based minimization of weakly convex functions. *SIAM Journal on Optimization*, 29(1):207-239, 2019.
>
> (P, 2021) Andrei Patrascu. New nonasymptotic convergence rates of stochastic proximal point algorithm for stochastic convex optimization. *Optimization*, 70(9):1891-1919, 2021.
>
> (P&N, 2018) Andrei Patrascu and Ion Necoara. Nonasymptotic convergence of stochastic proximal point methods for constrained convex optimization. *Journal of Machine Learning Research*, 18(198):1-42, 2018.
>
> (R&B, 2014) Ernest K Ryu and Stephen Boyd. Stochastic proximal iteration: a non-asymptotic improvement upon stochastic gradient descent. *Author website*, 2014. URL https://web.stanford.edu/ boyd/papers/spi.html.
>
> (T, 2016) Panos Toulis, Dustin Tran, and Edo Airoldi. Towards stability and optimality in stochastic gradient descent. In *Artificial Intelligence and Statistics*, pp. 1290-1298. PMLR, 2016.
>
> (Y&L, 2023) Xiao-Tong Yuan and Ping Li. Sharper analysis for minibatch stochastic proximal point methods: Stability, smoothness, and deviation. *Journal of Machine Learning Research*, 24(270):1-52, 2023.

---

> ### Author Response · Authors · 2024-11-26
> **Common Response 2: motivation for the theoretical analysis of isPPA**
>
> The Proximal Point Algorithm (PPA) is a classical optimization method
> that has been extensively studied from both theoretical and practical
> perspectives. In the deterministic case, PPA achieves superlinear
> convergence asymptotically (Rockafellar, 1976, Theorem 2) and can attain
> arbitrarily fast convergence, as characterized by the iteration
> complexity $\mathcal{O}(\frac{1}{\sum_{i=1}^k \alpha_i})$ (Güler, 1991,
> Theorem 2.1). To handle large-scale datasets, extending deterministic
> algorithms to stochastic variants with convergence guarantees has become
> a standard approach, including the PPA. However, the existing
> convergence and convergence rate analysis of stochastic PPA still
> require some restrictive assumptions that may not be satisfied in
> applications. This paper contributes to addressing this theoretical
> question and proposing an implementable algorithm under reasonable
> assumptions (see the Common Response 1 for the detailed comparison with
> existing works). In addition, this work highlights a fundamental
> discrepancy between the deterministic and stochastic versions of PPA.
> Specifically, deterministic inexact PPA can achieve linear convergence
> under the quadratic growth condition when stepsizes with positive lower
> bound are used. In contrast, we show there is a gap in terms of
> convergence rate between PPA and isPPA using the regularized Fréchet
> mean problem (3.1) in our manuscript. More details can be found in
> Section 3 of our manuscript.
>
> Indeed, both the PPA and isPPA are double-loop algorithms, requiring a
> fast and efficient solution to the inner problem for practical
> applicability. Recent research has shown that efficient algorithms can
> be designed to solve the inner problems due to the following two
> reasons:
>
> -   **The subproblem is strongly convex.** Since our algorithm
>     incorporates a squared $\ell_2$ term, each subproblem becomes
>     strongly convex, enabling efficient solutions using a wide range of
>     algorithms. For instance, the gradient descent algorithm achieves
>     linear convergence for solving the inner problem. In this case, the
>     overall complexity only incurs an additional logarithmic term,
>     $\log(\frac{1}{\epsilon})$ (as detailed in Common Response 3).
>     Moreover, since $\log(k)=o(k^\xi)$ for any $\xi > 0$, the overall
>     complexity of isPPA is dominated by the convergence rate of the
>     outer loop.
>
> -   **Fast solver leveraging the problem's specific structures.** When
>     the variable $x$ exhibits sparsity, the inner problem can be
>     simplified by solving a smaller linear system or applying a sieving
>     strategy. This approach has been successfully implemented in
>     state-of-the-art software, such as SDPNAL\+ (Yang et al., 2015), an
>     award-winning tool for solving semi-definite programming problems.
>     Similarly, leveraging problem-specific structures in the inner loop
>     can lead to significant speedups in practical applications. For
>     example, SSNAL (Li et al., 2018) optimizes the inner loop for
>     solving Lasso problems effectively.
>
> ------
>
> **References**
>
> Osman Güler. On the convergence of the proximal point algorithm for convex minimization. *SIAM Journal on Control and Optimization*, 29(2):403-419, 1991.
>
> Xudong Li, Defeng Sun, and Kim-Chuan Toh. A highly efficient semismooth newton augmented lagrangian method for solving lasso problems. *SIAM Journal on Optimization*, 28(1):433-458, 2018.
>
> R Tyrrell Rockafellar. Monotone operators and the proximal point algorithm. *SIAM Journal on Control and Optimization*, 14(5):877-898, 1976.
>
> Liuqin Yang, Defeng Sun, and Kim-Chuan Toh. SDPNAL\+: a majorized semismooth Newton-CG augmented Lagrangian method for semidefinite programming with nonnegative constraints. *Mathematical Programming Computation*, 7(3):331-366, 2015.

---

> ### Author Response · Authors · 2024-11-26
> **Common Response 3: iteration complexity**
>
> We first provide a simple example to illustrate the overall iteration
> complexity. We will discuss this in a more general sense below. When
> each $f(\cdot;s) + r(\cdot)$ is convex and L-smooth with a unified
> $L > 0$, we can apply the gradient descent (GD) algorithm to solve the
> subproblems with a linear convergence rate. Note that the subproblem is
> strongly convex due to the addition of a squared $\ell_2$-norm in isPPA.
> In this scenario, we can obtain an overall complexity of
> **$\mathcal{O}((\frac{1}{\epsilon})^{\frac{1}{\beta}}\log(\frac{1}{\epsilon}))$**.
> Let $N_{\mathrm{inner}}(k)$ denote the inner-loop iteration complexity
> bound of GD for solving the subproblem (1.3) at the outer iteration $k$.
> Since the subproblem is strongly convex, under the setting of Corollary
> 2.5, we have
> $$N_{\mathrm{inner}}(k) \leq \mathcal{O}\left( k^{-\beta} + k^{-\beta} \log(k)\right),$$
> where a detailed derivation can be found below in a more general sense.
>
> Let $N_{\mathrm{outer}}$ denote the number of outer-loop iterations
> (isPPA) required to obtain $x_{N_{\mathrm{outer}}}$ such that
> $\mathbb{E}[\mathrm{dist}(x_k,\mathcal{X}^{*})^2] \leq \epsilon$ for
> some $\epsilon>0$. Then, $$\begin{aligned}
>         N_{\mathrm{outer}} \leq \mathcal{O}\left(\left(\frac{1}{\epsilon}\right)^{\frac{1}{\beta}}\right).
>     \end{aligned}$$ Using (C.7) and (C.9b) from Lemma C.4 in our
> submitted manuscript, the overall complexity $N_{\mathrm{total}}$
> satisfies
> $$N_{\mathrm{total}} \leq \mathcal{O}\left(N_{\mathrm{outer}}\log\left(N_{\mathrm{outer}}\right)\right) \leq \mathcal{O}\left(\left(\frac{1}{\epsilon}\right)^{\frac{1}{\beta}}\log\left(\frac{1}{\epsilon}\right)\right).$$
>
> **General principle for the algorithm design for solving subproblem.**
> It is worth noting that more efficient algorithms can be designed to
> solve the subproblem if the problem exhibits certain structures, as
> highlighted in our previous response. Suppose the method $\mathcal{M}$
> used to solve subproblem (1.3) satisfies:
> $$\Phi_{\alpha_k,x_k}(x_k^{(t)};S_k^{1:m}) - \Phi_{\alpha_k,x_k,S_k^{1:m}}^{*}\leq C_1 (1-\rho\_{\mathcal{M}}\^{(k)})^t$$
> where constant $C_1>0$ and $\rho\_{\mathcal{M}}\^{(k)}\in (0,1)$ (see Line
> 215 in the revised version for the definition of
> $\Phi_{\alpha_k,x_k}(\cdot;S_k^{1:m})$). Let $N_{\mathrm{inner}}(k)$
> denote the inner-loop complexity bound of $\mathcal{M}$ for solving the
> subproblem at iteration $k$. Based on (SCB), we derive:
>
> $$
> \begin{aligned}
>         N_{\mathrm{inner}}(k) \leq \inf\left\lbrace t\in \mathbb{Z}_{+}\ \vert\ C_1 e^{-\rho\_{\mathcal{M}}\^{(k)} t} \leq \frac{\epsilon_k^2}{2\alpha_k}\right\rbrace = \left\lceil \frac{1}{\rho\_{\mathcal{M}}\^{(k)}} \log\left(\frac{2C_1}{\gamma^2} \alpha_k^{1-2\delta}\right) \right\rceil = \left\lceil \frac{1}{\rho\_{\mathcal{M}}\^{(k)}} \log\left(\frac{2C_1}{\gamma^2\alpha_0^{2\delta - 1}} k^{(2\delta-1)\beta}\right) \right\rceil,
> \end{aligned}
> $$
>
> where $\alpha_k = \alpha_0 k^{-\beta}$
> ($\alpha_0 > 0$, $\beta\in (0,1])$) and
> $\epsilon_k = \gamma \alpha_k^{\delta}$ ($\gamma > 0$,
> $\delta\geq \frac{3}{2}$). For example, when $\mathcal{M}$ is GD, it
> holds that $\rho\_{\mathcal{M}}\^{(k)} = \frac{\alpha_k^{-1}}{L_k}$ for
> some $L_k>0$. Assuming that $L_k \leq L$ for some $L>0$, we have
> $$\begin{aligned}
>         N_{\mathrm{inner}}(k) &\leq \left\lceil A_1 k^{-\beta} + A_2 k^{-\beta} \log(k)\right\rceil,
>     \end{aligned}$$ where
> $A_1 \triangleq L\alpha_0 (\log(2C_1\gamma^{-2}) + (1-2\delta) \log(\alpha_0)$
> and $A_2 \triangleq L\alpha_0 (2\delta - 1) \beta$.
>
> As this work focuses primarily on establishing the tight convergence
> rate of the isPPA, and given that the manuscript already exceeds 40
> pages, we plan to systematically investigate these subproblem-solving
> methods for specific problems in future work.

---

### Author Response · Authors · 2024-12-02

Dear Reviewers,

As the discussion phase comes to a close, we sincerely hope that our additional response has addressed your concerns. If so, we would greatly appreciate your consideration in raising the score. If there are any remaining concerns, we would be grateful if you could let us know, and we will make every effort to further refine and enhance our work.

Best regards,

The Authors of Paper 6540

---

### Meta-Review · Area_Chair_XXyP · 2024-12-14

**Metareview:**

This paper studies inexact stochastic proximal point algorithm (isPPA). In particular, the authors established the stability and almost sure convergence of isPPA under mild assumptions, which are weaker than the assumptions in existing results. While the results are interesting and sound, the authors are advised to address some concerns raised by the reviewers in the final version.

**Additional Comments On Reviewer Discussion:**

Clarified some questions on iteration complexity.

---

### Decision · Program_Chairs · 2025-01-22

Accept (Poster)